# Mechanism and mitigation of stainless steel dissolution in LiFSI-based lithium-ion battery electrolytes

Peng Yan [1], Marian Cristian Stan [1], Kazem Zhour [2,3], Diddo Diddens [1], Christian Wölke[1], Rayan Guerdelli [1], Martin Winter [1,4] & Isidora Cekic-Laskovic [1] ✉

Lithium bis(fluorosulfonyl)imide has emerged as a promising alternative to lithium hexafluorophosphate as conducting salt in battery electrolytes due to its favorable physicochemical properties. However, its tendency to promote the dissolution of Al and stainless steel severely limits its practical application, particularly in lithium ion batteries operating above 4 V $vs.$ Li/Li$^+$. Here we show that the dissolution of SUS316 in lithium bis(fluorosulfonyl)imide-based electrolytes is governed by a synergistic mechanism involving trace Cl$^-$ impurities and FSI$^-$ anions. Cl$^-$ initiates localized pitting, while subsequent interactions between FSI$^-$ anions and dissolved iron species lead to the formation of soluble complexes, thereby extending the dissolution process. We further demonstrate that the dissolution can be effectively suppressed by adding lithium difluoro(oxalato) borate. The proposed mechanism involves preferential adsorption of oxalate anions at surface of stainless steel, which limits the access of aggressive anions. Additional improvement is achieved by incorporating more dissolution resistive SUS316L components, resulting in ≈300 cycles until 80 % state of health in silicon-graphite||LiNi$_{0.8}$Co$_{0.1}$Mn$_{0.1}$O$_2$ cells. Furthermore, this improvement has also been confirmed in silicon-graphite||LiNi$_{0.8}$Co$_{0.1}$Mn$_{0.1}$O$_2$ pouch cells.

In the pursuit of clean energy solutions, lithium-ion batteries (LIBs) have become the primary power source for electric vehicles due to their high energy density and efficiency[1–4]. Advanced electrode materials, such as nickel-rich positive electrodes and silicon-based negative electrodes, offer the potential to push the boundaries of energy density[5–7]. However, these electrodes are associated with intricate challenges, particularly concerning electrolyte compatibility and electrochemical stability[8–10]. The traditional state-of-the-art electrolyte, comprising lithium hexafluorophosphate (LiPF$_6$) as a lithium-conducting salt and a mixture of organic carbonate solvents, encounters limitations in the aforementioned cell chemistry. The

thermal instability of LiPF$_6$ limits its usefulness in high temperature applications, while its moisture sensitivity, resulting in HF formation, can trigger multiple detrimental reactions such as transition metal (TM) dissolution and dendrite growth[11,12]. This has led to the exploration of electrolyte formulations, with extensive effort dedicated to alternative lithium-conducting salts.

Among these, lithium bis(fluorosulfonyl)imide (LiFSI) has been considered as a promising candidate due to its chemical and thermal stability, as well as moisture resistance[13–16]. In addition, the FSI$^-$ anion, characterized by its lower binding energy with Li$^+$ cation, enhances solubility and ion mobility within the electrolyte. Furthermore, the FSI$^-$

[1]Helmholtz-Institute Münster (HI MS), IMD-4, Forschungszentrum Jülich GmbH, Corrensstraße 48, Münster, Germany. [2]Institute of Physical Chemistry, University of Münster, Corrensstraße 28, Münster, Germany. [3]Multiscale Modelling of Heterogeneous Catalysis in Energy Systems, RWTH Aachen University, Aachen, Germany. [4]MEET Battery Research Center, University of Münster, Corrensstraße 46, Münster, Germany. ✉e-mail: i.cekic-laskovic@fz-juelich.de

anion tends to be electrochemically reduced and generates LiF-rich inorganic compounds on the surface of the corresponding electrode, which is beneficial for forming an effective solid electrolyte interphase (SEI) on various negative electrodes, including lithium metal, graphite (Gr), and silicon-graphite (Si-C)[17,18]. However, a considerable concern regarding the use of LiFSI is the anodic dissolution of the Al current collector at potentials above 4.2 V *vs.* Li/Li$^+$[19]. Studies have shown that employing borate-salt as a co-salt or salt additive can substantially suppress Al dissolution[20–22]. While numerous studies focused on Al dissolution and strategies to suppress it, stainless steel (SUS) dissolution in the presence of LiFSI salts has not been equally addressed[23–28]. Given the widespread utilization of SUS in various battery formats, including cylindrical and coin cells, as well as other cell formats with SUS parts, resolving the issue of SUS dissolution broadens the use of LiFSI-based electrolytes at high voltages LIB applications[26,29,30]. Luo et al. proposed a mechanism of SUS dissolution in LiFSI-based electrolytes, emphasizing the combined action of Cl$^-$ and fluorosulfates (decomposition products from FSI$^-$) in attacking the SUS surface[24]. However, the specific contribution of Cl$^-$ ions and the precise role of LiFSI in this process remained insufficiently understood.

To address the issue of SUS dissolution, researchers have developed several strategies. Among them, the use of high concentration electrolyte (HCE) and localized high concentration electrolyte (LHCE) has been proposed to reduce the solubility of dissolution products[24]. However, the economic and environmental viability of HCE or LHCE remains a concern, and their effectiveness in completely preventing SUS dissolution is still under assessment. An alternative approach involves applying protective coatings on the SUS surface, such as the utilization of Al-clad/Al-CVD coin cell parts to replace the SUS analogs[23,27,31,32]. However, this method is not entirely effective, with evidence suggesting the simultaneous occurrence of Al and SUS dissolution in Al-clad SUS coin cell parts[27]. Therefore, it is necessary to find further strategies to completely mitigate the SUS dissolution, in which the understanding of the specific mechanism of SUS dissolution is crucial.

In this work, complementary electrochemical and surface characterization techniques were employed to study the mechanism of SUS dissolution in LiFSI-based electrolytes. Both Cl$^-$ and FSI$^-$ anions can destabilize the SUS surface and cause severe dissolution. Specifically, lithium difluoro(oxalato) borate (LiDFOB) has been added to the LiFSI-based electrolytes to mitigate the SUS dissolution. With electrochemical and *post mortem* analysis, the possible mechanism of LiDFOB in inhibiting SUS dissolution was proposed. Finally, the effectiveness of LiDFOB has been proven by improved galvanostatic cycling performance in Gr||LiNi$_{0.8}$Co$_{0.1}$Mn$_{0.1}$O$_2$ (NMC811) and Si-C||NMC811 coin cells as well as Si-C||NMC811 pouch cells.

## Results and discussion
### Impact of "aggressive anions" on SUS dissolution in LiFSI-based electrolyte

Chloride anion impurities are often remnants from the synthesis of LiFSI and are considered "aggressive anions" due to their corrosive nature toward metals[33–35]. Studies have shown that these impurities play a major role in the dissolution of the aluminum current collector in LIBs[13]. Furthermore, it has also been suggested that the Cl$^-$ anion is responsible for the dissolution of SUS in LiFSI-based electrolytes[24]. Despite these findings, a detailed understanding of how Cl$^-$ anion impurities specifically affect SUS dissolution remains limited. Therefore, to further elucidate the effect of Cl$^-$ anion on the SUS dissolution, we have enriched the concentration of Cl$^-$ anions by adding 700 ppm of lithium chloride (LiCl) to a mixture of EC: EMC (3:7 by weight), resulting in an electrolyte that is nearly saturated with LiCl (referred to as LiCl$_{sat}$). It should be noted that 7.49 ppm of Cl$^-$ anion impurities were detected in the LiFSI salt by ion chromatography (IC) measurements.

Linear sweep voltammetry (LSV) was carried out to determine the electrochemical stability of the electrolytes against SUS. The choice of SUS316 is particularly relevant as it is commonly used in cell housing and components, making it critical to understand its behavior in the presence of Cl$^-$ anions. Figure 1a shows the logarithmic value of the current density obtained during LSV using SUS316 as the working electrode with various electrolytes, where the primary and secondary passivation are observed across all the samples[24]. Notably, a pronounced current increase is observed at ≈3.5 V in the cell containing LiCl$_{sat}$ electrolyte, which can be attributed to the SUS dissolution triggered either by solvent decomposition or by the presence of Cl$^-$ anions. Organic carbonate-based solvents, as well as the solvent impurity species (e.g., ethylene glycol or methanol), may decompose above 3 V *vs.* Li/Li$^+$ and generate protons, which destabilize the native Al$_2$O$_3$ layer and initiate the Al dissolution[36–38]. However, the use of fluorinated carbonate solvents, which have higher oxidation stability and lower tendency for proton release, has confirmed that protons do not contribute to the SUS dissolution. The detailed discussion is provided in Supplementary Note 3. *Operando* mass spectrometry (MS) measurements did not confirm any chlorine evolution, as shown in Fig. S1a. Furthermore, an increase in current is observed at ≈4.2 V in the cell with pure LiFSI-based electrolyte (FSI$_1$DFOB$_0$), indicating FSI$^-$ anion induced dissolution. A slight current increase (as indicated in Fig. S2) is detected at ≈3.5 V with the FSI$_1$DFOB$_0$ electrolyte, which suggests the presence of trace amounts of Cl$^-$ anion impurities in the LiFSI salt. This complicates the analysis of the dissolution process, as both Cl$^-$ and FSI$^-$ anions may contribute interactively. To further validate the distinct dissolution process, LSV was carried out using a mixed electrolyte comprising LiCl$_{sat}$ and FSI$_1$DFOB$_0$. The results clearly show a two-stage dissolution process, confirming the separate contribution of Cl$^-$ and FSI$^-$ anions to the SUS dissolution process. Cyclic voltammetry (CV) measurements, performed at a slow scan rate of 0.1 mV$^{-1}$ confirm the distinct dissolution process arising from Cl$^-$ and FSI$^-$ anions. Cells with both FSI$_1$DFOB$_0$ and FSI$_1$DFOB$_0$ + LiCl$_{sat}$ electrolytes exhibit typical dissolution behavior, characterized by a pronounced increase in current during the reverse scan, forming a hysteresis loop (Fig. 1e, f). When comparing the first CV cycle, the cell with FSI$_1$DFOB$_0$ shows the onset of anodic current at ≈4.2 V, whereas the cell with FSI$_1$DFOB$_0$+LiCl$_{sat}$ shows two distinct anodic current rises at ≈3.5 V and ≈4.2 V (Fig. 1g). These features are consistent with the trends observed in the corresponding LSV measurements. Such a finding becomes particularly relevant when the potential of the working electrode reaches 4.2 V *vs.* Li/Li$^+$, which aligns with the standard charging voltage for nickel-rich positive electrodes, thus pinpointing the importance of investigating SUS dissolution at this voltage. Chronoamperometry (CA) measurements were conducted at 4.2 V to further investigate these effects using the various electrolytes. As illustrated in Fig. 1b, a continuous current increase was observed for both electrolytes, indicating ongoing SUS dissolution induced by the Cl$^-$ and FSI$^-$ anions. Notably, cells containing LiCl$_{sat}$ showed substantially higher current density (≈13.80 mA cm$^{-2}$ after 20 h) compared to cells with FSI$_1$DFOB$_0$ ( ≈ 0.08 mA cm$^{-2}$ after 20 h), indicating more severe SUS dissolution in the presence of Cl$^-$ anions, whereas no SUS dissolution was observed in the presence of baseline electrolyte (Fig. S3).

The results from the CA experiments clearly indicate that SUS dissolution is more pronounced in the cell with LiCl$_{sat}$ compared to the cell with FSI$_1$DFOB$_0$. However, subsequent analysis using scanning electron microscopy (SEM) revealed distinct differences between these samples. Figure 1c, d show SEM images of the SUS316 spacers after 20 h in CA measurement, illustrating the changes in surface morphology between the spacers harvested from cells with the two electrolytes. The surface of the spacer exposed to FSI$_1$DFOB$_0$ electrolyte appears very clean, whereas the surface of the spacer treated with LiCl$_{sat}$ is covered with large amounts of decomposition products. The

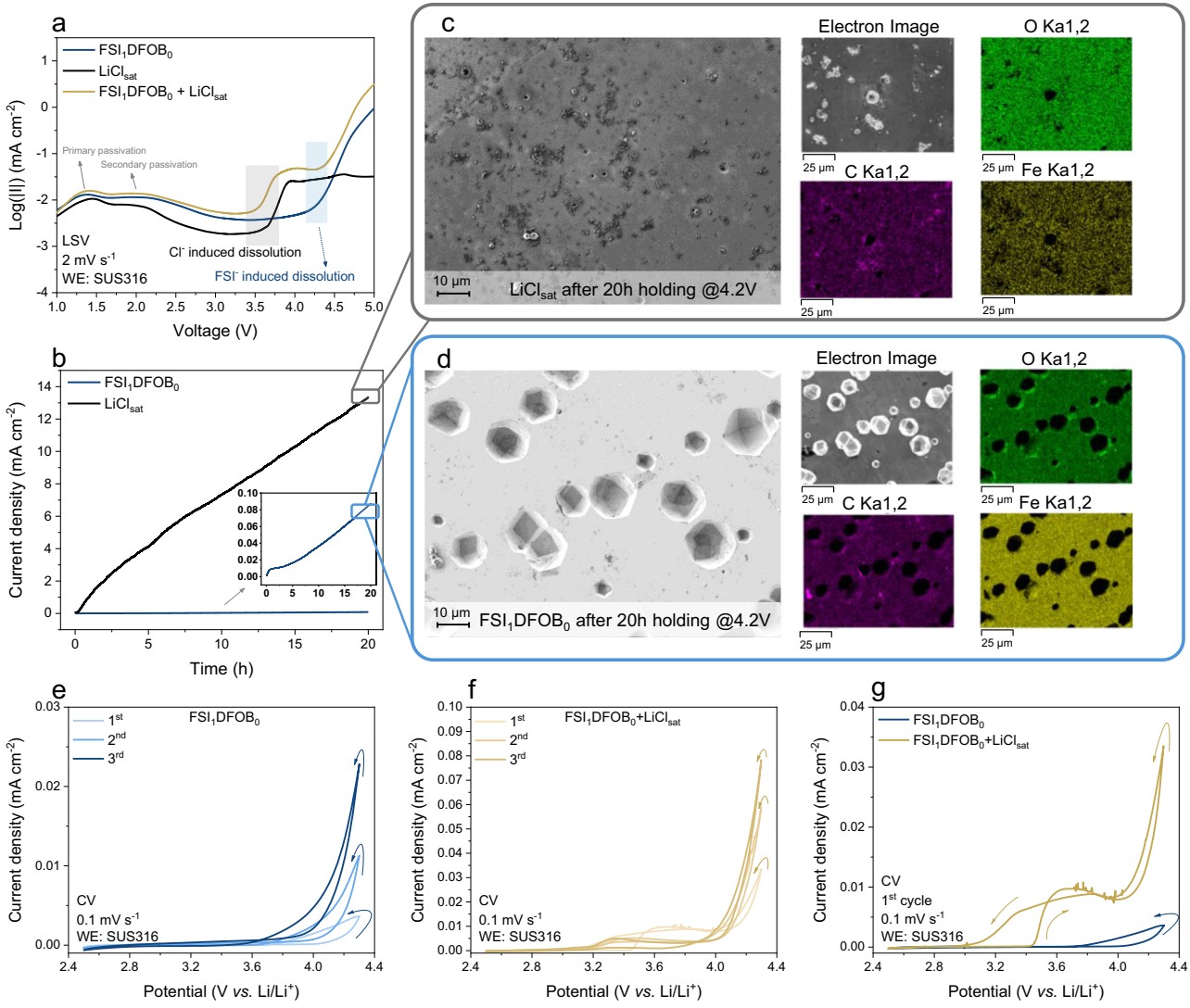

**Fig. 1 | Impact of aggressive anions on SUS dissolution in LiFSI-based electrolytes. a** Linear sweep voltammetry (LSV) curve of cells containing SUS316 as the working electrode (WE) with $FSI_1DFOB_0$, $LiCl_{sat}$ and $FSI_1DFOB_0 + LiCl_{sat}$ electrolyte. **b** Chronoamperometry (CA) curves of cells containing $FSI_1DFOB_0$ and $LiCl_{sat}$ electrolytes were recorded at a voltage of 4.2 V for 20 h. SEM and EDX images of polished SUS316 spacers harvested from cells after 20 h of CA measurements at 4.2 V and 20 °C with **c** $LiCl_{sat}$ and **d** $FSI_1DFOB_0$ electrolytes. Cyclic voltammetry (CV) curves of cells containing SUS316 as the working electrode with **e** $FSI_1DFOB_0$ and **f** $FSI_1DFOB_0 + LiCl_{sat}$ electrolyte. **g** Comparison of the first cycle in the cyclic voltammogram for $FSI_1DFOB_0$ and $FSI_1DFOB_0 + LiCl_{sat}$ electrolyte. A dedicated coin cell design was used for LSV and CA measurements, while a PAT cell design was used for CV measurements throughout this study, as shown in Figs. S13, S14.

energy dispersive X-ray spectroscopy (EDX) spectra confirm the presence of C, O and Fe on the surfaces of both samples. When comparing the elemental distribution of the spacer exposed to $LiCl_{sat}$ with that of a pristine spacer, it is clear that the increased amount of C and O are likely the result of electrolyte decomposition and SUS dissolution during the chronoamperometric step[39,40] (Table S1). Notably, the elemental distributions of Fe, Cr and Ni are not identical with the pristine spacer after treatment with $LiCl_{sat}$, suggesting the formation of decomposition compounds (mostly Fe-related compounds). It is known that $Cl^-$ anions facilitate the oxidation of Fe into $Fe^{2+}$, which then further oxidizes to form iron oxide ($Fe_2O_3$) and precipitates on the surface[24]. This is supported by the considerable presence of oxygen detected on the spacer surface exposed to $LiCl_{sat}$ electrolyte, as shown by EDX analysis (Table S1). This process leads to the small, widespread pitting observed on the surface of the spacer (Fig. 1c). In contrast, the spacer harvested from cells with $FSI_1DFOB_0$ electrolyte shows pitting, but with larger and deeper pits (Fig. 1d). The elemental distribution of the spacer with $FSI_1DFOB_0$ remains similar to the

pristine one, with the only difference being the presence of S and F elements, arising from LiFSI salt decomposition. This suggests that the $Fe^{2+}/Fe^{3+}$ ions do not predominantly form $Fe_2O_3$ but react further with $FSI^-$ anions or with their decomposition products. These reactions form soluble complexes, preventing the formation of $Fe_2O_3$, thus facilitating continued metal dissolution, which extends the local pitting. The dissolved Fe ions diffuse through the electrolytes and deposit on the surface of the negative electrode, changing the morphology of the Li metal surface as indicated by the SEM-EDX measurements of Li metal counter electrodes harvested after CA measurements (Fig. S4b). Cracks and inhomogeneous morphologies are observed on the Li metal surface harvested from cells with $FSI_1DFOB_0$ and $LiCl_{sat}$, which can be attributed to the accumulation of Fe ions, as confirmed by EDX analysis in Table S2.

Given that 7.49 ppm of $Cl^-$ anion impurities were detected in the LiFSI salt, the effects of different $Cl^-$ anion concentrations on the electrolyte behavior were further studied. As shown in Fig. 2a, the electrolytes with low $Cl^-$ anion concentrations (1 ppm, 5 ppm, and 10

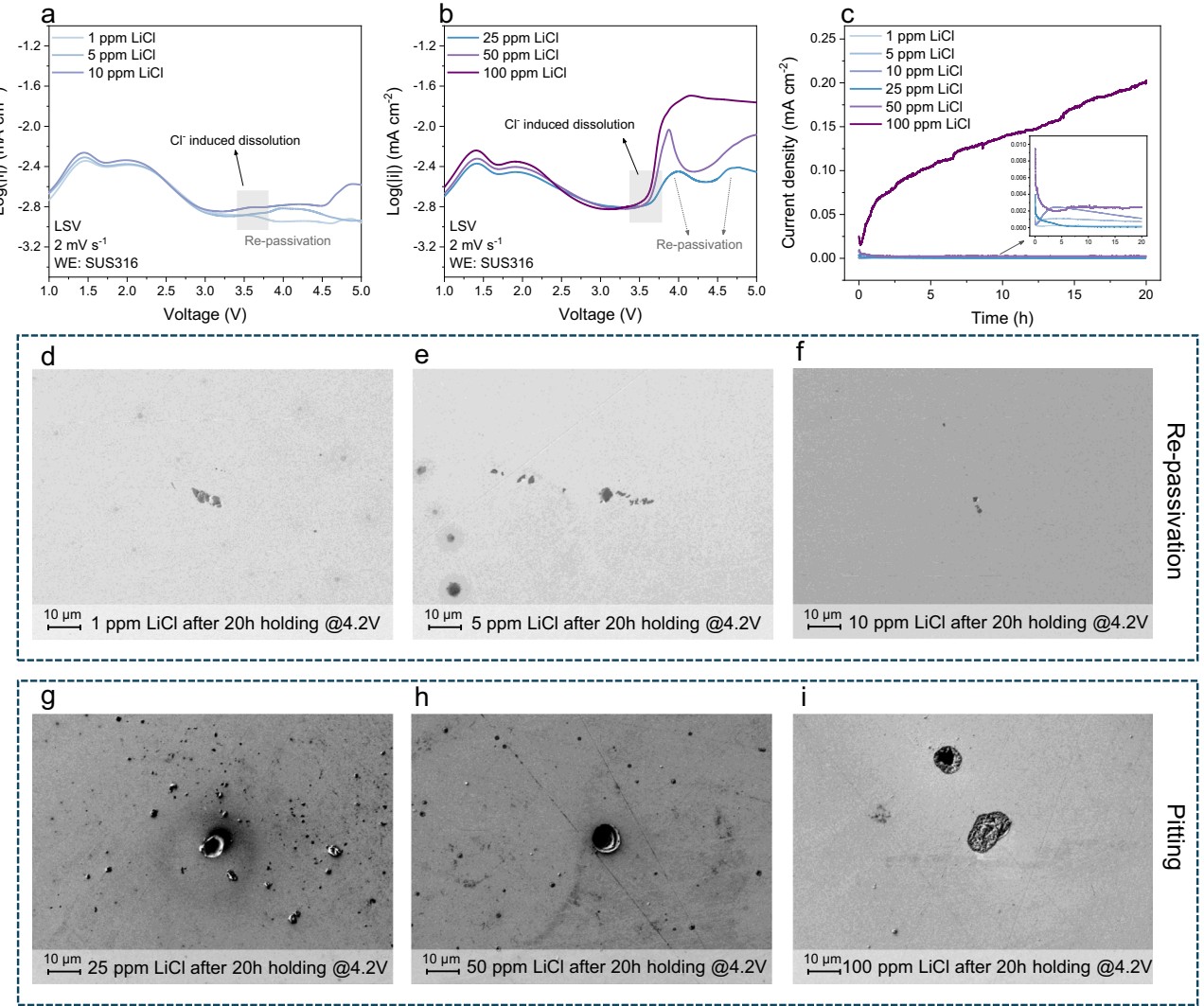

**Fig. 2 | Concentration-dependent impact of Cl⁻ on SUS dissolution. a** Linear sweep voltammetry (LSV) curves of cells containing SUS316 as the working electrode (WE) with electrolytes containing 1 ppm, 5 ppm and 10 ppm LiCl concentration. **b** LSV curves of the cells containing SUS316 as working electrodes with electrolytes containing 25 ppm, 50 ppm and 100 ppm LiCl concentration.

**c** Chronoamperograms of cells containing SUS316 as the working electrode with electrolytes containing different LiCl concentrations. SEM image of SUS316 surfaces harvested from cells with electrolytes containing **d** 1 ppm, **e** 5 ppm, **f** 10 ppm, **g** 25 ppm, **h** 50 ppm, **i** 100 ppm LiCl after 20 h of CA measurements at 4.2 V and 20 °C.

ppm LiCl) did not exhibit a critical voltage threshold for Cl⁻ anion-induced dissolution, showing stable current densities. As the Cl⁻ anion concentration increases to 25 ppm, the current rise at ≈3.5 V becomes prominent, indicating an extensive SUS dissolution process (Fig. 2b). It needs to be mentioned that two re-passivation processes are observed for 25 ppm LiCl; however, these re-passivation processes become less evident at higher Cl⁻ anion concentrations. At 50 ppm LiCl, the second re-passivation process is no longer observed, and at 100 ppm LiCl, re-passivation entirely disappeared, indicating continuous SUS dissolution at elevated Cl⁻ concentrations. CA measurements in Fig. 2c further illustrate that, in contrast to the electrolytes with low Cl⁻ concentration, the current increases after the initial passivation for the 50 ppm LiCl electrolyte, suggesting the absence of re-passivation. Upon further increasing the Cl⁻ concentration to 100 ppm, a substantial increase in the current density is observed, indicating the serious SUS dissolution caused by the high Cl⁻ concentration. SEM imaging and EDX elemental analysis of SUS316 spacers after 20 h of CA measurements revealed changes in morphology and elemental composition. The spacer surface treated with low Cl⁻ concentration solutions was covered with black deposits (Fig. 2d–f), identified as iron oxide compounds (Fig. S5),

while those exposed to high Cl⁻ anion concentration solutions exhibited pitting on the surface (Fig. 2g–i). Overall, Cl⁻ anion impurities are decisive in triggering pitting dissolution in SUS within LiFSI-based electrolytes, as even trace concentrations initiate pit formation, while levels above 50 ppm of Cl⁻ anion concentrations suppress the re-passivation process and enable a sustained pitting process.

**Inhibitive effect of borate salt on SUS dissolution in LiFSI-based electrolyte**
Lithium borate salts have been widely employed as an Al dissolution inhibitor, while its role in SUS dissolution has not been fully addressed yet. In this regard, LiDFOB was added as a co-salt in LiFSI-based electrolytes, as detailed in Table 1. Compared to $FSI_1DFOB_0$, the LSV results shown in Fig. 3a revealed an observable change upon the addition of LiDFOB. It is observed that the critical voltage of SUS dissolution shifted notably towards higher values, along with a decrease in the dissolution current density. Specifically, with the addition of 20% molar ratio of LiDFOB (referred to as $FSI_8DFOB_2$), the critical voltage for anodic dissolution markedly shifted from ≈3.5 V to 4.5 V. Further increasing the LiDFOB concentration to 40% and 50% shifts the critical

voltage to 4.6 V and 4.7 V, respectively. The current decreased steadily as the amount of LiDFOB increased, with the pure LiDFOB (referred to as $FSI_0DFOB_1$) showing the most stable current, with only a slight current increase observed at 5 V. Additional CA measurements confirm

the inhibitive effect of LiDFOB on the SUS dissolution, as evidenced by a pronounced decrease in current density upon addition of LiDFOB (Fig. 3b). SEM images of the SUS316 spacer after CA measurement further demonstrate less or even the absence of pits on the surface in the presence of LiDFOB (Fig. S4a). Moreover, by increasing the amount of LiDFOB, the surface morphology of the Li counter electrode shows substantial changes, as indicated by the SEM image in Fig. S4b. For comparison, the Li electrode recovered from cells with $FSI_1DFOB_0$ electrolyte shows deposits on the surface, mostly comprising C, O and Fe dissolved from the SUS316 spacer (Table S2). Furthermore, the protective role of LiDFOB against $Cl^-$ anions was also studied in the mixed electrolyte comprising $LiCl_{sat}$ and $FSI_0DFOB_1$ (referred to as $FSI_0DFOB_1 + LiCl_{sat}$) (Fig. 3c). Although a slightly higher current density for the primary and secondary passivation is observed, the critical voltage of $Cl^-$ induced dissolution was considerably suppressed and shifted from 3.5 V to 4.2 V in the $FSI_0DFOB_1 + LiCl_{sat}$ electrolyte. A similar trend is also observed in cyclic voltammograms with a slower scan rate. No anodic current is observed for cells with $FSI_5DFOB_5$

**Table 1 | Considered electrolyte formulations and their abbreviations**

| Abbreviation | Electrolyte formulation |
|---|---|
| $FSI_1DFOB_0$ | 1 M LiFSI in EC: EMC 3:7 by wt |
| $FSI_8DFOB_2$ | 0.8 M LiFSI + 0.2 M LiDFOB in EC: EMC 3:7 by wt.% |
| $FSI_6DFOB_4$ | 0.6 M LiFSI + 0.4 M LiDFOB in EC: EMC 3:7 by wt.% |
| $FSI_5DFOB_5$ | 0.5 M LiFSI + 0.5 M LiDFOB in EC: EMC 3:7 by wt.% |
| $FSI_4DFOB_6$ | 0.4 M LiFSI + 0.6 M LiDFOB in EC: EMC 3:7 by wt.% |
| $FSI_2DFOB_8$ | 0.2 M LiFSI + 0.8 M LiDFOB in EC: EMC 3:7 by wt.% |
| $FSI_0DFOB_1$ | 1 M LiDFOB in EC: EMC 3:7 by wt.% |

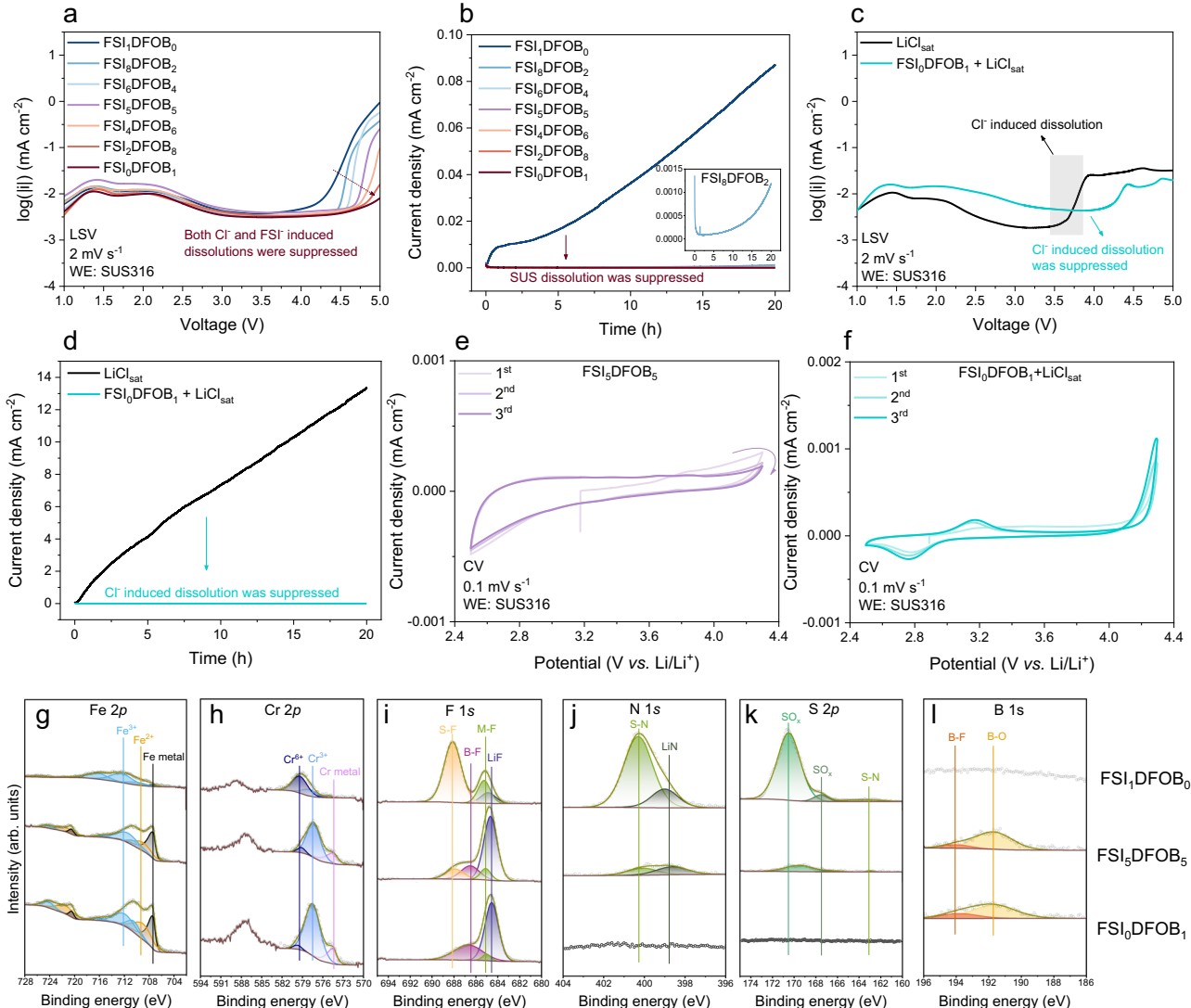

**Fig. 3 | Inhibitive effect of $DFOB^-$ anions on SUS dissolution in LiFSI-based electrolytes. a** Linear sweep voltammetry (LSV) curve of cells containing SUS316 as working electrode (WE) and LiDFOB containing electrolytes, and **b** corresponding chronoamperograms recorded at 4.2 V for 20 h. **c** LSV curves of cells containing SUS316 as working electrode in a mixture of $LiCl_{sat}$ and $FSI_0DFOB_1$ electrolytes and **d** corresponding chronoamperograms recorded at 4.2 V for 20 h. Cyclic voltammetry (CV) curves of cells containing SUS316 as the working electrode with **e** $FSI_5DFOB_5$ and **f** $FSI_0DFOB_1 + LiCl_{sat}$ electrolyte. Selected **g** Fe 2*p*, **h** Cr 2*p*, **i** F 1*s*, **j** N 1*s*, **k** S 2*p* and **l** B 1*s* XPS spectra of the harvested SUS316 spacers from the cells after 20 at 4.2 V and 20 °C with $FSI_1DFOB_0$ (top), $FSI_5DFOB_5$ (middle) and $FSI_0DFOB_1$ (bottom) electrolytes. The remaining C 1*s* and O 1*s* XPS spectra is shown in the Fig. S16.

(Fig. 3e) and only a low anodic current developed at ≈4.2 V for cells with $FSI_0DFOB_1 + LiCl_{sat}$ (Fig. 3f), reaching a maximum current density of 1.0 μA cm⁻². This value is substantially lower than the maximum current density of 80.0 μA cm⁻² observed in the cell containing $FSI_1DFOB_0 + LiCl_{sat}$. In addition, SEM analysis confirms the absence of pits on the SUS surface for cells with $FSI_0DFOB_1 + LiCl_{sat}$ (Fig. S6). The stable current density at 4.2 V and the strongly suppressed current density during LSV and CV measurements, together with the pitting-free surface of the SUS316 spacer after CA measurement, collectively confirm the inhibitive effect of LiDFOB in suppressing both Cl⁻ and FSI⁻ anion-induced SUS dissolution.

The use of $FSI_8DFOB_2$ electrolyte results in a shift of the critical voltage of the anodic dissolution from 3.5 V to 4.5 V, whereas only a minor increase in current is still observable after 10 h in the CA measurement for $FSI_8DFOB_2$ (Fig. 3b). The appearance of small pitting holes on the SUS316 spacer, identified in the SEM images in Figure S4a, implies that the kinetics also play a considerable role in SUS dissolution. The kinetics of the reaction might be influenced by factors such as the consumption rate of inhibiting additives, the diffusion of reactive species to the electrode surface, and the stability of the protective oxide layer[41,42]. Moreover, with further increases in the amount of LiD-FOB, no further increase in anodic current is observed within 20 h at 4.2 V, suggesting an enhanced inhibition of SUS dissolution.

X-ray photoelectron spectroscopy (XPS) analysis conducted on SUS316 spacers harvested from cells containing $FSI_1DFOB_0$, $FSI_5DFOB_5$ and $FSI_0DFOB_1$ electrolytes after a 20 h holding at 4.2 V is shown in Fig. 3g–l. These investigations revealed pronounced differences in both the intensity and shape of spectra between cells using pure LiFSI-based electrolytes ($FSI_1DFOB_0$) and LiDFOB-containing electrolytes ($FSI_5DFOB_5$ and $FSI_0DFOB_1$), which are attributed to the dissolution products deposited on the SUS316 surface in the presence of pure LiFSI-based electrolyte. Analysis of the Fe 2p and Cr 2p spectra reveals insight into the oxidation state of iron and chromium metal. The pristine SUS316 surface shows predominantly the signal of Fe metal at 707.5 eV in the Fe 2p spectra and the signal of Cr metal at 574.4 eV in the Cr 2p spectra, indicative of minor surface oxide from chromium and iron (Fig. S7). However, for samples harvested from cells containing $FSI_1DFOB_0$ electrolyte (Fig. 3g, h), the peaks for pure Fe and Cr are considerably reduced, accompanied by an increase in the $Fe^{3+}$ and $Cr^{6+}$ signals in the Fe 2p and Cr 2p spectra, suggesting a pronounced oxidation of iron and chromium metal. Additionally, signals of S–F, S–N and $SO_x$ in the F 1s, N 1s and S 2p spectra (Fig. 3i–k) reflect the presence of decomposition products from FSI⁻ anions in the $FSI_1DFOB_0$ electrolyte. The pronounced signals of $Fe^{3+}$ and $Cr^{6+}$, combined with these signals from FSI⁻ anion or its decomposition products, suggest the formation of complexes between the $Fe^{3+}$ and $Cr^{6+}$ and FSI⁻ or its decomposition products. In contrast, the SUS316 spacers harvested from cells with $FSI_5DFOB_5$ and $FSI_0DFOB_1$ show no signs of such severe dissolution, aligning with the observations from the SEM measurements. The spectra for these samples maintain stronger signals for metallic Fe and Cr, along with a higher $Fe^{2+}/Fe^{3+}$ and $Cr^{3+}/Cr^{6+}$ ratio, suggesting less oxidation. Additionally, the appearance of LiF, B-F and B-O signals in the F 1s and B 1s spectra for $FSI_5DFOB_5$ and $FSI_0DFOB_1$ samples suggests the decomposition of LiDFOB and the potential formation of a boron-containing layer on the SUS surface (Fig. 3i, l).

Given that DFOB⁻ anion comprises molecular moieties from both lithium tetrafluoroborate ($LiBF_4$) and lithium bis(oxalato)borate (LiBOB), an additional LSV experiment was conducted using electrolytes containing equivalent molar ratios of $LiBF_4$ and LiBOB separately dissolved in an EC: EMC solvent mixture at a ratio of 3:7 by weight. Due to the solubility limitations of LiBOB, only a 20% molar ratio was considered for these experiments. The LSV results in Fig. 4a indicate that while the addition of $LiBF_4$ does not change the critical voltage of SUS dissolution, the oxalate moiety is critical in mitigating

SUS dissolution in environments with Cl⁻ and FSI⁻ anions. Since the oxalate group is easily decomposed, forming a boron-containing film on the SUS316 surface (Fig. 3l), it is hypothesized that this film could shield SUS from the attack of aggressive Cl⁻ and FSI⁻ anions. However, further experiments confirmed that this boron-containing film is not able to prevent the SUS dissolution, as detailed in Supplementary Note 1. This brings the exploration of an alternative mechanism involving competing anion adsorption. In general, the oxalate group favors adsorption on metal surfaces, especially those that are capable of forming stable coordination complexes[43]. Zhao et al. have experimentally and theoretically demonstrated that DFOB⁻ anions preferentially accumulate inside the electric double layer (EDL) on the lithium cobalt oxide positive electrode, particularly in the inner Helmholtz plane, through electrostatic forces in a dual salt system with LiFSI, thereby excluding other species from the EDL[44]. Therefore, we propose that the DFOB⁻ anions preferentially adsorb on the SUS surface, and thus, this excludes "aggressive anions" from initiating the SUS dissolution. To further confirm this, alternating current voltammetry (ACV) was employed to measure the potential zero charge (PZC) using the considered electrolytes. As indicated in Fig. 4b, the PZC for Cl⁻ anion is characterized by a potential of 1.137 V vs. Li/Li⁺, while the PZC for FSI⁻ and DFOB⁻ anions show a potential of 1.111 V and 1.063 V vs. Li/Li⁺, respectively. The shift to a lower potential for the PZC indicates stronger specific adsorption of the anions onto the surface of the working electrode[44]. These values also indicate that the DFOB⁻ anions interact stronger with the SUS surface among the considered anions. In order to confirm these results, first-principles calculations were carried out to obtain the adsorption energy for the different anions on the SUS surface. The adsorption energy calculations (in eV) for LiCl, LiFSI and LiDFOB were obtained by relaxing these molecules on a (0001) $Fe_2O_3$ slab, representing the SUS surface, generated using density functional theory (DFT). The more negative the energy, the stronger the binding of the molecule to the (0001) $Fe_2O_3$. The adsorption energies of LiCl, LiFSI and LiDFOB on (0001) $Fe_2O_3$ are calculated to be −3.38, −3.28 and −3.68 eV, respectively (Fig. 4c). Among them, the adsorption energy of LiDFOB is considerably stronger than that of LiCl and LiFSI to the $Fe_2O_3$ surface, providing a competitive advantage over other anions. The specific adsorption of anions occurs even without an applied voltage, impacting the corrosion behavior of SUS. For instance, Cl⁻ anions can specifically adsorb onto the SUS surface, compromising the passivation film and triggering SUS corrosion. Therefore, storage experiments were performed by immersing SUS316 spacers in aqueous solutions containing different conducting salts to exclude the interference from the competitive adsorption from solvents and complex solvation structures in organic solvents. The SEM images of SUS316 spacers display deposits and pits when immersed in LiCl- and LiCl+LiFSI-based solutions (Fig. 4d, f), while a clean surface without pits was observed on SUS316 spacers immersed in LiCl+LiDFOB-based electrolyte (Fig. 4e). A similar trend was also observed in complementary experiments in organic electrolyte systems shown in Fig. S8.

During the SUS dissolution process, TMs such as Fe ions dissolved from SUS play a critical role in the roll-over failure in LIBs[45–47]. These TMs can accelerate electrolyte decomposition, damage the SEI and induce lithium dendrite growth. Gas evolution as a critical consequence of electrolyte decomposition is particularly important to analyze in this process to assess the inhibition effect of LiDFOB. The analysis of gas evolution during the dissolution process was performed using 3-electrode differential electrochemical mass spectrometry (DEMS) cells while carrying out an LSV measurement with a scan rate of 0.1 mV s⁻¹. As shown in Fig. 5a, the LSV results using DEMS cells exhibit similar behavior regarding SUS dissolution compared to the LSV results from coin cells, with minor deviations attributed to different cell configurations.

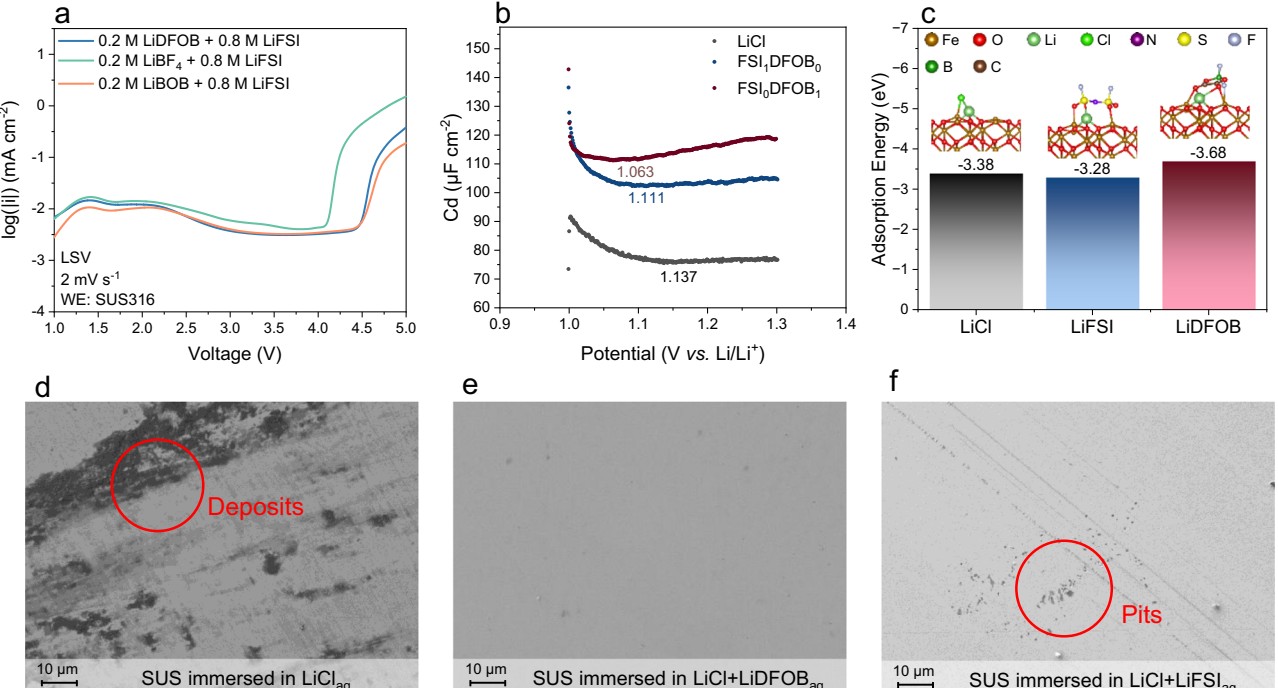

**Fig. 4 | Underlying mechanism of the inhibitive effect of DFOB⁻ anions on SUS dissolution. a** Linear sweep voltammetry (LSV) curves of cells containing SUS316 as the working electrode (WE) in boron-containing electrolytes. **b** Differential capacitance *vs.* potential curves for different electrolytes. The minimum value of the differential capacity curve corresponds to the zero-charge potential. **c** Calculated adsorption energies of LiCl, LiFSI and LiDFOB on (0001) $Fe_2O_3$. SEM image of SUS316 surface after 1 month immersion at 25 °C in **d** $LiCl_{aq}$ (100ppm LiCl in $H_2O$), **e** LiCl + $LiDFOB_{aq}$ (100ppm LiCl + 0.5 M LiDFOB in $H_2O$) and **f** LiCl + $LiFSI_{aq}$ (100ppm LiCl + 0.5 M LiFSI in $H_2O$) solutions. All samples in d-f were rinsed with deionized water and dried before imaging.

Four regions of gas evolution during the SUS dissolution process could be identified: Region (I) starts from 0 V *vs.* Li/Li⁺ and continues until the second self-corrosion potential[24], and it is in this region that a rapid release of $H_2$ is observed for all electrolytes after applying 0 V *vs.* Li/Li⁺. This is likely due to electrolyte or water reduction at the surface of the negative electrode[48]. In addition, a continuous release of $CO_2$ is observed for the cells with $FSI_1DFOB_0$ electrolytes in this region. Region (II) includes the primary and secondary passivation processes, where no gas evolution takes place. Region (III) is defined by the potential range where the Cl⁻ anion-induced dissolution appears, while a slight $H_2$ release, as a consequence of SUS dissolution at increased potential, is observed. Such potential-dependent $H_2$ evolution remains disputed and unclear, possibly originating from the electrochemical reaction of electrolyte or SEI decomposition driven by the dissolved TMs[48–51]. It is noted that only a small amount of $H_2$ is released in this stage, considering the re-passivation process, as indicated by the decrease in current after an initial increase in LSV. Region (IV) is defined by a pronounced $H_2$ and an early $CO_2$ release, particularly with the cells containing LiCl and $FSI_1DFOB_0$. The release of $CO_2$ in this region is attributed to electrolyte decomposition. DEMS cells containing LiCl release $CO_2$ earlier than others, possibly due to the lower electrochemical stability of free solvents, as evidenced by Raman analysis (Fig. S9). CO and $O_2$ release is also observed in DEMS cells with $FSI_1DFOB_0$ electrolytes, indicating electrolyte decomposition triggered by the SUS dissolution. In the presence of LiDFOB, as is the case with DEMS cells with $FSI_0DFOB_1$ and $FSI_5DFOB_5$, $H_2$, CO and $O_2$ gas evolution is substantially suppressed. These results further confirm that the presence of LiDFOB can suppress the SUS dissolution and thus also mitigate electrolyte decomposition.

## Proposed mechanism of SUS dissolution and inhibitive effect of LiDFOB

Comprehensive experimental and theoretical evaluation enabled discussion regarding the possible mechanism for SUS dissolution and LiDFOB inhibition. Electrochemical experiments with fluorinated solvents have confirmed that the protons released during solvent oxidation do not contribute to the SUS dissolution (Supplementary Note 3). Instead, the observed dissolution is predominantly associated with the presence of Cl⁻ and FSI⁻ anions. To clarify the inhibitive effect of LiDFOB, $LiPF_6$ was used as a representative source for fluoride anions, and it confirms that fluoride does not impact the SUS dissolution. Furthermore, although boron-containing surface species were detected, their presence does not provide protection for SUS against dissolution, and the competing anion adsorption by DFOB⁻ anions is the most plausible explanation for the suppression effect. A detailed discussion of the proposed mechanism is provided below.

The complex process of SUS dissolution is schematically outlined in Fig. 6 considering various scenarios. In the first scenario, when only Cl⁻ anions are present, the dissolution mechanism outlined in Fig. 6a implies: (1) The Cl⁻ anions accumulate on the SUS surface and attack the oxide film and expose the underlying SUS. (2) The exposed SUS begins to pit and is oxidized to $Fe^{2+}$ under the applied potential. (3) With growing potential, $Fe^{2+}$ is oxidized to $Fe_2O_3$, which precipitates on the surface and re-passivates the pits. The predominant $Fe_2O_3$ coverage on the SUS surface suppresses further dissolution, leading to reduced anodic current even at higher potentials. Furthermore, dissolved Fe ions migrate to and deposit on the surface of the negative electrode, where they damage the SEI and promote crack formation in the Li metal. In addition, the observed $CO_2$ and $H_2$ gases forming at high potential provide clear evidence of electrolyte and SEI decomposition.

The second scenario, schematically illustrated in Fig. 6b depicts the involvement of LiFSI in the SUS dissolution process, which occurs due to the presence of Cl⁻ anion impurities in the LiFSI-containing electrolyte: (1) Accumulation of both Cl⁻ and FSI⁻ anions on the surface and Cl⁻ anions attack on the oxide film and expose the underlying SUS.

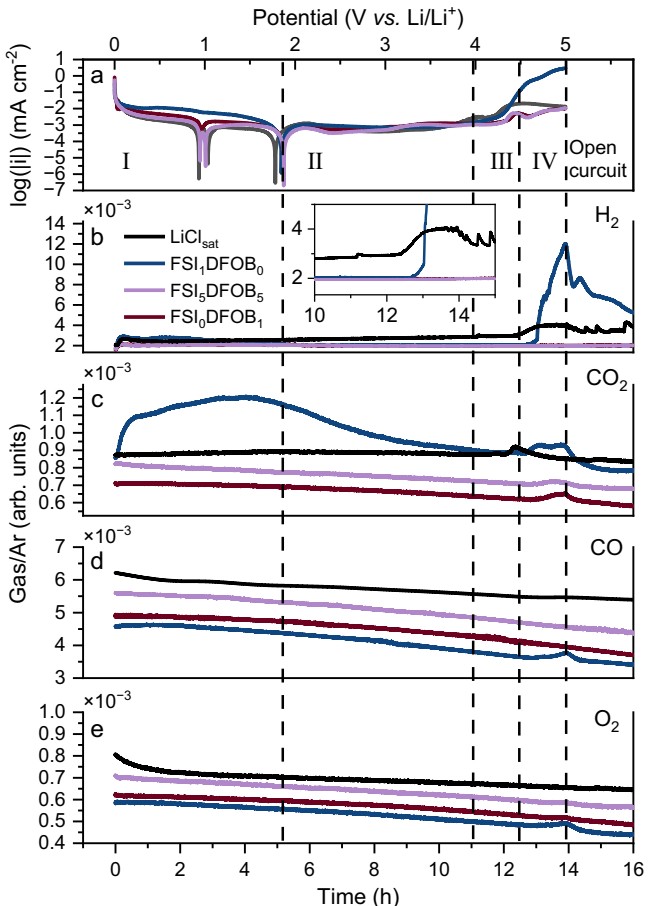

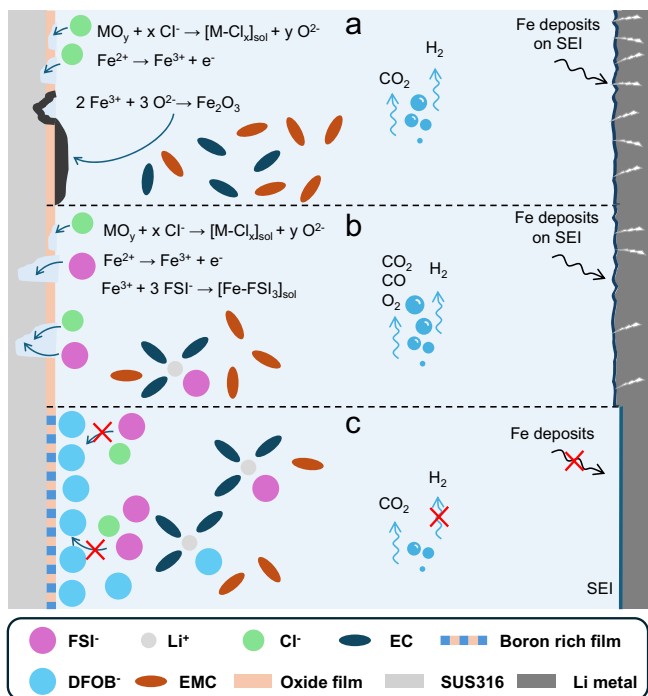

**Fig. 6 | Illustration of proposed mechanism scenarios. a** SUS dissolution process in the presence of $Cl^-$. **b** SUS dissolution process in the presence of LiFSI-based electrolyte. **c** Inhibiting effect of LiDFOB on the SUS dissolution in the LiFSI-based electrolyte. The exact structures of the reactant molecules and complexes involved in these reactions are not known. Abbreviations: EC ethylene carbonate; EMC ethylene methyl carbonate; SEI solid electrolyte interphase.

**Fig. 5 | Operando gas analysis of SUS dissolution in different electrolytes. a** LSV curves of a 3-electrode DEMS cells using SUS316 as working electrode (WE) in different electrolytes, recorded at a scan rate of $0.1\,mV\,s^{-1}$ from 0 to 5 V $vs.$ Li/Li⁺ at 25 °C. *Operando* gas evolution profiles of **b** $H_2$, **c** $CO_2$, **d** CO and **e** $O_2$ by DEMS are monitored during the LSV; the gases are normalized with Ar to eliminate the fluctuation of carry gas. Four regions of gas evolution could be identified: region (I) starts from 0 V $vs.$ Li/Li⁺ and continues until the second self-corrosion potential. Region (II) includes the primary and secondary passivation processes. Region (III) is defined by the potential range where the $Cl^-$ anion-induced dissolution appears. Region (IV) is defined by a substantial $H_2$ and $CO_2$ release.

(2) The exposed SUS begins to pit and oxidize to $Fe^{2+}$ under the applied potential. (3) $Fe^{2+}$ is further oxidized to $Fe^{3+}$ and forms soluble complexes with $FSI^-$ anions or its decomposition products. (4) $Fe^{3+}$ can further oxidize the Fe to $Fe^{2+}$ and accelerate the SUS dissolution. With this continued SUS dissolution, large pits are formed, and almost no re-passivation can be observed. Concurrently, Fe deposits accumulate on the Li metal, resulting in crack formation and an inhomogeneous surface. Furthermore, electrolyte decomposition induced by the SUS dissolution process results in gas evolution, including $H_2$, $CO_2$, CO and $O_2$, which would affect the safety properties of the LIBs.

In the third scenario, with the addition of LiDFOB, SUS dissolution in the presence of LiFSI is suppressed. A competing anion adsorption mechanism is proposed, as illustrated in Fig. 6c. LiDFOB preferentially adsorbs and accumulates on the SUS surface, thus excluding the $Cl^-$ and $FSI^-$ anions from the surface, preventing SUS dissolution at high potentials. Consequently, the addition of LiDFOB suppresses $H_2$, CO and $O_2$ evolution by mitigating TMs dissolution, underscoring its critical role in stabilizing LiFSI-based electrolytes. It should be emphasized that LiDFOB continues to decompose due to electrochemical instability, as evidenced by the 1000 h CA measurements and the

detected boron-containing film on the SUS surface (Fig. S17). However, these decomposition products from LiDFOB may also contribute to SEI or cathode electrolyte interphase (CEI) formation on both electrodes, effectively suppressing interfacial degradation and electrolyte depletion, therefore improving the electrochemical performance of the LIBs[46,52–54].

## Electrochemical performance with optimized electrolyte design

Consistent with the role of LiDFOB in suppressing the SUS dissolution, the electrochemical performance was further evaluated in a laboratory-scale Gr‖NMC811 coin cells employing SUS316 components. As depicted in Fig. 7a, cells utilizing the $FSI_1DFOB_0$ electrolyte exhibit a pronounced rise in anodic current during the constant voltage step, which originates from SUS or Al dissolution[23,27]. Further *post-mortem* analysis discussed in the Supplementary Note 4 identifies the SUS dissolution as the responsible process for the anodic current increase and leads to failure in the initial cycle. In contrast, cells using LiDFOB-containing electrolyte ($FSI_5DFOB_5$) demonstrate stable charge and discharge behavior. This can be further confirmed by the decent galvanostatic cycling stability over the first 150 cycles (Fig. S10a). However, a large deviation of specific discharge capacity along with a rapid drop in Coulombic efficiency (CE) in the subsequent cycles indicates ongoing dissolution processes. Although Al dissolution is effectively mitigated by LiDFOB-containing electrolytes, as demonstrated in Supplementary Note 2, *post-mortem* analysis in Supplementary Note 4 confirms the persistence of SUS dissolution during prolonged cycling. Further enhancement of electrochemical stability is achieved by employing SUS316L components, which exhibit higher dissolution resistance than SUS316, as evidenced by the higher critical voltage observed in LSV for cells employing SUS316L (Fig. S11). As a result, the Gr‖NMC811 cells with SUS316L parts using $FSI_5DFOB_5$ electrolyte show effective CE for more than 1200 cycles, resulting in a

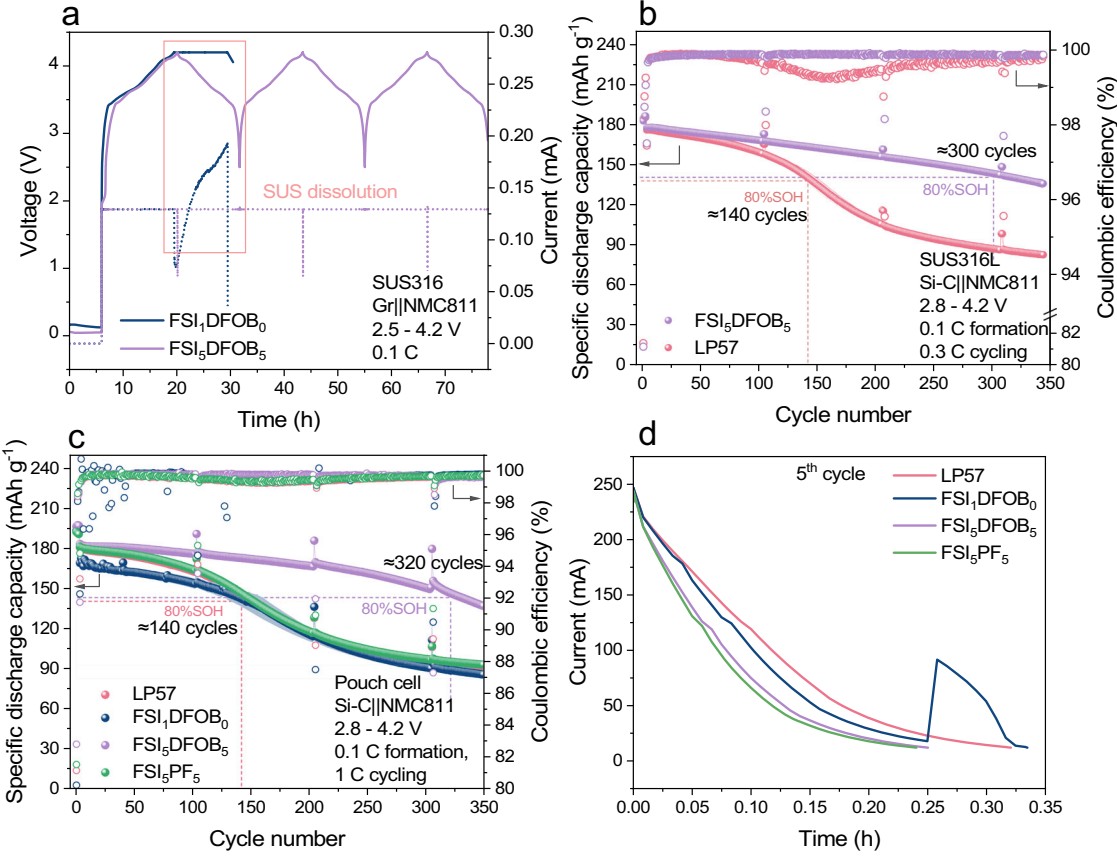

**Fig. 7 | Validation of LiFSI-based electrolytes in Si-C‖NMC811 coin and pouch cells. a** Voltage *vs.* time profile of coin cells with SUS316 parts using $FSI_1DFOB_0$ and $FSI_5DFOB_5$. **b** Specific discharge capacity *vs.* cycle number curves of Si-C‖NMC811 coin cells containing SUS316L parts with $FSI_5DFOB_5$ and LP57. **c** Specific discharge capacity *vs.* cycle number curves of Si-C‖NMC811 pouch cells with $FSI_1DFOB_0$, $FSI_5DFOB_5$, $FSI_5PF_5$ and LP57. **d** Current *vs.* time profile during the constant voltage step at the 5th cycle for cells with the considered electrolytes. 1C = 200 mA g⁻¹ for NMC811 positive electrode. Note: Replacing the coin cell components from SUS316 to SUS316L does not affect the galvanostatic cycling performance for both Gr‖NMC811 and Si-C‖NMC811 cells using LP57 electrolytes, as detailed in the Supplementary Note 6. A detailed coin cell and pouch cell configuration is shown in Fig. S12. Abbreviation: SOH: state of health.

substantially improved galvanostatic cycling performance of ≈1150 cycles to 80% state-of-health ($SOH_{80\%}$), representing nearly six-fold increase in cycle life compared to cells with LP57 electrolyte (Fig. S10b).

To assess the compatibility of the optimized electrolyte with more demanding negative electrodes, Si-C negative electrodes containing 20% silicon were also considered. Si-C‖NMC811 cells with $FSI_5DFOB_5$ maintain an effective CE with nearly double the lifespan compared to cells with LP57, reaching ≈300 cycles (Fig. 7b). This enhanced galvanostatic cycling performance in both Gr‖NMC811 and Si-C‖NMC811 cells originates not only from the suppressed Al and SUS dissolution, but also from the complementary role of the two conducting salts. LiFSI offers enhanced physicochemical properties (increased thermal and chemical stability compared to $LiPF_6$, ion mobility, etc.)[39], and has been reported for stabilizing the negative electrode by forming LiF-rich SEI[17,54]. In addition, LiDFOB is beneficial for effective SEI/CEI formation and has been reported to form boron-containing interphase species, which are associated with more effective SEI and CEI layers[52,54,55].

To eliminate any influence from SUS components and to evaluate the effect of electrolytes with different salts, Si-C‖NMC811 pouch cells were evaluated. Consistent with the results observed in coin cells, cells with $FSI_5DFOB_5$ displayed a stable CE and reached ≈320 cycles at $SOH_{80\%}$, whereas the cells with LP57 reached ≈140 cycles at $SOH_{80\%}$ (Fig. 7c). The consistent trend observed in both coin and pouch cells suggests that the benefits of the optimized electrolyte are not restricted to laboratory-scale coin cells and can extend to an industry-relevant format. Importantly, pouch cells do not contain SUS components, and the performance differences observed between $FSI_5DFOB_5$ and LP57 primarily reflect interphase stability. To further demonstrate the necessity of combining LiDFOB and LiFSI, $FSI_1DFOB_0$ and $FSI_5PF_5$ (0.5 M LiFSI + 0.5 M $LiPF_6$) were also evaluated. A pronounced fluctuation in CE and specific discharge capacity is observed for cells with $FSI_1DFOB_0$ during initial cycles. The increased current during the constant voltage step in the 5th cycle confirms the presence of Al dissolution, explaining this fluctuation (Fig. 7d). Interestingly, although the addition of $LiPF_6$ suppresses Al dissolution and results in a stable CE, the overall galvanostatic cycling performance remains similar to the cells with LP57. This indicates that the LiFSI alone, after excluding the impact of Al and SUS dissolution, is insufficient to ensure long-term cycling stability in such demanding cell chemistry. The effective film-forming capability of LiDFOB is required to suppress ongoing electrolyte decomposition and stabilize both electrodes. This conclusion is further supported by the improved performance observed in Li‖NMC811 and Li‖Gr cell setups with $FSI_5DFOB_5$ electrolytes, as detailed in Supplementary Note 5. The results demonstrate that while $FSI_5DFOB_5$ electrolyte considerably enhances cell stability by suppressing Al and SUS dissolution, its intrinsic physicochemical advantages and effective SEI/CEI formation capability also contribute to long-term cycling. Moreover, substituting SUS316 with SUS316L further boosts performance, underscoring the critical role of material selection coupled with electrolyte formulation.

By effectively mitigating both SUS and Al dissolution through the strategic use of LiDFOB and LiFSI-based electrolytes, this study

considerably enhanced electrochemical performance across different cell chemistries and cell formats. The optimized electrolytes are well-suited for practical high-energy LIBs configurations, including emerging large-format designs such as Tesla 46XY cylindrical cells, which utilize stainless steel components and commonly employ NMC811 and graphite electrodes[29,30]. The insights gained here directly contribute to industrial-scale battery advancements, leading to improved stability and performance for LIBs.

In summary, $Cl^-$ and $FSI^-$ anions depicted as "aggressive anions" in the LiFSI-based electrolyte can destabilize the SUS surface and trigger severe dissolution of SUS components, representing a critical limitation for LIB cells. Herein, studies using controlled addition of $Cl^-$ anions show that $Cl^-$ anions initiate the pitting process in a concentration-dependent manner. A mechanism was proposed in which $Cl^-$ anions break the oxide film and initiate the pits, whereas $FSI^-$ anions form soluble complexes with $Fe^{2+}/Fe^{3+}$ that accelerate SUS dissolution. A mitigation strategy has been designed in which LiDFOB is introduced to suppress SUS dissolution through competing anion adsorption. The accumulation of $DFOB^-$ anions on the SUS surface excludes $Cl^-$ and $FSI^-$ anions from the surface and substantially suppresses dissolution. Furthermore, gas evolution is also suppressed in the presence of LiDFOB. Importantly, the inhibition of SUS dissolution prevents further Al dissolution when SUS components are used, resulting in a complete metal dissolution-free environment. The critical role of metal (SUS and Al) stability for the electrochemical performance is demonstrated by the galvanostatic cycling measurements of the Gr||NMC811 coin cells with $FSI_5DFOB_5$ electrolyte. Further improvements are achieved by incorporating more dissolution-resistant SUS316L cell components, enabling ≈300 cycles for Si-C||NMC811 cell chemistry at $SOH_{80\%}$. Electrolytes containing LiDFOB and LiFSI retain their performance advantages in Si-C||NMC811 pouch cells, delivering ≈320 cycles at $SOH_{80\%}$. This strategy not only extends cell lifetime but also emphasizes the importance of the electrochemical stability of the metal in the non-aqueous electrolytes for the design of practical high-energy LIBs.

## Methods

### Materials

Battery grade ethylene carbonate (EC), ethyl methyl carbonate (EMC), fluoroethylene carbonate (FEC), lithium bis(fluorosulfonyl)imide (LiFSI), lithium difluoro(oxalato)borate (LiDFOB), lithium tetrafluoroborate (LiBF$_4$), lithium bis(oxalato)borate (LiBOB) and lithium hexafluoro-phosphate (LiPF$_6$) were purchased from E-Lyte Innovations GmbH with a purity of >99% and were used as received. Methyl (2,2,2-trifluoroethyl) carbonate (FEMC) was purchased from E-Novation Chemicals LLC with a purity of >99% and was purified by drying over CaH$_2$ followed by distillation. Lithium chloride (LiCl) was purchased from Sigma-Aldrich with a purity of >99%. Tris(trimethylsilyl) phosphate (TMSPa) was purchased from Tokyo Chemical Industry Co., Ltd with a purity of >98%. To formulate the electrolyte, the specific amount of lithium salt or the salt mixture was dissolved in a solvent mixture by weight inside an Ar-filled glove box, as detailed in Table 1. A lithium chloride (LiCl) saturated electrolyte was stirred for 2 days at 25 °C in a glovebox and filtered with a syringe filter (PTFE 20 µm) to remove the undissolved LiCl before use.

### Electrodes and cell assembly

For the assembly of graphite (Gr)||LiNi$_{0.8}$Co$_{0.1}$Mn$_{0.1}$O$_2$ (NMC811) coin cells, calendared NMC811 single-sided coated positive electrode sheets (with an areal capacity of 1.0 mAh·cm$^{-2}$ and mass loading of 5.26 mg cm$^{-2}$) were provided by Umicore. The NMC811 positive electrode composes 95 wt.% active materials, 3 wt.% carbon black and 2 wt.% binder with a thickness of 39 µm (on 25 µm thick Al foil). Calendared graphite single-sided coated electrode sheets (with an areal capacity of 1.14 mAh·cm$^{-2}$ and mass loading of 3.47 mg cm$^{-2}$) were provided by CIDETEC, corresponding to an N/P ratio of 1.14. The graphite negative electrode is composed of 94 wt.% active materials, 2 wt.% C45, 2 wt.% carboxymethyl cellulose (CMC) and 2 wt.% styrene-butadiene rubber (SBR) with a thickness of 33 µm (on 20 µm thick Cu foil).

For the assembly of silicon-graphite (Si-C)||NMC811 coin cells, both electrodes were purchased from Li-FUN technology. The single-sided coated NMC811 positive electrode exhibited a higher areal capacity of 2.73 mAh·cm$^{-2}$ (14.96 mg cm$^{-2}$ mass loading, 96.4 wt.% active material, 59 µm thick on 15 µm thick Al foil), while the Si-C negative electrode featured with an areal capacity of 3.5 mAh·cm$^{-2}$ (7.72 mg cm$^{-2}$ mass loading, 94 wt.% active material, 69 µm thick on 12 µm thick Cu foil), resulting in an N/P ratio of 1.28.

For coin cell assembly, the positive and negative electrodes were punched into ø14 mm and ø15 mm disks, respectively. Before use, both electrodes were dried under vacuum at 120 °C (<10$^{-2}$ mbar) for 12 h and then kept in the argon-filled glovebox until cell assembly. Celgard 2500 membrane (25 µm thick) was punched into ø16 mm and used as a separator.

For Li metal cells (Li||NMC811 and Li||Gr), 150 µm thick Li metal foils (China Energy Lithium Co., Ltd) were used as negative electrodes, while the NMC811 provided by Umicore and graphite provided by CIDETEC were used as positive electrodes, respectively.

The CR2032 coin cells (SUS316 and SUS316L grade, purchased from Xiamen TOB New Energy Technology Co. Ltd and Seika Sangyo GmbH) were assembled by sandwiching the electrodes and separator between a 0.5 mm and a 1.0 mm thick stainless steel (SUS) spacer (ø15.8 mm), supported by a wave spring (ø14.5 mm × 1.2 mm). For Gr||NMC811 cells, 30 µL of electrolyte was used, while 45 µL was used for Si-C||NMC811 cells. The illustration of the coin cell configuration can be found in Fig. S12. The cells were assembled in an argon-filled glovebox (MBraun, <0.1 ppm O$_2$, <0.1 ppm H$_2$O) and sealed using an automated electric crimper (Hohsen Corporation).

Commercially available Si-C||NMC811 double-side coated multi-layer pouch cells (nominal capacity of 235 mAh with the same active materials from Li-FUN technology) were used to evaluate the electrolytes at the pouch cell level. The cells were cut open and dried at 90 °C for 12 h under vacuum (<10$^{-2}$ mbar) before use. Then, the cells were filled with 700 µL of electrolytes and sealed at the upper end of the gas pocket (165 °C, 5 s) using a pouch cell sealer (GN-HS350V, Gelon LIB Co) in the argon-filled glovebox. A specially developed holder was used for the cells with a constant pressure of ≈2 bar, as shown in Fig. S12. After three formation cycles and at the discharged state, the cells were cut from the top, opened and sealed under the conditions described previously.

### Karl Fischer titration

For solvents and prepared electrolytes, the water content was determined using a C30S coulometric Karl Fischer titrator (Mettler-Toledo), equipped with a diaphragm-type platinum generator electrode. HydranalTM-Coulomat CG and AG-Oven (Honeywell Fluka) were used as reagents. All samples were prepared in an argon-filled glovebox (MBraun, <0.1 ppm O$_2$, <0.1 ppm H$_2$O, 25 °C ± 2 °C) and transferred using cap-sealed syringes to the titrator for direct injection.

For lithium salts, the C30S coulometric titrator was coupled with an InMotion Karl-Fischer Pro Oven (Mettler-Toledo). Samples were weighed and enclosed in glass vials in an argon-filled glovebox (MBraun, <0.1 ppm O$_2$, <0.1 ppm H$_2$O, 25 °C ± 2 °C). The sealed vials were transferred to the oven, where they were automatically pierced and heated to 160 °C. The generated gases were transferred into the titrator by dry nitrogen carrier gas (150 mL min$^{-1}$). HydranalTM-Coulomat CG and AG-Oven (Honeywell Fluka) were used as reagents, HydranalTM water standard Karl Fischer Oven (5%) as standard.

### Metal dissolution study

A special coin cell format was used to study the metal dissolution, with Al foil or polished SUS316 spacer as the working electrode and Li foil (500 µm thick, China Energy Lithium Co., Ltd) as the counter

electrode. Linear sweep voltammetry (LSV) measurements were conducted from 0 to 5 V at a scan rate of 2 mV s⁻¹, whereas the potentiostatic chronoamperometry (CA), followed by LSV to 4.2 V and holding the voltage for 20 h. The detailed configurations of coin cells used for LSV and CA are illustrated in Fig. S13.

Cyclic Voltammetry (CV) measurements were conducted using a PAT cell (EL-CELL) in a three-electrode fashion, with an Al foil or a polished SUS316 spacer as the working electrode, a Li ring as the reference electrode and a Li foil (500 µm thick) as the counter electrode. A lower plunger with different metal materials (Al or SUS) was used for the Al and SUS dissolution study, and the detailed configuration is shown in Fig. S14. CV measurements were conducted from 2.5 to 4.3 V *vs.* Li/Li⁺ at a scan rate of 0.1 mV s⁻¹.

Alternating current voltammetry (ACV) was conducted using a PAT cell (EL-CELL) in a three-electrode fashion, with a polished SUS316 spacer as the working electrode, Li ring as the reference electrode and Li foil (500 µm thick) as the counter electrode. The ACV measurements were conducted in a potential range of 1.0 to 3.0 V *vs.* Li/Li⁺ using a staircase potentiostatic signal with 2000 potential steps (dE = 1 mV). At each step, a 5-s equilibration time was applied before measurement to ensure a quasi-stationary state. The excitation signal was a single sine wave with a frequency of 10 Hz and an amplitude of 10 mV. Data density of 6 points per decade was employed. Prior to the scan, the cells were rested at open-circuit potential for 2 h. The capacitance is calculated by the following equation (Eq. (1))[44]:

$$C = \frac{1}{\omega Z_{Im}} \qquad (1)$$

Where $C$ is the capacitance, $\omega = 2\pi f$, $Z_{Im}$ is the imaginary part of the impedance.

All measurements were conducted using a VMP3 workstation (BioLogic) equipped with a climate chamber (BINDER) at 20 °C. Al foils (21 µm thick, Nippon) were punched into ø12 mm disks and kept inside the argon-filled glovebox before use. For clarity and readability, the electrochemical results (LSV, CV, CA and ACV) presented in the figures are obtained from representative cells. These representative curves were selected based on their close alignment with the median performance observed across at 2 replicates.

### SUS corrosion study

The specific conducting salts were dissolved into deionized H₂O and EC/EMC mixtures to formulate the aqueous and organic solutions, respectively (detailed compositions in Table 2). The aqueous solutions were prepared under a fume hood (25 °C ± 2 °C) while organic solutions were prepared in an argon-filled glovebox (MBraun, <0.1 ppm O₂, <0.1 ppm H₂O, 25 °C ± 2 °C).

For corrosion study, SUS316 spacers were immersed in the formulated solutions within cap-sealed glass vials and stored for one month in a fume hood and a glovebox for aqueous solution and organic solution, respectively. After the storage period, the spacers were thoroughly washed with EMC (for samples in organic solutions) or deionized water (for samples in aqueous solutions) to remove residual salts and solvents. The samples were then dried in an oven (50 °C) and subsequently examined using SEM to assess the surface morphology.

### Electrochemical performance evaluation

Coin and pouch cells were cycled using a MACCOR 4000 battery testing unit (Maccor Inc, USA) in a temperature-controlled chamber (BINDER) at 20 °C. Prior to cycling, all cells were kept at open circuit voltage for 6 h. Galvanostatic cycling was performed using a constant current-constant voltage (CC−CV) charging protocol and a constant current (CC) discharging protocol.

Specifically, Gr||NMC811 coin cells were galvanostatically cycled between 2.5 and 4.2 V at 1 C following three formation cycles at 0.1 C

**Table 2 | Considered aqueous solutions for the corrosion study**

| Abbreviation | Electrolyte formulation |
|---|---|
| LiCl_aq | 100 ppm in H₂O |
| LiCl + LiDFOB_aq | 100 ppm LiCl + 0.5 M LiDFOB in H₂O |
| LiCl + LiFSI_aq | 100 ppm LiCl + 0.5 M LiFSI in H₂O |
| LiCl_org | 100 ppm in EC/EMC 3:7 by wt% |
| LiCl + LiDFOB_org | 100 ppm LiCl + 0.5 M LiDFOB in EC/EMC 3:7 by wt% |
| LiCl + LiFSI_org | 100 ppm LiCl + 0.5 M LiFSI in EC/EMC 3:7 by wt% |

(1 C = 200 mA g⁻¹ for NMC811 positive electrode). Si-C||NMC811 coin cells were galvanostatically cycled between 2.8 and 4.2 V at 1/3 C after three formation cycles at 0.1 C, with a recovery step at 0.1 C every 100 cycles. Pouch cells of the same cell chemistry were cycled at 1 °C. For the Li||NMC811 coin cells, the cells were galvanostatically cycled in two voltage ranges (2.5 – 4.2 V and 2.5 – 4.3 V) at 1 C after three formation cycles at 0.1 C. Li||Gr cells were galvanostatically cycled from 0.01 to 1.5 V at 1 C after 3 formation cycles at 0.1 C. The capacity retention and State of Health (SOH) were calculated based on the specific discharge capacity of the first cycle after formation cycles (defined as the 4th total cycle). To ensure reproducibility, each galvanostatic cycling performance evxaluation was performed using at least three replicate cells. The data points and error bars presented in the plots represent the mean values and standard deviations, respectively.

### 3D printing device and polishing device

To achieve the mirror quality of the SUS surface, the SUS316 and SUS316L spacers (1 mm thickness) were polished using a polishing machine, Tegramin-25 (Struers). To ensure the precise mounting of the 1 mm spacers in the polishing machine holder, special in-house insets were prepared by 3D-printing, employing the masked direct light projection technique on an Elegoo Mars 2 Pro printer with Phrozen Aqua 4 k resin. The desired mirror quality was accomplished by polishing the spacer using three diamond polishing suspensions with a grain size of 9 µm, 6 µm and 1 µm (DP-Suspension P, Struers) on polishing cloths (MD-Sat, MD-Dac and MD-Nap, Struers). After each polishing step, the spacers were thoroughly cleaned in an ultrasonic bath for 5 min using deionized water to remove the residual materials. The effectiveness of polishing is discussed in Supplementary Note 7.

### Scanning electron microscopy with energy dispersive X-ray spectroscopy

SUS Spacers were analyzed via scanning electron microscopy (SEM) after CA measurements. After electrochemical measurements, the cells were disassembled in an argon-filled glovebox (MBraun, <0.1 ppm O₂, <0.1 ppm H₂O, 25 °C ± 2 °C). The spacers were washed with 1 mL EMC to remove the residual electrolyte, transferred to the SEM using a vacuum-sealed sample holder, and examined on an Auriga CrossBeam workstation (Zeiss) at 3 kV acceleration voltage with a 5 mm working distance using the SE2 detector. Energy dispersive X-ray spectroscopy (EDX) was performed at an acceleration voltage of 3 kV with an Ultim Extrem detector (Oxford Instruments), and the elemental composition was evaluated with AZtech software (Oxford Instruments).

### X-Ray photoelectron spectroscopy

The X-ray photoelectron spectroscopy (XPS) measurements were conducted using a K-Alpha photoelectron spectrometer (Thermo VG Scientific) equipped with a monochromatic Al K_α source ($E_{photon}$ = 1486.6 eV). The cells under consideration were opened in an argon-filled glovebox (MBraun, <0.1 ppm O₂, <0.1 ppm H₂O, 25 °C ± 2 °C) and the spacers were rinsed with EMC (1 mL) to remove the residual electrolyte. Spacers were then transferred to the XPS chamber using a vacuum-sealed sample holder. The pressure within

the analysis chamber was maintained below $2 \cdot 10^{-9}$ mbar. The device is regularly calibrated for energy levels according to ISO 15472 using copper, silver and gold reference samples. The intensity scale is calibrated according to the device manufacturer's specifications. Sputter depth profiling of the pristine SUS spacer was conducted using monoatomic argon at an energy of 500 eV. The depth profiling procedure consisted of three sputter cycles of 15 s each. The CasaXPS software (Casa Software, U.K.) was used for peak fitting, with literature-guided peak assignment[24,26,40,56,57]. The energy scale in the spectra was adjusted using the C 1 s peak at 284.8 eV (C–C) as an internal reference.

### Fourier Transform Raman spectroscopy

The Fourier Transform Raman (FT-Raman) measurements of the liquid electrolytes were performed in 5 mm NMR-Tubes (Bruker) on a VERTEX 70 FT-IR spectrometer (Bruker) with a RAM II FT-Raman Module (Bruker). The device was equipped with a nitrogen-cooled Ge-diode detector and a 1064 nm laser source. The spectra were obtained using the OPUS 7.0 software. For the sample preparation, 400 μL electrolytes were filled in cap-sealed NMR tubes (ø5 mm, Wilmad) in an argon-filled glovebox (MBraun, <0.1 ppm $O_2$, <0.1 ppm $H_2O$, 25 °C ± 2 °C) and then the NMR tubes were transferred for FT-Raman measurements.

### Differential electrochemical mass spectrometry

The differential electrochemical mass spectrometry (DEMS) was carried out by coupling a potentiostat VSP (Bio-Logic) with a mass spectrometer MS GSD 350 (Pfeiffer Vacuum) consisting of a cross-beam ionization chamber (QMG 250 Prisma Pro) attached to a high-vacuum HiCube 80 Eco turbomolecular pump using a stainless steel heated capillary with a length of 1 m and an inner diameter of 0.125 mm. Further parameters for the MS are listed in Table 3.

DEMS Measurements were conducted in a three-electrode ECC-DEMS cell (EL-Cell) with a gas inlet and outlet connected via the capillary to the MS. The ECC-DEMS cell is assembled with a polished SUS316 spacer as the working electrode, Li ring as the reference electrode, and Li foil (500 μm thick) as the counter electrode in an argon-filled glovebox (MBraun, <0.1 ppm $O_2$, <0.1 ppm $H_2O$, 25 °C ± 2 °C). The measurements were conducted without a climate chamber at 25 °C ± 2 °C. An LSV measurement was conducted for the assembled ECC-DEMS cells from 0 to 5 V vs. Li/Li+ at a scan rate of 0.1 mV s−1. Prior to each DEMS measurement, the Ar gas was flowed through the ECC-DEMS cells for 2 h to stabilize the gas environment and no detectable evolution of gases was observed during the open circuit potential period (Fig. S15).

For clarity and readability, the LSV results and gas signals presented in the figures are obtained from representative cells. These representative curves were selected based on their close alignment with the median performance observed across two replicates.

### Theoretical calculations

Plane wave (PW) based density functional theory (DFT) calculations[58,59] were performed using the Vienna Ab initio Simulation Package (VASP)[60,61]. The projector augmented wave (PAW) method[62] was employed to represent the core states of atoms. The generalized gradient approximation (GGA) with Perdew–Burke–Ernzerhof (PBE) parametrization[63] was utilized in the calculations. A Hubbard correction (DFT + U) of 5 eV was applied to account for the electronic correlation of the Fe 3 $d$ orbitals[64]. The plane wave cutoff energy was set to 520 eV, with the convergence criterion for the iterative solution of the Kohn-Sham equations set at $10^{-5}$ eV, and the residual forces on all atoms were constrained to 0.01 eV Å−1. The DFT-D2 method of Grimme was used to correct for dispersion interactions[65,66]. Spin polarization was considered with an antiferromagnetic spin arrangement of Fe in the direction perpendicular to the surface. The Brillouin zone was sampled using an $11 \times 11 \times 7$ Monkhorst-Pack k-point mesh[67] for the $Fe_2O_3$ bulk unit cell,

**Table 3 | Measurement parameters used in the MS**

| Settings | Value |
|---|---|
| Multiple Ion Detection (MID) (m/z) | 2, 16, 18, 26, 28, 32, 35, 37, 40, 44, 70, 72, 74 |
| Energy of the electron beam (eV) | 70 |
| Detector | Faraday/SEM |
| Scan rate (ms) | 128 |
| Ar flow rate (mL min−1) | 1 |

while a $2 \times 2 \times 1$ mesh was used for the $Fe_2O_3$ surface. The optimized lattice parameters for the R-3c phase of the $Fe_2O_3$ unit cell used in this study were found to be $a = b = 5.13$Å and $c = 13.72$Å. To create the slab model, the (0001) surface[67] was considered with a $2 \times 3$ supercell along the $a$ and $b$ directions to provide sufficient surface area for molecule adsorption. A vacuum of 30 Å was added on top of the $Fe_2O_3$ slab to eliminate interactions between periodic images. The surface relaxation was performed while keeping the bottom half of the atoms fixed to preserve their bulk positions. This surface was then subjected to molecular dynamics (MD) simulations to adsorb the molecules and predict their positions and conformations on the surface. Subsequently, DFT calculations were conducted on the optimized structures to determine the adsorption energy ($E_{ad}$) using the equation below (Eq. (2)):

$$E_{ad} = E_{sys} - E_{sur} - E_{mol} \qquad (2)$$

where $E_{sys}$ represents the total energy of the complete structure, while $E_{sur}$ and $E_{mol}$ represent the total energies of the pristine $Fe_2O_3$ surface and the isolated molecule, respectively.

All MD simulations have been performed with the GROMACS-2019.6 package[68]. The molecular interactions of the salt ion pairs (LiFSI, LiDFOB, LiCl) were described by the CL&P force field[69–71], whereas the parameters for the $Fe_2O_3$ slab were taken from Latorre et al.[72], using the geometric mixing rule for all cross-interactions. The individual ion pairs were inserted above the optimized $Fe_2O_3$ slabs from the initial DFT calculations. The systems were equilibrated by a 200 ns run in the NVT ensemble at a temperature of 500 K, maintained by the velocity-rescale thermostat[73] with a coupling time of 1 ps. Subsequently, the systems were gradually annealed from 500 K to 0 K over a period of 100 ns. Lennard-Jones interactions were truncated at 0.5 nm due to the small size of the systems. Electrostatic interactions were computed by the particle-mesh Ewald method[74] using a cut-off radius of 0.5 nm and a grid spacing of 0.08 nm with sixth-order interpolation. Periodic boundary conditions were applied in all three dimensions. The final structures were taken as input for the DFT calculations of the adsorption energies.

## Data availability

The DFT and MD data generated in this study have been deposited in the Zenodo database under accession code[75] [https://doi.org/10.5281/zenodo.16813415]. The source data containing the raw data for all figures in the main manuscript and the Supplementary Information are provided in a single Excel file within the Zenodo repository[75].

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

## Acknowledgments

The authors gratefully acknowledge Nick Fehlings for performing the IC measurements, and Bahareh A. Sadeghi and Susanne Schierz for supporting the laboratory work. The authors acknowledge funding from the European Union's Horizon 2020 research and innovation program under Grant Agreement Nos. 957189 (BIG-MAP), 957213 (BATTERY2030PLUS) and 101147342 (SAFELOOP).

### Author contributions

P.Y. conceived the idea together with M.C.S. and I.C-L. K.Z. and D.D. performed DFT calculations and MD simulations. P.Y. and M.C.S. performed LSV, CV and CA measurements. R.G. performed the 3D printing. P.Y. and M.C.S. performed SEM and EDX measurements. P.Y. prepared the samples for XPS measurements and conducted data analysis. M.C.S performed FT-Raman measurements. P.Y. performed DEMS measurements. P.Y. performed the galvanostatic cycling measurements. P.Y. performed the Karl Fischer titration measurements. P.Y. wrote the manuscript with contributions from M.C.S., C.W., M.W., and I.C-L. All authors contributed to interpreting the findings, reviewing, and revising the manuscript.

## Funding

## Competing interests

The authors declare no competing interests.
