## [Transparent Peer Review file · Nature Communications]

Mechanism and Mitigation of Stainless Steel Dissolution in LiFSI based Lithium Ion Battery Electrolytes

Corresponding Author: Dr Isidora Cekic-Laskovic

Version 0:

Reviewer comments:

Reviewer #1

(Remarks to the Author)

In the present work, the authors examine the dissolution behavior of a stainless steel in LiFSI-based electrolytes, focusing on the effects of different anions (impurities). As even a tiny amount of water in organic solvent strongly affects the corrosion of stainless steel, it is impossible to judge whether the proposed dissolution mechanism can be acceptable at the present state because the authors do not provide the moisture level of the LiFSI-based electrolytes used. Therefore, the reviewer asks the authors to quantify the moisture level and then reconsider the mechanism if some water is included in the electrolytes.

Reviewer #2

(Remarks to the Author)

The manuscript investigates the dissolution mechanism of SUS (stainless steel) in LiFSI-based electrolytes and suggests the use of a LiDFOB additive to enhance stability by passivating the SUS interface. Although the dissolution issues of SUS in such electrolyte systems have been extensively studied, including the Cl⁻ induced dissolution documented in *Electrochimica Acta* (2022, 419, 140353) and *ChemElectroChem* (2025, 12, e202400632), the novelty of this manuscript's approach to the dissolution mechanism appears limited. Additionally, the role of LiDFOB in SUS passivation is already well-documented.

Given these considerations, I am inclined to suggest that the manuscript may not meet the criteria for publication in *Nature Communications*. Further emphasis on unique findings or advanced methodologies that significantly build upon existing knowledge in this field would be necessary to enhance its contribution to the literature.

Reviewer #3

(Remarks to the Author)

Isidora Cekic-Laskovic et al. investigated the dissolution mechanisms of SUS316 in lithium-ion batteries and proposed an electrolyte-optimization strategy to enhance the anti-corrosion performance of SUS316 and Al foil. This work is meaningful for coin cells. Upon the work, the results find that Cl⁻ as impurity promotes SUS316 to generate pit corrosion (50 ppm as the fringe conditions), and FSI- corrosion effect on SUS spacer behaves more severe pit corrosion.

However, the corrosion of Al foil is the most devastating for the electrode corrosion of coin cells in three dimensions.

Because it triggers the binding-force failure between the cathode materials and the binder. Some deficiencies should be addressed:

1. Why were the aqueous solutions used in Line 308 to evaluate the static (corrosive) effects of three ions on SUS316? Is it to prove that the solvent does not have an impact on the competitive adsorption and passivation of DFOB-? Since the working environment of SUS316 in the battery is the ester-based organic solvents.

2. There is no clear link between Gas evolution and SUS corrosion. The authors may need to combine the correlation of solvent decomposition and SUS dissolution and deposition on the Li anode. Gas evolution has nothing to do with the corrosion mechanism demonstrated in Line 350 "Proposed SUS dissolution mechanism", although it is not conducive to maintaining the stable cycle life of the batteries.

3. The authors may need to add the Operando gaseous evolution of the coin-cell at open circuit potential. The result can be used as the baseline to elucidate the initial state of the polarization process of different electrolytes.

4. Line 406, the authors think that the pronounced rise in anodic current stemmed from SUS dissolution; however, according to Figure S13a, the anodic current may also be introduced by Al corrosion. It can be clarified by SEM morphologies of the SUS316 case and Al foil in batteries after 0.1C formation.
5. Similarly, a rapid drop in Coulombic efficiency (CE) may also be introduced by Al dissolution. It is suggested that the authors disassemble the cycled batteries and analyze the corrosion conditions of SUS and Al foil.
6. CO curve of FSI0DFOB1 in Figure 5d does not correspond to that of Figure S2c. Please check them.
7. The collapse of a full battery involves multiple issues such as degradation of the cathode materials, corrosion of current collectors, and degradation of the anode materials. Therefore, it is more appropriate to evaluate the anti-corrosion performance of DFOB- using a lithium metal anode. The compatibility between anode materials and FSI5DFOB5 needs to be further evaluated in Graphite||Li half cells. In this way, Figure 7b/d will be more persuasive.
8. Some Figure details need attention: Figure 5a, Figure 4a, Figure S9.
9. Please present the distinction of anodic dissolution in FSI0.5DFOB0.5 between the cell of the positive case of SUS316L and the cell of Al-CVD on SUS316L. (Figure S13 FSI5DFOB5 _Al and FSI5DFOB5 _Al_316).

Version 1:

Reviewer comments:

Reviewer #1

(Remarks to the Author)

The reviewer is satisfied with the response from the authors and recommends the manuscript for publication.

Reviewer #3

(Remarks to the Author)

All my concerns have been addressed. This paper can be published in its current form.

Reviewer #4

(Remarks to the Author)

In this manuscript, the authors' report approaches to mitigate Stainless Steel corrosion in LiFSI/LiDFBO-based electrolytes for lithium-ion battery applications. The study is interesting and includes many detailed characterizations and experiment design. However, I agree with Reviewer #2 and Reviewer #3 that the dissolution issues of stainless steel in LiFSI have been heavily investigated, and the major issue is to address the corrosion of the positive electrode current collectors (aluminum foils), considering the practical application of commercial and industry relevant cells. Accordingly, the novelty of the work is quite limited.

Furthermore, the authors assigned corrosion issues merely owing to the presence of Cl⁻ anion impurities and ignored the main reasons of the solvent decomposition beyond 4.2 V vs. Li⁺/Li which release protons and initiate the corrosion of the current collectors (Ma, T., et al. (2017), *J. of Physical Chemistry Letters*, 8(5), 1072–1077; Nyholm, L., et al. (2023). *Chemical Engineering Science*, 282, 119346). More seriously, for the full cell results in LP57, the interpretations of the results are misleading. The authors are advised to address the following comments before new submissions:

1. As a general comment, the authors need to clearly refer to which cell design was used for each experiment, this among the blurry things in the manuscript. The Figs. on the different cell design should be placed in the main manuscript rather than the SI.
2. What is the level of impurities in the LiFSI, LiDFOB and LiPF6 salts? Previous reports (e.g. Nyholm, L., et al. (2023). *Chemical Engineering Science*, 282, 119346) showed that the presence of high amount of fluoride anion impurity, which can play a crucial role in passivation. This can be also one of the reasons for the observed passivation in LiDFOB solutions.
3. The passivation mechanism should be revised according after considering the decomposition of the carbonate solvent and the presence of other impurities in the salt.
4. The corrosion behavior for both current collectors and stainless steel should be examined using cyclic voltammetry as well. This will clearly illustrate the possible passivation behavior.
5. The corrosion behavior of the current collector and full cell should be performed in pouch cells and then compared to the results in stainless steel, so the authors can differentiate between the current collector corrosion and the stainless steel one.
6. The authors showed that the stability of the NMC811-graphite cells is quite poor; however, the LiPF6 should offer stable electrochemical performance "As a result, the NMC811||graphite cells with SUS316L parts using FSI5DFOB5 electrolyte show effective CE for more than 1200 cycles, as compared to an ineffective CE from 200 cycles for those with LP57 (1M LiPF6 in EC:EMC, 3:7) electrolyte (Figure 7c)." There should be no corrosion issues or serious degradation of the NMC811 in the potential window 2.5-5.0 V and LiPF6 salt in carbonate electrolytes. Accordingly, the results should be carefully revised, and the interpretation should be corrected.
7. Finally, the authors are encouraged to explain and judge all possible explanations of the corrosion phenomenon and cell failure rather than steering the interpretation to a specific mechanism and ignoring other possibilities.

Version 2:

Reviewer comments:

Reviewer #4

(Remarks to the Author)

The authors have revised the manuscript and addressed most of the raised comments. However, the manuscript still needs further polishing to be more concise and to offer a clear message to the readers.

Furthermore, each Figure, including supplementary, needs to have a title.

The performance of the full cells should be further explained, considering other factors besides the corrosion issues, such as CEI layer formation and composition with different salts.

Does the performance of full cells of NMC//graphite in pouch format offer the same long-term stability with different salts (e.g., Fig. 7c, 1150 cycles for 80% SoH)?

Version 3:

Reviewer comments:

Reviewer #4

(Remarks to the Author)

The authors have addressed most of the raised comments; however, if the authors can not validate the NMC811 || graphite in pouch cells, the results in coin cells should also be omitted to avoid misleading findings "While NMC811 || graphite pouch-cell results are not currently available, ".

Dear Editor,

Please find enclosed the revised **Manuscript No. NCOMMS-25-20665** entitled:

“Mitigating Stainless Steel Dissolution in LiFSI-based Electrolytes: Insights into Dissolution Mechanisms and Enhancing Cycle Life in Lithium-ion Batteries “,

by authors: *Peng Yan, Marian Cristian Stan, Kazem Zhou, Diddo Diddens, Christian Wölke, Rayan Guerdelli Martin Winter, and Isidora Cekic-Laskovic*

First, we would like to express our sincere gratitude for the careful attention given to our manuscript and the opportunity to address the reviewers` concerns through an appeal containing an extensively revised manuscript that clearly demonstrates the unique advancements this work provides beyond current understanding in the field. We thank the Reviewers for the attentive analysis, as well as the positive and critical remarks and valuable suggestions to improve the quality of our work. In line with their guidance, we have incorporated all recommended revisions in the updated manuscript and have highlighted the changes in **yellow**.

Point-by-point responses to the reviewers` comments are presented on the following pages.

Reviewer #1:

In the present work, the authors examine the dissolution behavior of a stainless steel in LiFSI-based electrolytes, focusing on the effects of different anions (impurities). As even a tiny amount of water in organic solvent strongly affects the corrosion of stainless steel, it is impossible to judge whether the proposed dissolution mechanism can be acceptable at the present state because the authors do not provide the moisture level of the LiFSI-based electrolytes used. Therefore, the reviewer asks the authors to quantify the moisture level and then reconsider the mechanism if some water is included in the electrolytes.

First of all, we appreciate the reviewer’s constructive feedback. The reviewer has raised an insightful comment regarding the water impurity that the electrolytes and their components might contain and that would play a role on the observed behavior and could finally affect the proposed reaction mechanism. As the reviewer pointed out, it is acknowledged that electrolyte’s impurities can potentially induce effects of an, as yet, unknown nature. To address the effect of water content, Karl Fischer titration experiments were conducted on all LiFSI-based electrolyte components utilized in this study. In addition, we performed additional Karl Fischer titration analysis on the LP57, FSI₁DFOB₀ and FSI₅DFOB₅ electrolytes to ensure that no excess moisture was introduced during the electrolyte preparation process. The measured moisture values summarized in Table S 5 indicate that all lithium salts have a very low moisture level and are under the detection limits of the Karl-Fischer titration device. The solvent mixture of EC/EMC also shows a low moisture level of 16.52 ppm. In addition, the formulated electrolytes

generally maintain low moisture levels, except for FSI₅DFOB₅, which shows a slightly higher value of 39.32 ppm.

The following text has been included in the revised manuscript:

“Karl Fischer titration

For solvents and formulated electrolytes, Karl Fischer measurements were performed on a C30S coulometric titrator (Mettler-Toledo), which has a platinum generator electrode with a diaphragm. HydranalTM-Coulomat CG and AG-Oven (Honeywell Fluka) were used as reagents.

For lithium salts, Karl Fischer measurements were performed on a C30S coulometric titrator (Mettler-Toledo), which has a platinum generator electrode with a diaphragm and is combined with an InMotion Karl-Fischer Pro Oven (Mettler-Toledo). Samples were enclosed to glass vials, automatically pierced inside the oven, heated to 160 °C, and the gases generated were transferred into the titrator with dry nitrogen as carrier gas (150 mL min⁻¹). HydranalTM-Coulomat CG and AG-Oven (Honeywell Fluka) were used as reagents, HydranalTM water standard Karl Fischer Oven (5%) as standard.”

And the following text and figures have been included in the Supplementary Information:

“Investigation of the Impact of Trace Amount Water on SUS Dissolution

Karl Fischer titration experiments were conducted for electrolyte components to determine the moisture level of the LiFSI-based electrolyte used in this study. The measured values summarized in Table S 5 indicated that all lithium salts have a very low moisture level and are under the detection limits of the Karl-Fischer titration device. The EC/EMC mixture also showed a quite low moisture level of 16.52 ppm. To ensure that no excess moisture was introduced during the electrolyte preparation process, additional Karl Fischer titration analysis was performed on the formulated electrolytes LP57, FSI₁DFOB₀ and FSI₅DFOB₅. It can be observed that the formulated electrolytes generally maintain low moisture levels, except for FSI₅DFOB₅, which shows a slightly higher value of 39.32 ppm.

Table S 5. Values from Karl Fischer titration experiments for electrolyte components.”

Components	Moisture level (ppm)
LiFSI	under the detection limits (<10 ppm)
LiPF ₆	under the detection limits (<10 ppm)
LiDFOB	under the detection limits (<10 ppm)
EC/EMC mixture	16.65
FSI ₁ DFOB ₀	15.12
FSI ₅ DFOB ₅	39.32
LP57	12.35

Additional to the direct method of measuring the content of the electrolytes and their individual components, we have also carried out an experiment where 5 vol.% of tris(trimethylsilyl)phosphate (TMSPa) was added on top to the FSI₁DFOB₀ and FSI₅DFOB₅ electrolytes. TMSPa is known to react with residual water and form trimethylsilanol (TMSOH) and H₃PO₄^[1], which can be used as water scavenger additives. Gogoi et al. have reported the disappearance of the NMR signal associated with residual water after the addition of TMSPa additives into electrolytes^[2]. The results of the Karl Fischer titration analysis are summarized in Table S 6 confirm a reduction in moisture level in both electrolytes after the addition of the TMSPa additive.

Furthermore, linear sweep voltammetry (LSV) and chronoamperometry (CA) measurements of these electrolytes containing TMSPa were carried out to evaluate to confirm the impact of trace moisture on SUS dissolution. As shown in Figure S 24, there is no clear effect observed for TMSPa to the electrochemical behavior of the electrolyte on SUS electrode compared to electrolytes without TMSPa. The results indicate similar current response between the electrolyte with and without TMSPa, therefore, we can conclude that the residual moisture content in our electrolytes is sufficiently low and does not impact the SUS dissolution behavior under the conditions investigated in this study.

The following text and figures have been included in the Supplementary Information:

“To investigate the potential impact of trace amounts of moisture on SUS dissolution behavior, tris(trimethylsilyl)phosphate (TMSPa) was used as moisture scavenger to further reduce the moisture level in selected electrolytes. TMSPa is known to react with residual water and form trimethylsilanol (TMSOH) and H_3PO_4 ^[13]. 5 vol % of TMSPa was added on top of electrolytes and their moisture level was examined further by Karl Fischer titration analysis^[13]. The values summarized in Table S 6 confirm a reduction in moisture level in both electrolytes after the addition of the TMSPa additive.

Table S 6. Values from Karl Fischer titration experiments for electrolytes with TMSPa additives.

Components	Moisture level (ppm)
FSI ₁ DFOB ₀ + TMSPa	10.26
FSI ₅ DFOB ₅ + TMSPa	12.34

Further linear sweep voltammetry (LSV) and chronoamperometry (CA) measurements were conducted for the electrolytes containing TMSPa to evaluate the potential impact of trace moisture on SUS dissolution. As shown in Figure S 24, cells with TMSPa-containing electrolytes demonstrated similar electrochemical behavior in both LSV and CA measurements when compared to those without TMSPa. This indicates that the residual moisture content in the considered electrolytes is sufficiently low and does not impact the SUS dissolution behavior under the conditions investigated in this study.

Figure S 24. (a) Linear sweep voltammograms of cells containing SUS316 as working electrode and TMSPa containing electrolytes and (b) corresponding chronoamperograms recorded at 4.2 V for 20 h.”

References:

- [1] N. Gogoi, W. Wahyudi, J. Mindemark, G. Hernández, P. Broqvist, E. J. Berg, *J. Phys. Chem. C* **2024**, *128*, 1654–1662.
- [2] N. Gogoi, E. Bowall, R. Lundström, N. Mozhzhukhina, G. Hernández, P. Broqvist, E. J. Berg, *Chem. Mater.* **2022**, *34*, 3831–3838.

Reviewer #2:

The manuscript investigates the dissolution mechanism of SUS (stainless steel) in LiFSI-based electrolytes and suggests the use of a LiDFOB additive to enhance stability by passivating the SUS interface. Although the dissolution issues of SUS in such electrolyte systems have been extensively studied, including the Cl⁻ induced dissolution documented in *Electrochimica Acta* (2022, 419, 140353)

and *ChemElectroChem* (2025, 12, e202400632), the novelty of this manuscript's approach to the dissolution mechanism appears limited.

We appreciate the reviewer's insightful comment on this matter. There have been many studies that discuss the dissolution behaviour of SUS in a more or less fundamental approach. The mentioned reports by the reviewer (*Electrochimica Acta* (2022, 419, 140353) and *ChemElectroChem* (2025, 12, e202400632)) address SUS dissolution in LiFSI-based electrolytes, and their findings and conclusions are further summarized below. We also emphasize in these discussions the novelty of our work to better clarify the points raised by the reviewer.

The work published in *Electrochimica Acta* (2022, 419, 140353) mentioned by the reviewer investigates the dissolution behavior of Al and SUS dissolution in LiFSI-based electrolytes and proposed a Cl^- and FSI (FSO_3^-) anion induced dissolution mechanism^[3]. However, in their study the authors did not provide a detailed discussion on the role of Cl^- impurities, such as the critical concentration needed to initiate SUS dissolution. In addition, their proposed strategies by using high concentration electrolytes (HCE) or localized high concentration electrolytes (LHCE) are insufficient to completely prevent SUS dissolution. The use of HCE or LHCE can reduce the amount of organic solvents and free FSI⁻ anions, thus lowering the amount of aggressive FSI⁻ anions and decreasing the solubility of metal ions/ corrosion products. This approach is similar to the strategies used for inhibiting the Al dissolution in LiFSI/LiTFSI-based electrolytes. However, SUS dissolution is more severe compared to Al dissolution and remains observable under the use of HCE and LHCE, as evidenced by the color changes in electrolytes after CV cycling experiments with SUS in both HCE and LHCE. Furthermore, recent studies by Wang et al. and Stan et al. have independently confirmed that the LiFSI-based electrolyte still dissolves SUS even at high concentration^[4,5].

Regarding the *ChemElectroChem* (2025, 12, e202400632) paper mentioned by the reviewer, this previous work from our group investigated the influence of LiFSI concentration and SUS grade (SUS316 vs. SUS316L) on SUS dissolution^[5]. It was demonstrated that the use of SUS316L and HCE indeed mitigates SUS dissolution, but these approaches are still insufficient to completely prevent dissolution. The proposed strategies were to use a blended salt of LiFSI and LiPF_6 together with Al-CVD@SUS316L cases, which can enable good galvanostatic cycling stability in LMR|| μ -Si cells. However, this method is not entirely effective, with evidence suggesting the simultaneous occurrence of Al and SUS dissolution in Al-clad SUS coin cell parts^[6].

Additionally, the role of LiDFOB in SUS passivation is already well-documented.

The role of LiDFOB in preventing Al current collector dissolution through the formation of a passivation layer composed of Al-F, Al_2O_3 , and B-O species has been well documented^[7-9]. In addition, LiDFOB has also been widely employed to stabilize the high-voltage electrode by forming an effective CEI and suppressing the transition metal dissolution^[10-15]. Among these studies, many combined LiDFOB with LiFSI and attributed the superior electrochemical performance to the inhibition of Al dissolution and the formation of effective SEI/CEI. However, none of these works have addressed the role of LiDFOB in mitigating the SUS dissolution. For instance, Chen *et al.* reported that electrolyte comprising 0.8M LiFSI and 0.2M LiDFOB enabled good cycling stability in LiFePO_4 ||Gr cell chemistry at elevated temperature (70 °C)^[16]. They observed reduced iron dissolution from LiFePO_4 active materials upon the addition of LiDFOB. However, this iron dissolution is intrinsic to the electrode material and fundamentally differs from the SUS corrosion behavior addressed in our study. Furthermore, no mechanistic analysis was provided in their work. The only relevant work was reported by Li et al^[17]. They examined the electrochemical stability of electrolytes composed of 0.6M imide salts (LiTFSI, LiFSI) and 0.4M orthoborate salts (LiBOB, LiDFOB) in SUS and Al-clad cell cases. Although their results showed that LiFSI-LiBOB led to severe SUS corrosion/dissolution while other combinations did not, their investigation primarily focused on performance ranking rather than elucidating the mechanisms.

Given these considerations, I am inclined to suggest that the manuscript may not meet the criteria for publication in Nature Communications. Further emphasis on unique findings or advanced methodologies that significantly build upon existing knowledge in this field would be necessary to enhance its contribution to the literature.

In our present study, novel insights into the role played by the Cl⁻ anion impurities from organic solvents with respect to SUS dissolution are provided by experimental results and highlighted schematically in the proposed reaction mechanism. Furthermore, the addition of LiDFOB in the electrolyte formulation was shown to improve the electrochemical performance and in the same time inhibit the SUS dissolution induced by FSI⁻ and Cl⁻ anions. The reaction mechanism proposed indicates the selective adsorption of the DFOB⁻ anions on the surface of the SUS material, thus preventing the SUS dissolution initiated by the Cl⁻ anions and evolved by the FSI⁻ anions. Our study has also an impact on the usage of LiFSI-based electrolyte in practical high-energy LIBs, particularly those that make use of cylindrical cell formats with SUS casing, as it is the case of 46XY formats that are increasingly used within the automotive industry.

To emphasize the novelty of our work and better clarify the points raised by the reviewer, we have modified the introduction part in the main text:

“In this work, complementary electrochemical measurements and surface characterization techniques were employed to study the mechanism of SUS dissolution in LiFSI-based electrolytes, with a particular emphasis on the critical role played by Cl⁻ and FSI⁻ anions. Our findings demonstrate that both Cl⁻ and FSI⁻ anions can destabilize the SUS surface, resulting in severe dissolution. In addition, our study addresses SUS dissolution by proposing a mitigation strategy based on the introduction of lithium difluoro(oxalato)borate (LiDFOB). Specifically, we demonstrate for the first time that LiDFOB effectively inhibits SUS dissolution by preferential adsorption of DFOB⁻ anions on the SUS surface, thus preventing the attack from Cl⁻ and FSI⁻ anions. This proposed mechanism is validated by detailed electrochemical and *post mortem* analyses, and the effectiveness of LiDFOB is further confirmed through considerably improved galvanostatic cycling performance in NMC811||graphite and NMC811||Si-Gr cell chemistries, showing the practical relevance of our findings for high energy LIBs employing SUS components.”

And the following text have been included in the revised manuscript:

“Lithium borate salt has been widely employed as Al dissolution inhibitor, while its role in SUS dissolution has not been fully addressed. Li *et al.* reported that the combination of LiFSI and LiBOB induced severe SUS dissolution, whereas LiDFOB-containing electrolytes did not exhibit noticeable SUS dissolution. However, the study did not provide any further details on the mechanism involved^[38].”

References:

- [3] C. Luo, Y. Li, W. Sun, P. Xiao, S. Liu, D. Wang, C. Zheng, *Electrochim. Acta* **2022**, *419*, 140353.
- [4] L. Wang, Z. Luo, H. Xu, N. Piao, Z. Chen, G. Tian, X. He, *RSC Adv.* **2019**, *9*, 41837.
- [5] M. C. Stan, P. Yan, G. M. Overhoff, N. Fehlings, H. Kim, R. T. Hinz, T. T. K. Ingber, R. Guerdelli, C. Wölke, M. Winter, G. Brunklau, I. Cekic-Laskovic, *ChemElectroChem* **2025**, e202400632.
- [6] S. S. Zhang, *J. Electrochem. Soc.* **2023**, *170*, 110527.
- [7] K. Park, S. Yu, C. Lee, H. Lee, *J. Power Sources* **2015**, *296*, 197–203.
- [8] X. Liu, C. Shen, N. Gao, Q. Hou, F. Song, X. Tian, Y. He, J. Huang, Z. Fang, K. Xie, *Electrochim. Acta* **2018**, *289*, 422–427.
- [9] X. Shanguan, G. Jia, F. Li, Q. Wang, B. Bai, *J. Electrochem. Soc.* **2016**, *163*, A2797.
- [10] J. Cha, J.-G. Han, J. Hwang, J. Cho, N.-S. Choi, *J. Power Sources* **2017**, *357*, 97–106.
- [11] Y. Yang, X. Wang, J. Zhu, L. Tan, N. Li, Y. Chen, L. Wang, Z. Liu, X. Yao, X. Wang, X. Ji, Y. Zhu, *Angew Chem Int Ed* **2024**, *63*, e202409193.
- [12] T. Zheng, T. Xu, J. Xiong, W. Xie, M. Wu, Y. Yu, Z. Xu, Y. Liang, C. Liao, X. Dong, Y. Xia, Y. Cheng, Y. Xia, P. Müller-Buschbaum, *Advanced Science* **2024**, 2410329.

- [13] P. Liang, J. Li, Y. Dong, Z. Wang, G. Ding, K. Liu, L. Xue, F. Cheng, *Angew Chem Int Ed* **2024**, e202415853.
- [14] M. Klein, M. Binder, M. Koželj, A. Pierini, T. Gouveia, T. Diemant, A. Schür, S. Brutti, E. Bodo, D. Bresser, J. L. Gómez-Urbano, A. Balducci, *Small* **2024**, 20, DOI 10.1002/sml.202401610.
- [15] Y. Zhang, L. Zeng, Z. Ding, W. Wu, L. Deng, L. Yao, *Chem Commun (Camb)* **2023**, 59, 12593–12596.
- [16] L. Chen, J. Lu, Y. Wang, P. He, S. Huang, Y. Liu, Y. Wu, G. Cao, L. Wang, X. He, J. Qiu, H. Zhang, *Energy Storage Materials* **2022**, 49, 493.
- [17] X. Li, J. Zheng, M. H. Engelhard, D. Mei, Q. Li, S. Jiao, N. Liu, W. Zhao, J.-G. Zhang, W. Xu, *ACS Appl. Mater. Interfaces* **2018**, 10, 2469–2479.

Reviewer #3:

Isidora Cekic-Laskovic et al. investigated the dissolution mechanisms of SUS316 in lithium-ion batteries and proposed an electrolyte-optimization strategy to enhance the anti-corrosion performance of SUS316 and Al foil. This work is meaningful for coin cells. Upon the work, the results find that Cl⁻ as impurity promotes SUS316 to generate pit corrosion (50 ppm as the fringe conditions), and FSI-corrosion effect on SUS spacer behaves more severe pit corrosion. However, the corrosion of Al foil is the most devastating for the electrode corrosion of coin cells in three dimensions. Because it triggers the binding-force failure between the cathode materials and the binder. Some deficiencies should be addressed:

As the reviewer pointed out, this work holds particular significance for coin-cell configurations. However, we would like to emphasize that this is not limited to this format; it is highly relevant for other cell formats in which stainless steel components are in contact with electrolytes, such as large format cylindrical cells (e.g. Tesla 4680 format). The use of steel casing could largely increase the mechanical strength and improve the safety. Gorsch et al. have conducted a teardown of Tesla 4680 cylindrical cells and confirmed the use of nickel-plated steel as casing materials^[18]. Though the nickel plating could increase the dissolution resistance of steel, it might still undergo steel dissolution under high-voltage or high-temperature conditions in the presence of FSI and trace Cl⁻ anions, especially in cases where the nickel-plated layer has sustained mechanical damage. Therefore, understanding and mitigating the corrosion behavior of stainless steel in advanced electrolyte systems remains essential for ensuring long-term stability across various battery formats. The insights gained here directly contribute to industrial-scale battery advancements, leading to improved stability and performance of LIBs.

In addition, we agree that the corrosion/dissolution of Al foil also plays a role in LiFSI-based electrolytes and the concerns regarding Al dissolution will be addressed in subsequent responses.

1. Why were the aqueous solutions used in Line 308 to evaluate the static (corrosive) effects of three ions on SUS316? Is it to prove that the solvent does not have an impact on the competitive adsorption and passivation of DFOB⁻? Since the working environment of SUS316 in the battery is the ester-based organic solvents.

We thank the reviewer for this insightful question. Aqueous solutions were used in our study to compare the static corrosive effects of specific anions (Cl⁻, FSI⁻, and DFOB⁻) on SUS316 under controlled conditions, without interference from the competitive adsorption from solvents and complex solvation structures in organic solvents.

However, we fully acknowledge that the actual working environment of SUS316 in batteries involves presence of ester-based organic electrolytes, which might behave differently. Therefore, the aqueous experiments serve primarily as a comparative baseline, not a direct mimic of the battery environment. To address this limitation, we also carried out complementary experiments in organic electrolytes. As shown in Figure S 10, pitting is observed on SUS immersed in electrolytes including LiCl and LiFSI, whereas no visible pitting is observed on SUS immersed in electrolytes with LiDFOB. The consistency

between trends observed in aqueous and organic systems supports our hypothesis regarding the competing adsorption of anions.

The following text and figures have been included in the main text and Supplementary Information to clarify the aim of the experiments:

“Therefore, storage experiments were performed by immersing SUS316 spacers in aqueous solutions containing different conducting salts to exclude the interference from the competitive adsorption from solvents and complex solvation structures in organic solvents. The SEM images of SUS316 spacers display deposits and pits when immersed in LiCl- and LiCl+LiFSI-based solutions (Figure 4d,f), while clean surface without pits was observed on SUS316 spacer immersed in LiCl+LiDFOB-based electrolyte (Figure 4e). The same trend was also observed in complementary experiments in organic electrolyte systems shown in Figure S 10. In general, the DFOB⁻ anions preferentially adsorb and accumulate on the SUS surface, thereby excluding FSI⁻ and Cl⁻ anions from attacking the SUS surface.”

Figure S 10. SEM image of SUS316 surface after 1 month immersed in (a) LiCl (100 ppm LiCl in EC/EMC 3:7 by wt%), (b) LiCl+LiDFOB (100 ppm LiCl + 0.5 M LiDFOB in EC/EMC 3:7 by wt%) and (c) LiCl+LiFSI (100 ppm LiCl + 0.5 M LiFSI in EC/EMC 3:7 by wt%) electrolytes.

2. *There is no clear link between Gas evolution and SUS corrosion. The authors may need to combine the correlation of solvent decomposition and SUS dissolution and deposition on the Li anode. Gas evolution has nothing to do with the corrosion mechanism demonstrated in Line 350 “Proposed SUS dissolution mechanism”, although it is not conducive to maintaining the stable cycle life of the batteries.*

We thank the reviewer for the suggestion to clarify the correlation between solvent decomposition, SUS dissolution and deposition on the Li electrode. Furthermore, we acknowledge the reviewer's point that there is indeed no direct relationship between gas evolution and SUS corrosion/dissolution. Therefore, we have revised the manuscript accordingly, correlating the observed gas evolution to the electrolyte decomposition process, which may be initiated or catalyzed by the Fe ions dissolved from SUS spacer or casing materials.

The following text have been included in the revised manuscript:

“During the SUS dissolution process, transition metals (TMs) such as Fe ions dissolved from SUS play a critical role in the roll-over failure in LIBs^[43–45]. These TMs can accelerate electrolyte decomposition, damage the SEI and induce lithium dendrite growth. Gas evolution as a critical consequence of electrolyte decomposition, is particularly important to analyze in this process to assess the inhibition effect of LiDFOB. The gas evolution during the dissolution process was performed using 3-electrode differential electrochemical mass spectrometry (DEMS) cells connected to MS while carrying out an LSV measurement with a scan rate of 0.1 mV s⁻¹.

In terms of other electrolytes, DEMS cells with FSI₁DFOB₀ release CO₂ slightly earlier compared to LiDFOB-containing electrolytes. In addition, CO and O₂ release is also observed in DEMS cells with FSI₁DFOB₀ electrolytes, indicating that electrolyte decomposition is triggered by the SUS dissolution. Notably, gas evolution is substantially suppressed in the presence of LiDFOB and no H₂, CO and O₂ were observed in DEMS cell with FSI₀DFOB₁ and FSI₃DFOB₅. These gas evolution results confirmed that SUS dissolution induces electrolyte decomposition, and that the introduction of LiDFOB can considerably mitigate electrolyte decomposition by inhibiting the dissolution process.”

In addition, we have added the discussion on the Li metal surface to confirm the deposition of Fe ions originated from the SUS dissolution. The following text, figures and table have been included in the main text and Supplementary Information:

“These reactions form soluble complexes, preventing the formation of Fe_2O_3 and facilitating continued metal dissolution, which extends the local pitting. The dissolved Fe ions can diffuse through the electrolytes and deposit on the electrode surface. The Li metals recovered from the cells were then examined using SEM-EDX to investigate the migration of transition metal ions (TMs). Figure S 7 shows the SEM images of Li metals harvested after CA measurements. Cracks and inhomogeneous morphologies are observed on the Li metal surface harvested from cells with $\text{FSI}_1\text{DFOB}_0$ and LiCl_{sat} , which can be attributed to the accumulation of Fe ions, as confirmed by EDX analysis in Table S 2.”

Figure S 7. SEM images of (a) SUS316 spacers and (b) Li metal recovered from cells containing LiCl and LiDFOB-based electrolytes after 20 h of CA measurement at 4.2 V vs. $\text{Li} | \text{Li}^+$.

Table S 2. Elemental distribution on the surface of Li collected by EDX.

	Li (%)	C (%)	O (%)	F (%)	N (%)	S (%)	Fe (%)
LiCl_{sat}	0.00	38.34	44.65	0.00	0.00	0.00	17.01
$\text{FSI}_1\text{DFOB}_0$	0.00	16.70	46.75	4.95	1.51	9.73	20.36
$\text{FSI}_2\text{DFOB}_8$	85.76	3.98	7.07	1.71	1.48	0.00	0.00
$\text{FSI}_5\text{DFOB}_5$	86.95	3.31	6.22	1.87	1.55	0.00	0.00
$\text{FSI}_0\text{DFOB}_1$	80.31	8.86	7.73	1.55	1.25	0.00	0.00

The dissolution mechanism and proposed graph demonstrated in Line 350 “Proposed SUS dissolution mechanism” have also been modified accordingly:

“It needs to be highlighted that the SUS surface is largely covered by Fe_2O_3 , slowing down the continued SUS dissolution process and decreasing the anodic current even at higher potentials. Furthermore, dissolved Fe ions migrate and deposit on the electrode surface, where they damage the SEI and promote crack formation. In addition, the evolution of CO_2 and H_2 observed during the LSV measurements provides clear evidence of electrolyte and SEI decomposition.

With this continued SUS dissolution, large pits are formed and almost no re-passivation can be observed. Concurrently, Fe deposits accumulate on the Li metal surface, resulting in crack formation and an inhomogeneous surface. Furthermore, electrolyte decomposition induced by the SUS dissolution process results in a large volume of gas evolution including H_2 , CO_2 , CO and O_2 , which could affect the safety properties of the LIBs.

LiDFOB preferentially adsorbs and accumulates on the SUS surface, excluding the Cl^- and FSI^- anions from the surface, thus preventing SUS dissolution at high potentials. In addition, the release of H_2 , CO and O_2 is also inhibited due to the mitigated TMs with the addition of LiDFOB, highlighting the importance of LiDFOB in stabilizing LiFSI-based electrolytes.”

Figure 6. Illustration of proposed mechanism scenarios. (a) SUS dissolution process in the presence of Cl⁻ anions. (b) SUS dissolution process in the presence of LiFSI-based electrolyte. (c) Inhibiting effect of LiDFOB on the SUS dissolution in the LiFSI-based electrolyte. The exact structures of the reactant molecules and complexes involved in these reactions are not known.

3. The authors may need to add the Operando gaseous evolution of the coin-cell at open circuit potential. The result can be used as the baseline to elucidate the initial state of the polarization process of different electrolytes.

We thank the reviewer for pointing out the importance of *operando* gaseous evolution open circuit potential. Indeed, in our *operando* gaseous evolution measurements using the ECC-DEMS cell configuration (Figure 1), we applied a 2 h open circuit potential (OCP) holding period with continuous Ar flow prior to the LSV, in order to stabilize the gas environment. This OCP holding period provides a reliable baseline to elucidate the initial state of the polarization process for different electrolytes. As shown in Figure S 2, the *operando* gaseous evolution at OCP reveals prominent initial signals for $m/z = 28$, which can correspond to either CO or N₂. Given the fact that N₂ is the largest component of air, the signal is attributed to N₂ in the OCP holding period. Similarly, signals corresponding to other air components (H₂O: $m/z = 18$; O₂: $m/z = 32$, CO₂: $m/z = 44$) were also observed at the beginning of the

measurement. These signals rapidly declined as the Ar purge effectively remove the residual air in the connecting tubes. Notably, the H₂O signal shows a slow decay, possibly due to residual moisture adsorbed on the equipment (e.g. DEMS cell chamber, mass spectrometry, gas lines, etc.). Importantly, no detectable evolution of gases was observed during the OCP period.

Figure 1. Schematic overview of an ECC-DEMS cell. The SUS316 spacer was used as a WE, while lithium was used as both the CE and RE. The evolved gases are carried by a continuous Ar flow introduced through the gas inlet, pass through the spiral-type flow field below the working electrode, and come out from the gas outlet for real-time detection *via* mass spectrometry

The following text and figures have been included in the main text and Supplementary Information:

“DEMS measurements were conducted in an ECC-DEMS cell (EL-Cell) with a gas inlet and outlet connected via the capillary to the MS. A SUS316 spacer was used as a working electrode and Li was used as a counter and reference electrode. An LSV measurement was conducted for the assembled ECC-DEMS cells from 0 to 5 V at a scan rate of 0.1 mV s⁻¹. Prior to each DEMS measurement, Ar gas was flowed through the ECC-DEMS cells for 2 h to stabilize the gas environment and no detectable evolution of gases was observed during the open circuit potential (OCP) period (Figure S 2).”

Figure S 2. Operando gaseous evolution of the DEMS cell at open circuit potential, the gases are normalized with Ar to eliminate the fluctuation of carrier gas.

4. Line 406, the authors think that the pronounced rise in anodic current stemmed from SUS dissolution; however, according to Figure S13a, the anodic current may also be introduced by Al corrosion. It can be clarified by SEM morphologies of the SUS316 case and Al foil in batteries after 0.1C formation.

We thank the reviewer for raising the important point regarding the origin of the anodic current, which may arise not only from SUS dissolution but also from Al dissolution. In response to this concern, we disassembled cells containing FSI₁DFOB₀ electrolytes after the anodic current increase was observed during the charging process and conducted SEM analysis on both SUS spacer and the coated Al foil. Considering that the Al dissolution can also take place on the coated side of Al and potentially trigger the binding-force failure between the electrode materials and the binder, we carefully scratched off the active materials using tweezers to directly expose the Al foil under the coating. With this sample preparation, we could observe the morphology of the Al foil on the coated side *via* SEM analysis. The pristine NMC811 electrode was used as the reference and two cells using different coin cell parts (SUS316 and SUS316L) were disassembled to compare the dissolution behavior of SUS and Al by using different SUS grades. The cells using SUS316L coin cell parts show a rise of anodic current at the 3rd charging step, suggesting higher dissolution resistance compared to cells using SUS316 parts (Figure S 16).

Looking into the Al foil from the NMC811 coated side, we could clearly observe surface scratches and remaining active materials after the mechanical removal of the electrode layer from the pristine NMC811 electrode sample (Figure S 17a). In addition, small indentations were observed on the Al surface, likely due to the particles being pressed into the Al foil during the calendaring process. In sample

FSI₁DFOB₀/316, the Al shows pronounced dissolution, forming large crevices on the coated side (Figure S 17b). Notably, similar crevices were also observed on the uncoated side of Al foil (Figure S 17h). The morphology of these crevices presented localized breakdown due to stress-induced fracture, which is a common characteristic of pitting corrosion^[19,20]. In contrast, no obvious pitting or crevices formation was observed on both sides of Al foil from sample FSI₁DFOB₀/316L (Figure S 17c,i).

Upon examining the SUS spacers from both cells, large pits were observed, predominantly distributed along the spacer edges (Figure S 17n,o). This is expected as the central area of the SUS spacer is covered by the electrode, and only the edge part has direct exposure to the electrolytes. The morphology and size of the pits were similar between sample FSI₁DFOB₀/316 and FSI₁DFOB₀/316L, but considerable deposits were observed on the SUS spacer for sample FSI₁DFOB₀/316. In general, these results indicate that the pronounced rise in anodic current observed in Figure 7a (using SUS316 as coin cell parts) originated from both SUS dissolution and Al dissolution, which contribute to cell failure in the presence of LiFSI-based electrolytes. Interestingly, the use of SUS316L (higher dissolution resistance) appears to prevent the Al dissolution. Nevertheless, SUS dissolution was evident on both cells, suggesting that the SUS dissolution is more severe compared to the Al dissolution and is the dominant factor responsible for the rise of anodic current in Figure 7a.

To further validate that the SUS dissolution is the primary cause of cell failure and the observed rise in anodic current, we introduced LiPF₆ as a co-salt into the LiFSI-based electrolyte to passivate the Al current collector. LiPF₆ is widely recognized as an effective Al dissolution inhibitor and is usually used as an additive or co-salt to protect the Al current collector^[21–23] in LiFSI and LiTFSI-based electrolytes. To ensure the comparability, we used an equivalent conducting salt ratio in the blended electrolyte (0.5M LiFSI + 0.5M LiPF₆ in EC:EMC 3:7 by wt%) and noted as FSI₅PF₅. As shown in Figure S 16, the voltage vs. time profiles of cells containing FSI₅PF₅ still exhibit a pronounced rise in anodic current during the constant voltage step. This indicates that the presence of LiPF₆ could not prevent the cell failure during the charging step. Furthermore, SEM images confirmed that no evidence of Al dissolution was observed on the electrode current collector, while clear pitting was still observed on the SUS spacer (Figure S 17p). These results reinforce the conclusion that SUS dissolution plays the dominant role in cell failure in LiFSI-based electrolytes.

The following text and figures have been included in the main text and Supplementary Information:

“As depicted in **Figure 7a**, the cells assembled with NMC811 as electrode material on an Al foil current collector and graphite as electrode material utilizing the FSI₁DFOB₀ electrolyte exhibit a pronounced rise in anodic current during the constant voltage step, which can be an effect of SUS or Al dissolution. With further *post mortem* analysis discussed in the Supplementary Information, this rise in anodic current is mostly attributed to the SUS dissolution.

Investigation of Al Dissolution during the Cycling Process

It has been noticed that NMC811||Graphite cells containing FSI₁DFOB₀ are not able to charge to 4.2 V, and a pronounced rise in anodic current is observed during the CV step (**Figure 7a**). This anodic current may be introduced by SUS or Al dissolution. To distinguish the contribution from SUS and Al dissolution, the cells containing FSI₁DFOB₀ electrolytes were disassembled after the charging process and analyzed using SEM to examine both the SUS spacer and coated Al foil. Given that Al dissolution may also occur on the coated side of Al and potentially disturb the adhesion between the active materials and binder, the active material was carefully removed using tweezers to expose the underlying Al foil. This sample preparation enabled direct observation of the Al surface morphology on the coated side. Pristine NMC811 served as the reference, and two cells assembled with different steel grades (SUS316 vs. SUS316L) were disassembled to examine the respective dissolution behavior of SUS and Al. The cells assembled with SUS316L parts exhibited an increase in anodic current at the 3rd charging step, suggesting higher dissolution resistance compared to cells using SUS316 parts (**Figure S 16**).

Inspection of the Al foil from the NMC811 coated side revealed visible surface scratches and residual active materials remaining after mechanical removal of the electrode layer in the pristine NMC811 sample (Figure S 17a). Additionally, small indentations were identified on the Al surface, likely resulting from the compression of active material particles into the Al foil during the calendaring process.

In sample $\text{FSI}_1\text{DFOB}_0/316$, pronounced Al dissolution was observed, characterized by the formation of large crevices on the coated side (Figure S 17b). Notably, similar crevices were also observed on the uncoated side of Al foil (Figure S 17h). The morphology of these crevices presented localized breakdown due to stress-induced fracture, which is a common characteristic of pitting corrosion^[6,7]. In contrast, no obvious pitting or crevices formation was observed on both sides of Al foil from sample $\text{FSI}_1\text{DFOB}_0/316\text{L}$ (Figure S 17c,i).

Upon examining the SUS spacers from both cells, extensive pitting was observed, predominantly distributed along the edges of the spacer (Figure S 17n,o). This is consistent with expectations, as the central area of the SUS spacer is covered by the electrode, while the edge areas remain exposed to the electrolytes. The morphology and size of the pits were similar between sample $\text{FSI}_1\text{DFOB}_0/316$ and $\text{FSI}_1\text{DFOB}_0/316\text{L}$, but considerable deposits were observed on the SUS spacer for sample $\text{FSI}_1\text{DFOB}_0/316$. In general, these results indicate that the pronounced rise in anodic current observed in Figure 7a (using SUS316 as coin cell parts) originated from both SUS dissolution and Al dissolution, which contributed to cell failure in the presence of LiFSI-based electrolytes. Interestingly, the use of SUS316L (higher dissolution resistance) appears to prevent the Al dissolution. Nevertheless, SUS dissolution is evident on both cells, suggesting that the SUS dissolution is more severe compared to the Al dissolution and is the dominant factor responsible for the rise of anodic current in Figure 7a.

Figure S 16. Voltage vs. time profiles of cells with SUS316/316L parts using $\text{FSI}_1\text{DFOB}_3$ and FSI_5PF_5 electrolytes.

To further validate that the SUS dissolution is the primary cause of cell failure and the observed rise in anodic current, we introduced LiPF_6 as a co-salt into the LiFSI-based electrolyte to passivate the Al current collector. LiPF_6 is widely recognized as an effective Al dissolution inhibitor and is usually used as an additive or co-salt to protect the Al current collector in LiFSI and LiTFSI-based electrolytes^[21-23]. To ensure the comparability, we used an equivalent salt ratio in the blended electrolyte (0.5M LiFSI + 0.5M LiPF_6 in EC:EMC 3:7 by wt%) and noted as FSI_5PF_5 . As shown in Figure S 16, the voltage vs. time profiles of cells containing FSI_5PF_5 still exhibit a pronounced rise in anodic current during the constant voltage step. This indicates that the presence of LiPF_6 could not prevent the cell failure during the charging step. Furthermore, SEM images confirmed that no evidence of Al dissolution was observed on the electrode current collector (Figure S 17d,j), while clear pitting was still observed on the SUS spacer (Figure S 17p). These results reinforce the conclusion that SUS dissolution plays the dominant role in cell failure in presence of LiFSI-based electrolytes.”

Figure S 17. SEM image of (a) Al foil from the coated side of the NMC811 electrode, (g) Al foil from the uncoated side, and (m) pristine SUS316L spacer. The following images correspond to the Al foil from the coated side, uncoated side, and the SUS spacer after disassembling different cells: (b, h, n) cells with FSI₁DFOB₀ and SUS316 coin cell parts (1st cycle); (c, i, o) cells with FSI₁DFOB₀ and SUS316L coin cell parts (3rd cycle); (d, j, p) cells with FSI₅PF₅ and SUS316 coin cell parts (1st cycle); (e, k, q) cells with FSI₅DFOB₅ and SUS316 coin cell parts (282nd cycle); (f, l, r) cells with FSI₅DFOB₅ and SUS316L coin cell parts (1365th cycle).

5. Similarly, a rapid drop in Coulombic efficiency (CE) may also be introduced by Al dissolution. It is suggested that the authors disassemble the cycled batteries and analyze the corrosion conditions of SUS and Al foil.

Following the reviewer's point, we disassembled the cells containing FSI₅DFOB₅ after the galvanostatic cycling procedure. Similarly, two cells using different coin cell parts (SUS316 and SUS316L) were

disassembled. No clear evidence of Al dissolution was observed on both sides of the electrodes in both cases (Figure S 17e,f,k,l). Regarding the SUS spacers, we did not observe pits on both SUS316 and SUS316L spacers (Figure S 17q,r). However, we could observe considerable blackish deposits on the edge of the SUS316 spacer, indicating iron oxidation. Furthermore, the surface of the SUS316 spacer appeared more uneven compared to the pristine sample in Figure S 22a, indicating that the dissolution may have occurred in the form of crevice corrosion rather than pitting. It is worth noting that no obvious current increase was observed during cycling, implying that the SUS dissolution in cells with FSI₅DFOB₅ proceeded much more slowly compared to that in cells with FSI₁DFOB₀. This likely explains the absence of pits and the presence of only surface deposits. Based on observations from SEM, we can conclude that the SUS dissolution is the main reason for the rapid drop in Coulombic efficiency in Figure 7b.

The following text and figures have been included in the main text and Supplementary Information:

“As depicted in **Figure 7a**, the cells assembled with NMC811 as electrode material on an Al foil current collector and graphite as electrode material utilizing the FSI₁DFOB₀ electrolyte exhibit a pronounced rise in anodic current during the constant voltage step, which can be an effect of SUS or Al dissolution. With further *post mortem* analysis discussed in the Supplementary Information, this rise in anodic current is mostly attributed to the SUS dissolution. This severe SUS dissolution detrimentally affects the galvanostatic cycling performance of the cell, leading to failure in the initial cycle. In contrast, the cell using LiDFOB-containing electrolyte (FSI₅DFOB₅) demonstrates stable charging and discharging behavior, as confirmed by the decent cycling stability over the first 150 cycles (**Figure 7b**). However, a large deviation of discharge capacity along with a rapid drop in Coulombic efficiency (CE) in the subsequent cycles is observed, indicating ongoing SUS dissolution. This can be confirmed with further *post mortem* analysis discussed in Supplementary Information.

Similarly, a rapid drop in CE in cells with FSI₅DFOB₅ and SUS316 coin cell parts may also be introduced by Al dissolution, as shown in Figure 7b. To confirm the reason, cells containing FSI₅DFOB₅ after certain cycle numbers were disassembled and the dissolution states of SUS and Al foil were examined.

Two cells using different coin cell parts (SUS316 and SUS316L) were disassembled. No clear evidence of Al dissolution was observed on both sides of the electrodes in both cases (Figure S 17e,f,k,l). With respect to the SUS spacers, no distinct pitting was identified on either SUS316 or SUS316L (Figure S 17q,r). However, prominent blackish deposits were detected along the edge of the SUS316 spacer, indicative of iron oxidation. Furthermore, the surface of the SUS316 spacer appeared more uneven compared to the pristine sample in Figure S 22a, indicating that the dissolution may have occurred in the form of crevice corrosion rather than pitting. Notably, no obvious current increase was observed during cycling, implying that the SUS dissolution in cells with FSI₅DFOB₅ proceeded much more slowly compared to that in cells with FSI₁DFOB₀. This is likely to explain the absence of pits and the presence of only surface deposits. Based on observations from SEM, it can be concluded that the SUS dissolution is the main reason for the rapid drop in CE in Figure 7b.”

6. CO curve of FSI₁₀DFOB₁ in Figure 5d does not correspond to that of Figure S2c. Please check them.

We thank the reviewer for pinpointing this misalignment, we have accordingly corrected the Figure 5d.

Figure 5. (a) LSVs of a 3-electrode DEMS cells using SUS316 as working electrode with different electrolytes at scan rate of 0.1 mV s⁻¹. Operando gaseous evolution of (b) H₂, (c) CO₂, (d) CO and (e) O₂ by DEMS are monitored during the LSV measurement, the gases are normalized with Ar to eliminate the fluctuation of carry gas.

7. The collapse of a full battery involves multiple issues such as degradation of the cathode materials, corrosion of current collectors, and degradation of the anode materials. Therefore, it is more appropriate to evaluate the anti-corrosion performance of DFOB- using a lithium metal anode. The compatibility between anode materials and FSI5DFOB5 needs to be further evaluated in Graphite||Li half cells. In this way, Figure 7b/d will be more persuasive.

We thank the reviewer for this suggestion. Indeed, the roll-over effect of the battery involves multiple issues, such as the loss of active materials from both electrodes, loss of lithium inventory, corrosion of current collectors, electrolyte decomposition, etc. To understand the degradation mechanism, efforts have been made by combining different post mortem analysis and advanced in-operando techniques^[24].

The use of the Li metal as an electrode in the half-cell configuration is beneficial for the performance evaluation of electrode materials, due to the simple structure and approximately infinite lithium reservoir^[25]. In addition, the electrochemical properties relative to Li⁺|Li can be obtained in a half-cell measurement configuration, allowing comparison of different electrode materials^[25]. However, half cells using Li metal have several important limitation to consider: First, Li metal electrode suffer from serious safety problem of dendrite lithium^[26,27]. For example, organic carbonate electrolytes exhibit severe interfacial resistance growth and dendrite formation against Li metal^[26], whereas their behavior against graphite/LTO electrodes can differ substantially^[27]. In addition, Li metal creates an infinitely large lithium reservoir and a highly reactive interface. This can either under- or overestimate side reactions and SEI formation relative to practical electrodes. Furthermore, Graphite||Li half cells are typically galvanostatically cycled within a narrow voltage window (0.01 to 1.5 V vs. Li|Li⁺), which is

considerably lower than the NMC811||Graphite cells that operate above 4 V. These high voltage conditions are critical, as they can trigger oxidative decomposition, CEI formation, and gas evolution, which are essential to evaluating electrolyte performance under realistic conditions.

However, to confirm that our optimized electrolyte is also compatible with the NMC811||Li and Graphite||Li cell chemistries, we have conducted additional galvanostatic cycling performance evaluation with selected electrolytes in both cell configurations and supplemented our results with relevant literature findings.

The compatibility of LiFSI and LiDFOB in the NMC811||Li has already been studied in the past. Zhang *et al.* investigated a blended salt localized high-concentration electrolyte (LHCE) using LiFSI and LiDFOB with excellent high-voltage compatibility for NMC811||Li pouch cells^[15]. In their study, the use of pouch cell setup eliminates the SUS parts and makes the LiFSI-based electrolyte less challenging. The use of LHCE containing 0.5M LiFSI and 0.5M LiDFOB outperformed LHCE with 1M LiDFOB and commercial carbonate electrolyte (1M LiPF₆ in EC/DMC 3:7 by volume ratio) and reached a capacity retention of 70.2 % after 300 cycles. In addition, Klein *et al.* used the LiDFOB as an additive in LiFSI-based electrolyte and also showed increased galvanostatic cycling stability compared to the baseline electrolyte counterparts without LiDFOB^[14]. The optimized electrolyte with LiFSI/LiDFOB could effectively protect the Al current collector and from effective CEI incorporating LiF and BF_x. Notably, the Swagelok cell type was used in their study. Though the Swagelok is mostly made from SUS, but the use of Mylar foil could effectively prevent the contact between the electrolyte and the body of the Swagelok. In general, the blended salt approach should be compatible with the NMC811 electrode materials without the impact of SUS dissolution.

With this understanding, we evaluate the optimized electrolytes in NMC811||Li cells. Interestingly, the cells with FSI₁DFOB₀ electrolytes can be galvanostatically cycled at 4.2 V when the Li metal electrode is employed (Figure S 18a), which is different compared to cells NMC811||Graphite galvanostatically cycled at 4.2 V (Figure 7a). With 3-electrode cell we could observe that the actual voltage of NMC811 vs. Li reference is at 4.28 V rather than 4.2 V due to the higher potential of lithiated graphite (Figure S 18d). To further confirm it, we increased the upper cut-off voltage (UCV) to 4.3 V and observe again the rise of anodic current (Figure S 18e). Though the NMC811||Li cells could be galvanostatically cycled at 4.2 V, the low CE (< 98 %) of both cells (cell with SUS316 and cell with SUS316L) suggests the existence of SUS/Al dissolution. Cell using SUS316 parts ends at the 32nd cycle with a rise of anodic current during the constant voltage step (Figure S 18b). Cell using SUS316L parts shows higher CE compared to the cell with SUS316 and could be galvanostatically cycled up to 133 cycles. However, the rise of anodic current at the 38th cycle also indicates the existence of SUS/Al dissolution (Figure S 18c).

As for cells with FSI₅DFOB₅, a similar trend could also be observed compared to the full cell in Figure 7b. NMC811||Li cells using SUS316L coin cell parts demonstrated longer cycling stability compared to cells using SUS316 coin cell parts (Figure S 19a). NMC811||Li cells using SUS316 have a fast drop of CE at ≈100 cycles, which is similar to the trend observed in full cells. In addition, NMC811||Li cells using SUS316L coin cell parts with FSI₅DFOB₅ electrolytes outperform cells with LP57, indicating that the optimized electrolyte is also suitable for NMC811||Li cells. Notably, the use of different SUS grades has no impact on the galvanostatic cycling performance of cells containing LP57, which is the same as in full cell. However, the life time of the NMC811||Li cells (no matter which electrolyte) is inferior to the NMC811||Graphite full cell. The faster fading of the NMC811||Li cells is attributed to the lithium dendrite growth leading to possible micro-shorts, and the high surface area lithium (HSAL) accelerates the electrolyte decomposition^[27,28]. In addition, we employed 1C charge current rate also in cycling stability measurements for the NMC811||Li cells, which is critical for the Li metal electrode^[29]. Zhang *et al.* show that slow charging (0.2C) and fast discharging (3C) considerably improve performance of NMC811||Li cells, while fast charging (1C) and slow discharging (0.33C) deteriorate the cycling performance due to the formation of HSAL^[29].

Similar fast fading was also observed in Graphite||Li cells. As shown in Figure S 19b, no cells could hold a stable cycle after 100 cycles, which is even worse than NMC811||Li cells. Smith *et al.* observed similar trends by comparing the long term cycling performance between NMC811||Li,

NMC811||Graphite and Graphite||Li cells and conclude this as the higher areal capacity of graphite, thicker surface layer on graphite side which leads to more side reactions on the Graphite||Li cells^[28]. However, differences could still be observed between cells with different electrolytes. Cells with LP57 have a capacity decay already after 40 cycles, while cells with FSI₁DFOB₀ and FSI₅DFOB₅ shows a good cycling stability until the roll-over due to the dendrite growth with the 1C current rate. LiFSI and LiDFOB has been proven as suitable salts for graphite electrode. Liu et al. shows that with ultra-low concentration 0.16M LiDFOB /EC-DMC, the Graphite||Li cells shows a high capacity retention of 89.4% after 184 cycles^[30]. In addition, Kang et al. have confirmed an increased cycling stability and superior thermal stability of the graphite electrode in LiFSI electrolyte^[31].

To further confirm the roll-over failure in Graphite||Li cells, the cycled cells were disassembled and both graphite and Li metal were examined with SEM images. It is in general that no clear changes could be observed on the graphite particles compared to the pristine graphite, except that the cells with FSI₅DFOB₅ shows particle fracture (Figure S 20). Furthermore, The Li surface of cells with FSI₅DFOB₅ shows also cracking on the Li metal and HSAL on the surface (Figure S 20 h), which can lead to accelerated electrolyte decomposition and electrode degradation. The EDX confirmed the considerable amount of electrolyte decomposition products on both graphite and Li metal surface (Table S 3 and Table S 4). It can be concluded that the roll-over failure is mostly attributed to the HSAL on the Li metal surface. In general, we can see that the FSI₅DFOB₅ electrolyte has good compatibility with graphite electrode materials.

The following text and figures have been included in the main text and Supplementary Information:

“The electrodes were dried at 120 °C under vacuum ($<10^{-2}$ mbar) for 12 h and stored in the argon-filled glovebox before cell assembly. Celgard 2500 membrane was used as a separator. Li metal with a thickness of 500 μm (used for LSV and CA measurements) was purchased from China Energy Lithium CO. Ltd while Li metal with a thickness of 150 μm (used for NMC811||Li and Graphite||Li cells) was purchased from Albemarle. Both types of Li metals were punched into $\phi 15$ mm disks and kept inside the argon filled glovebox as well.

NMC811||Si-C cells with FSI₅DFOB₅ maintain an effective CE with nearly double the lifespan compared to cells with LP57, reaching ≈ 300 cycles (**Figure 7g**). Furthermore, the FSI₅DFOB₅ electrolytes exhibit improved performance in both NMC811||Li and graphite||Li cells, confirming the compatibility of the optimized electrolyte also in cells with Li metal electrode, which is detailed in Supplementary Information.

Investigation of the Anti-Dissolution Performance of LiDFOB in NMC811||Li and Graphite||Li Cell

To confirm that the optimized electrolyte is also compatible with the NMC811||Li and graphite||Li cell chemistries, additional galvanostatic cycling performance was conducted with selected electrolytes in both configurations.

A stable cycling at 4.2 V was achieved in cells containing the FSI₁DFOB₀ electrolyte when lithium metal was used as the counter electrode (Figure S18a), which is different compared to cells NMC811||Graphite cycled at 4.2V (Figure 7a). With a three-electrode cell revealed that the actual potential of NMC811 vs. Li reference electrode is 4.28 V, rather than the nominal 4.2 V, due to the higher potential of lithiated graphite (Figure S18d). To further confirm it, the upper cut-off voltage (UCV) was increased to 4.3 V, which again resulted in a pronounced rise in anodic current (**Error! Reference source not found.e**). Although the NMC811||Li cells could sustain cycling at 4.2 V, the low CE (< 98 %) observed in both SUS316 and SUS316L parts suggests the existence of SUS/Al dissolution. The cell using SUS316 parts ends at the 32nd cycle with a rise of anodic current during the constant voltage step (Figure S18b). In comparison, cells with SUS316L parts demonstrate improved CE and extend the cycling stability up to 133 cycles. However, the rise of anodic current at the 38th cycle also points to the existence of SUS/Al dissolution.

Figure S 18. (a) Specific discharge capacity vs. cycle number for NMC811||Li cells containing SUS316/316L parts with FSI₁DFOB₀ electrolyte. Current vs. time profile during the constant voltage step at various cycles for cells assembled with (b) SUS316 and (c) SUS316L coin cell parts. (d) Voltage vs. time profile of a three-electrode cell using LP57 electrolyte. (e) Voltage vs. time profiles of NMC811||Li cells with SUS316/316L parts using FSI₁DFOB₀ electrolytes.

As for cells with FSI₅DFOB₅, a similar trend could also be observed compared to the NMC811||graphite cells in Figure 7b. NMC811||Li cells using SUS316L coin cell parts demonstrate extended cycling stability compared to cells assembled with SUS316 coin cell parts (Figure S 19a). In particular, NMC811||Li cells assembled with SUS316 parts have a fast drop of CE at ≈ 100 cycles, consistent with the behavior observed in full cells. In addition, NMC811||Li cells assembled with SUS316L parts with FSI₅DFOB₅ electrolytes outperform those with LP57, indicating that the optimized electrolyte is also effective in NMC811||Li cell configuration. Notably, the use of different SUS grades has no impact on the galvanostatic cycling performance of cells containing LP57, consistent with observations from full cells. However, the overall life time of the NMC811||Li cells (no matter which electrolyte) is inferior to the NMC811||Graphite full cell. The faster fading of the NMC811||Li cells is attributed to the lithium dendrite growth, which can lead to possible micro-shorts. Furthermore, the formation of high surface area lithium (HSAL) accelerates electrolyte decomposition^[11]. It should be noted that 1C charge current rate was employed in the cycling stability measurements for the NMC811||Li cells, which is critical for the Li metal electrode^[12]. Zhang et al. show that slow charging (0.2C) and fast discharging (3C) considerably improve the performance of NMC811||Li cells, while fast charging (1C) and slow discharging (0.33C) deteriorate the galvanostatic cycling performance due to the formation of HSAL^[12].

A similar rapid capacity fading was also observed in Graphite||Li cells. As shown in **Error! Reference source not found.**, none of the tested cells maintained a stable cycle beyond 100 cycles, with performance degradation even more severe than that observed in NMC811||Li cells. Smith et al. observed similar trends by comparing the long term cycling performance between NMC811||Li, NMC811||Graphite and Graphite||Li cells and conclude this as the higher areal capacity of graphite, thicker surface layer on graphite side which leads to more side reactions on the Graphite||Li cells^[11]. Despite the overall instability of this configuration, notable differences in performance were observed among cells employing different electrolytes. Cells with LP57 have a capacity decay already after 40 cycles, while cells with FSI₁DFOB₀ and FSI₅DFOB₅ show a good cycling stability until the roll-over due to the dendrite growth under the applied 1C current rate.

To further confirm the roll-over failure in Graphite||Li cells, the cycled cells were disassembled and both graphite and Li metal were examined with SEM images. It is in general that no clear changes are observed on the graphite particles compared to the pristine graphite, except that the cells with FSI₅DFOB₅ show particle fracture (Figure S 20). Furthermore, The Li surface of cells with FSI₅DFOB₅

shows clear cracking and presence of HSAL (Figure S 20 h), These features are known to accelerate electrolyte decomposition and promote electrode degradation. EDX further confirmed the presence of substantial electrolyte decomposition products on both the graphite and lithium metal surfaces (Table S 3 and Table S 4). Based on these observations, it can be concluded that the roll-over failure is primarily caused by the formation of HSAL on the lithium metal surface. Overall, the FSI₅DFOB₅ electrolyte demonstrates favorable compatibility with graphite electrode materials.”

Figure S 19. (a) Specific discharge capacity vs. cycle number curves of NMC811||Li cells containing SUS316/316L parts with FSI₅DFOB₅ and LP57 electrolytes. (b) Specific discharge capacity vs. cycle number curves of Graphite||Li cells containing SUS316 parts with FSI₁DFOB₀, FSI₅DFOB₅ and LP57 electrolytes.

Figure S 20. SEM image of (a) pristine graphite and (e) pristine Li metal. The following images correspond to the graphite and Li metal after disassembling different cells at end of life (EOL): (b,f) cells with LP57; (c,g) cells with FSI₁DFOB₀; (d,h) cells with FSI₅DFOB₅.

Table S 3. Elemental distribution weight percentage on the surface of Gr collected by EDX.

	C (%)	O (%)	F (%)	N (%)	S (%)	B (%)	P (%)
LP57	57.21	25.65	12.53	0	0	0	4.6
FSI ₁ DFOB ₀	60.36	15.52	13.01	1.30	9.81	0	0
FSI ₅ DFOB ₅	34.50	30.70	12.50	1.50	6.00	14.8	0

Table S 4. Elemental distribution on the surface of Li collected by EDX.

	C (%)	O (%)	F (%)	N (%)	S (%)	B (%)	P (%)
LP57	16.50	39.20	32.90	0	0	0	11.40
FSI ₁ DFOB ₀	9.10	51.00	16.90	3.30	19.70	0	0
FSI ₅ DFOB ₅	12.09	65.63	9.19	2.45	6.19	4.4	0

8. Some Figure details need attention: Figure 5a, Figure 4a, Figure S9.

We thank the reviewer for pointing out the shortcomings in some figures. We have corrected Figure 4a, Figure 5a and Figure S9 (now Figure S11) accordingly.

Figure 4. (a) Linear sweep voltammogram of cells containing SUS316 as working electrode in boron-containing electrolytes. (b) Differential capacitance vs. potential curves for different electrolytes. The minimum value of the differential capacity curve corresponds to the zero-charge potential. (c) Calculated adsorption energies of LiCl, LiFSI and LiDFOB on (0001) Fe_2O_3 . SEM image of SUS316 surface after 1 month immersed in (d) LiCl (100ppm LiCl in H_2O), (e) LiCl+LiDFOB (100ppm LiCl + 0.5 M LiDFOB in H_2O) and (f) LiCl+LiFSI (100ppm LiCl + 0.5 M LiFSI in H_2O) solutions.

Figure 5. (a) LSVs of a 3-electrode DEMS cells using SUS316 as working electrode with different electrolytes at scan rate of 0.1 mV s^{-1} . Operando gaseous evolution of (b) H_2 , (c) CO_2 , (d) CO and (e) O_2 by DEMS are monitored during the LSV measurement, the gases are normalized with Ar to eliminate the fluctuation of carry gas.

Figure S 11. Selected Raman spectra of different conducting salt containing electrolytes (a) FSI₁DFOB₀, (b) FSI₅DFOB₅, (c) FSI₀DFOB₁ and (d) 100 ppm LiCl.

9. Please present the distinction of anodic dissolution in FSI_{0.5}DFOB_{0.5} between the cell of the positive case of SUS316L and the cell of Al-CVD on SUS316L. (Figure S13 FSI₅DFOB₅_Al and FSI₅DFOB₅_Al_316).

Following this comment from the reviewer, we have disassembled the cells after CA measurements and examined the positive cases of SUS316L (corresponding to FSI₅DFOB₅_Al_316) and Al-CVD on SUS316L (corresponding to FSI₅DFOB₅_Al). As shown in Figure S 15, no signs of corrosion/dissolution were observed on the surfaces of either sample. The surface morphologies remained unchanged compared to the pristine cases, as confirmed by SEM. This makes sense as the presence of LiDFOB could prevent both SUS and Al dissolution and no current increases could be observed during the CA measurements. In contrast, we observed the rise of anodic current when the FSI₁DFOB₀ electrolyte is used in cells investigating both Al dissolution and SUS induced Al dissolution. SEM images revealed clear evidence of Al dissolution on the surface of Al foil. Notably, the dissolution morphology differed when Al foil was in direct contact with the SUS spacer, indicating that the presence of SUS can accelerate Al dissolution.

The following text and figures have been included in the Supplementary Information:

“Modified coin cell setups using Al foil as the working electrode were used for these experiments, as shown in Figure S 1b,c. The variations of current density during the CA measurements are displayed for the FSI₁DFOB₀ and FSI₅DFOB₅ electrolytes, with and without SUS spacers (Figure S 15a,b). In cells using Al foil solely, the addition of LiDFOB notably inhibits the Al dissolution caused by the presence of LiFSI, as confirmed by the suppressed current density during CA measurements (Figure S 15a) and intact Al foil (Figure S 15h) as well as Al-CVD case (Figure S 15j) harvested after CA measurements. This is consistent with findings reported in the literature^[2-4].

Notably, when Al foil is placed on top of SUS316 spacer in coin cells, the cells containing FSI₁DFOB₀ electrolyte display considerably higher dissolution behavior compared to Al, as evidenced by the 30 times higher current density after 20 h (≈ 0.11 mAh cm⁻² for FSI₁DFOB₀_Al_316 vs. ≈ 0.004 mAh cm⁻² for FSI₁DFOB₀_Al) (Figure S 15b). This effect can be attributed to the formation of galvanic cells

between Al and SUS, which lowers the overpotential for the oxidation of Al and consequently enhances the dissolution of Al at high voltage^[5]. SEM images further support these findings, showing a rough and irregular surface (Figure S 15c), differs from the pits observed from the Al foil recovered from sample FSI₁DFOB₀_Al (Figure S 15g). This special morphology is similar to the stress-induced fracture observed on the uncoated electrode side recovered from cells assembled with SUS316 and containing FSI₁DFOB₀ (Figure S 17h). Similarly, when LiDFOB is included in the electrolyte (FSI₅DFOB₅), the coin cell containing Al foil on top of SUS316 spacer shows a very stable current density till the end of measurements, resulting in a smooth Al surface and intact SUS316 case as observed from the SEM images (Figure S 15d,f). This indicates that the inhibiting effects of LiDFOB on SUS dissolution also extend to suppressing Al dissolution in the presence of SUS spacers and coin cell cases.”

Figure S 15. Chronoamperograms of cells with (a) Al foil or (b) Al foil on top of SUS316 spacer as a working electrode with FSI₁DFOB₀ and FSI₅DFOB₅ electrolyte recorded at 4.2 V. SEM image of Al foil recovered from (c) FSI₁DFOB₀_Al_316 and (d) FSI₅DFOB₅_Al_316. SEM image of the inner surface of the SUS316L case from (e) pristine sample and (f) FSI₅DFOB₅_Al_316. SEM image of Al foil recovered from (g) FSI₁DFOB₀_Al and (h) FSI₅DFOB₅_Al. SEM image of the inner surface of the Al-CVD case from (i) pristine sample and (j) FSI₅DFOB₅_Al.

References:

- [18] J. Gorsch, J. Schneiders, M. Frieges, N. Kisseler, D. Klohs, H. Heimes, A. Kampker, M. Muñoz Castro, E. Siebecke, *Cell Rep. Phys. Sci.* 2025, 6, 102453.
- [19] P. Marcus, V. Maurice, H.-H. Strehblow, *Corrosion Science* 2008, 50, 2698–2704.
- [20] E. Yoon, J. Lee, S. Byun, D. Kim, T. Yoon, *Adv Funct Materials* 2022, 32, 2200026.
- [21] K. M. Scheer, M. Tulloch, I. Hamam, J. J. Abraham, M. B. Johnson, M. Metzger, *J. Electrochem. Soc.* 2025, 172, 010511.
- [22] C. Behling, J. Lüchtefeld, K. J. J. Mayrhofer, B. B. Berkes, *Electrochemistry Communications* 2024, 159, 107646.
- [23] M. Dahbi, F. Ghamouss, F. Tran-Van, D. Lemordant, M. Anouti, *Journal of Power Sources* 2011, 196, 9743–9750.
- [24] N. Meddings, M. Heinrich, F. Overney, J.-S. Lee, V. Ruiz, E. Napolitano, S. Seitz, G. Hinds, R. Raccichini, M. Gaberšček, J. Park, *J. Power Sources* 2020, 480, 228742.
- [25] G. Zhang, Y. Zhou, L. Wang, Y. Li, H. Xu, *Chem. Eng. J.* 2024, 494, 153236.
- [26] C. Fang, X. Wang, Y. S. Meng, *Trends Chem.* 2019, 1, 152–158.
- [27] R. C. McNulty, E. Hampson, L. N. Cutler, C. P. Grey, W. M. Dose, L. R. Johnson, *J. Mater. Chem.A* 2023, 11, 18302–18312.
- [28] A. Smith, P. Stüble, L. Leuthner, A. Hofmann, F. Jeschull, L. Mereacre, *Batter. Supercaps* 2023, 6, DOI 10.1002/batt.202300080.
- [29] Y. Zhang, W. Bao, E. Jeffs, B. Liu, B. Han, W. Mai, X. Li, W. Li, Y. Xu, B. Bhamwala, A. Liu, L. Ah, K. Ryu, Y. S. Meng, H. Gan, *ACS Energy Lett.* 2025, 10, 872–880.
- [30] Z. Liu, W. Hou, H. Tian, Q. Qiu, I. Ullah, S. Qiu, W. Sun, Q. Yu, J. Yuan, L. Xia, X. Wu, *Angew Chem Int Ed* 2024, 63, e202400110.
- [31] S.-J. Kang, K. Park, S.-H. Park, H. Lee, *Electrochim. Acta* 2018, 259, 949–954.

We trust that all suggestions and requested corrections have been fully addressed in the revised manuscript, and that the improved version will be considered for publication in the *Nature Communication* journal.

Thanking you in advance and looking forward to your reply at your earliest convenience,

Yours sincerely,

Corresponding author:

Isidora Cekic-Laskovic, PhD
Helmholtz Institute Muenster, IMD-4
Forschungszentrum Jülich GmbH
Corrensstraße 48
48149 Muenster, Germany
i.cekic-laskovic@fz-juelich.de
Tel.: +49 251 83-36805
Fax: +49 25183-30020

Reviewer #4 (Remarks to the Author):

In this manuscript, the authors' report approaches to mitigate Stainless Steel corrosion in LiFSI/LiDFBO-based electrolytes for lithium-ion battery applications. The study is interesting and includes many detailed characterizations and experiment design. However, I agree with Reviewer #2 and Reviewer #3 that the dissolution issues of stainless steel in LiFSI have been heavily investigated, and the major issue is to address the corrosion of the positive electrode current collectors (aluminum foils), considering the practical application of commercial and industry relevant cells. Accordingly, the novelty of the work is quite limited.

To respond the reviewer's concerns, we would like to highlight, as presented in our manuscript, that Al dissolution co-exists with SUS dissolution in LiFSI-based electrolyte at elevated voltage (≥ 4.2 V), as confirmed by the large crevices observed from both coated and uncoated side of Al foil (Figure S22b,h) together with the pits observed from the SUS316 spacers (Figure S22n). This explains the rise in anodic current observed in Figure 7a originated from both SUS and Al dissolution, which was found to be at the origin of cell failure. Since LiPF_6 is widely recognized as an effective Al dissolution inhibitor to protect the Al current collector in LiFSI and LiTFSI-based electrolytes, we then introduced LiPF_6 into LiFSI-based electrolyte (0.5M LiFSI + 0.5M LiPF_6 in EC:EMC 3:7 by wt%, noted as FSI_5PF_5) to passivate the Al current collector. As shown in Figure S21, a pronounced rise in anodic current was observed during the constant voltage step, while no evidence of Al dissolution was observed on the cathode Al current collector (Figure S22d,j). In contrast, clear pitting was still observed on the SUS spacer, which confirmed that SUS dissolution is the major issue in for cell failure in LiFSI-based electrolyte at elevated voltages, Furthermore, we evaluated the electrolytes at the pouch cells level (excluding SUS components), following reviewers' suggestion, and confirmed that SUS dissolution, rather than Al dissolution, is dominant failure in cells containing SUS components. A detailed discussion is provided as follows:

Following this comment from the reviewer, we evaluated the selected electrolytes in a pouch cell environment. We used the commercially available NMC811||Si-C (20% Si) dry two electrode pouch cells with a nominal capacity of 235 mAh from Li-FUN technology due to the identical cell chemistry and manufacture as the NMC811||Si-C (20% Si) coin cells used in this study. This ensures a meaningful and consistent comparison between cell formats.

Figure 7e shows a consistent trend of galvanostatic cycling performance in a pouch cell format compared to what we observed in NMC811||Si-C (20% Si) coin cell (Figure 7d). Pouch cells with $\text{FSI}_5\text{DFOB}_5$ deliver a stable CE and reached 88% capacity retention after 250 cycles, in contrast to the $<60\%$ capacity retention observed in pouch cells with LP57 after 250 cycles. The superior galvanostatic cycling performance in pouch cells suggested that the $\text{FSI}_5\text{DFOB}_5$ electrolytes shows also compatibility with nickel-rich NMC cathode and silicon containing anode. $\text{FSI}_1\text{DFOB}_0$ and FSI_5PF_5 (0.5M LiFSI+0.5M LiPF_6) electrolytes were also evaluated to access the impact of Al dissolution in pouch cell environment. Unlike the charge failure observed in coin cells using $\text{FSI}_1\text{DFOB}_0$ and FSI_5PF_5 (Figure S21), the pouch cells could be charged to the upper cutoff voltage. Although pronounced fluctuation in CE and discharge capacity were observed for cells with $\text{FSI}_1\text{DFOB}_0$ during initial cycles, induced by Al dissolution (as confirmed by increased current during constant voltage step in the 5th cycle shown in Figure 7e), the pouch cell can still be cycled.

This confirms that SUS dissolution, rather than Al dissolution, is dominant failure in cells containing SUS components.

The following text and figures have been included in the main text:

“

Figure 7. (a) Voltage vs. time profiles of coin cells with SUS316 parts using FS1₅DFOB₀ and FS1₅DFOB₅. (b) Specific discharge capacity vs. cycle number curves of coin cells containing SUS316 parts with FS1₅DFOB₅ and LP57 electrolytes. (c) Specific discharge capacity vs. cycle number curves of NMC811||Gr coin cells containing SUS316L parts with FS1₅DFOB₅ and LP57. (d) Specific discharge capacity vs. cycle number curves of NMC811||Si-C coin cells containing SUS316L parts with FS1₅DFOB₅ and LP57. (e) Specific discharge capacity vs. cycle number curves of NMC811||Si-C pouch cells with FS1₅DFOB₀, FS1₅DFOB₅, FS1₅PF₅ and LP57. (f) Current vs. time profile during the constant voltage step at 5th cycle for cells with considered electrolytes. Note: Replacing the coin cell components from SUS316 to SUS316L does not affect the galvanostatic cycling performance for both NMC811||Gr and NMC811||Si-C cells using LP57 electrolytes, as detailed in the Supplementary Note 6. A detailed coin cell and pouch cell configuration is shown in Figure S3.

Further evaluation is performed in the NMC811||Si-C pouch cell to exclude the impact from SUS components. Consistent with the results observed in coin cells, cells with FS1₅DFOB₅ exhibited a stable CE and reached 88 % capacity retention after 250 cycles, in contrast to the <60 % capacity retention observed in cells with LP57 after 250 cycles (Figure 7e). FS1₅DFOB₀ and FS1₅PF₅ (0.5M LiFSI+0.5M LiPF₆) are also evaluated to assess the impact of Al dissolution in pouch cell setup. A pronounced fluctuation in CE and discharge capacity is observed for cells with FS1₅DFOB₀ during initial cycles. The increased current during constant voltage step in the 5th cycle confirms the existence of Al dissolution, explaining this fluctuation (Figure 7f).

The proposed mechanism of action implies that both FSI⁻ and Cl⁻ anions are hindered by oxalate anions from reaching the surface of SUS. Further improvement has been achieved by incorporating more dissolution resistive SUS316L cell parts, resulting in ≈300 cycles until 80 % state of health for NMC811||Si-C cell chemistry and ≈1150 cycles for NMC811||Gr cell chemistry. Furthermore,

this improvement has also been confirmed in NMC811//Si-C pouch cell with 88 % capacity retention after 250 cycles.

Finally, the effectiveness of LiDFOB has been proven by improved galvanostatic cycling performance in NMC811//Gr and NMC811//Si-C coin cells and NMC811//Si-C pouch cells.

Further improvements are achieved by incorporating more dissolution-resistant SUS316L cell components, enabling ≈ 300 cycles for NMC811//Si-C cell chemistry and ≈ 1150 cycles for NMC811//Gr cell chemistries. This enhanced performance is also confirmed in NMC811//Si-C pouch cell with 88 % capacity retention after 250 cycles. This strategy not only extends cell lifetime but also emphasizes the importance of the electrochemical stability of the metal in the non-aqueous electrolytes for the design of practical high-energy LIBs.”

With regard to our work’s novelty, we believe that the significance of SUS dissolution will gain tremendous importance considering the fact that the tab-less design of the current jelly rolls cells such as 46XY uses stainless-steel components, that in some situations and depending on the cell chemistry can get in contact with the electrolyte, while the positive CAM is charging. Understanding and mitigating SUS dissolution is therefore not only a fundamental question but also a problem of growing practical relevance for next-generation high-energy cell designs. While SUS dissolution in LiFSI-based electrolytes has indeed been discussed previously, these studies have predominantly focused on fundamental observations. In our present work, we brought new insight into the role of Cl⁻ impurities in organic solvents in SUS dissolution and demonstrated that even trace amounts of Cl⁻ ions can initiate pitting corrosion and thereby accelerate the overall dissolution behavior in LiFSI-based electrolyte. More importantly, we proposed a new strategy to mitigate the SUS dissolution in the LiFSI-based electrolyte by including LiDFOB as a co-salt. Compared with the two commonly proposed approaches (surface coating on SUS and high concentration electrolytes (HCEs)), our strategy provides practical advantages.

Surface coating (e.g. Al-clad/Al-CVD^[1-3] or carbon-coating^[4]) can temporarily block the electrolyte from contacting the SUS surface, but once the micro-cracks form, or even the coating is not dense enough, the electrolyte would penetrate the coating layer and continue dissolve the SUS^[1,4]. It is well established that HCEs and localized high-concentration electrolytes (LHCEs) are highly effective in suppressing Al corrosion in LiFSI-based systems. However, this approach shows limited efficiencies toward SUS dissolution. Though Luo et al and Liu et al. proposed that HCE/LHCE can mitigate the SUS dissolution^[3,5], recent studies by Wang et al. and our group have independently confirmed that the LiFSI-based electrolyte still dissolve SUS at high concentrations after a delay time^[2,6]. Finally, the economic and environmental viability of these two methods remains a concern.

In contrast, the use of LiDFOB as a co-salt in LiFSI-based electrolyte can effectively mitigate the SUS dissolution without increasing the total salt concentration. With 0.5M LiDFOB and 0.5M LiFSI (total 1M), the NMC811//Gr cells containing SUS316L components reach over 1100 cycles at 80% state of health (Figure 7c), with no Al/SUS dissolution observed after post-mortem analysis at 1365th cycle (Figure S 22 f,l,r). In addition, the possible mechanism of LiDFOB in inhibiting SUS dissolution was also proposed. In general, our work is meaningful for practical application of commercial and industry relevant cells, especially for those cells employing SUS components to improve mechanical strength and safety (e.g. 46XY format cylindrical cells^[7]), leading to improved stability and performance of LIBs.

The following text and figures have been included in the main text and Supplementary Information:

“By effectively mitigating both SUS and Al dissolution through the strategic use of LiDFOB and LiFSI-based electrolytes, this study considerably enhanced electrochemical performance across different cell chemistries and cell levels. The optimized electrolytes are well-suited for practical high-energy LIBs configurations, including emerging large-format designs such as Tesla 46XY cylindrical cells, which utilize stainless steel components and commonly employ NMC811 cathodes paired with graphite anodes^[29,30]. The insights gained here directly contribute to industrial-scale battery advancements, leading to improved stability and performance for LIBs.”

Furthermore, the authors assigned corrosion issues merely owing to the presence of Cl⁻ anion impurities and ignored the main reasons of the solvent decomposition beyond 4.2 V vs. Li⁺/Li which release protons and initiate the corrosion of the current collectors (Ma, T., et al. (2017), J. of Physical Chemistry Letters, 8(5), 1072–1077; Nyholm, L., et al. (2023). Chemical Engineering Science, 282, 119346). More seriously, for the full cell results in LP57, the interpretations of the results are misleading.

We thank the reviewers for pointing out this important mechanism for Al dissolution that the proton release during solvent decomposition above 4.2V vs. Li | Li⁺ can initiate the dissolution of current collector. We fully agree that this factor plays a significant role in Al corrosion. However, its influence on SUS dissolution has remained unclear. To gain further insights into this issue, we have conducted additional electrochemical measurements using fluorinated carbonate solvents, which exhibit higher oxidative stability and lower tendency for proton release, as suggested by Ma *et al.*^[8]. We replaced the EC and EMC in the FSI₁DFOB₀ electrolyte with their fluorinated analogues of EC and EMC, i.e. FEC and FEMC in the same ratio (denoted as F_FSI₁DFOB₀), and conducted further CA and CV measurements to evaluate the influence of proton release on both Al and SUS dissolution.

We first evaluated the influence of proton release on Al dissolution. As shown in Figure S 18a, a lower current density is observed for cells containing F_FSI₁DFOB₀ during the CA measurements with Al foil as the working electrode, suggesting that the Al dissolution is reduced with the use of fluorinated solvents. However, the oxidative current of cells containing F_FSI₁DFOB₀ increases continuously over time but with a lesser intensity, indicating that the dissolution rate is reduced though not completely mitigated. This suggests that proton release is important but not the only factor governing Al dissolution. Further CV measurements revealed the voltage onsets of Al dissolution and also the passivation behaviors. Figure S 18b,c showed the cyclic voltammograms of cells containing FSI₁DFOB₀ and F_FSI₁DFOB₀. The shape of the first cycle differs from the subsequent ones and the anodic current is observed starting from 3 V vs. Li|Li⁺. Such an early onset has been attributed to the oxidation of carbonate solvents or the solvent impurity species (e.g. EG or MeOH), which can generate protons and destabilize the native Al₂O₃ layer^[8–10]. When EC/EMC were replaced by FEC/FEMC (Figure S18e), the onset of anodic current shifted ~0.1 V to higher voltage, indicating that fluorinated solvents with their higher oxidative stability indeed suppress proton release and delay Al₂O₃ layer breakdown. Interestingly, both LiFSI-based electrolytes showed a decrease in current during the reverse scan (4.3 V to 2.5 V vs. Li|Li⁺), and the current is further decreased during subsequent scans. This behavior contrasts with the observation of Scheer *et al.* and Nyholm *et al.*, where increased current during reverse scan and accelerated Al dissolution were reported^[9,10]. This different trend observed here is most likely linked to the low Cl⁻ impurity (7.49 ppm detected by IC measurement) in the LiFSI used in our study. Han *et al.* demonstrated

that the extra pure LiFSI (0.45 ppm Cl⁻) does not dissolve Al during CV scans between 3 – 5 V vs. Li|Li⁺, while Al dissolution is observed when 50 ppm Cl⁻ is added into electrolytes^[11]. Although the LiFSI used in our study is not as pure as in the publication of Han *et al.*, its low Cl⁻ content is sufficient to slow down the Al dissolution upon consecutive scans. The occurrence of Al dissolution was evidenced for the cells with FSI₁DFOB₀ (Figure S18g) by the clear pits and deposits observed on the Al foil after 3 CV scans. However, less pits were found on Al foils in cells with F_FSI₁DFOB₀ (Figure S18h), observations that are consistent with the reduced proton generation and reduced Al dissolution in fluorinated solvents.

We then focused on whether suppressing proton release would also affect SUS dissolution. In strong contrast to the behavior observed by Al, the cells containing F_FSI₁DFOB₀ exhibited a substantially higher current density than those with non-fluorinated FSI₁DFOB₀ during CA measurements (Figure S19a). These results imply that increasing the oxidative stability of the solvent and thereby reducing proton release does not mitigate SUS dissolution. Instead, the dissolution is even accelerated in the fluorinated solvent system. Additional CV measurements further support this conclusion. For both LiFSI-based electrolytes, their CV curves displayed the typical dissolution behavior, characterized with a pronounced increase in current during the reverse scan, forming a hysteresis loop (Figure S19b,c). Notably, in the first cycle, cells with F_FSI₁DFOB₀ has an earlier voltage onset for SUS dissolution and reached a much higher anodic current maximum compared to the cells with FSI₁DFOB₀ (Figure S19e). When the scan is reversed, cells with F_FSI₁DFOB₀ still exhibits relatively large currents over a wide potential window and shows a pronounced hysteresis between forward and reverse scans, characteristic of a strongly activated dissolution process. It was also observed that the intensity of the anodic current is increasing with each subsequent cycle, thus indicating an intensification of the SUS dissolution process for cells with F_FSI₁DFOB₀ (Figure S19f). These electrochemical results are consistent with the post-mortem surface analysis, SEM images of SUS316 after CV scans showed direct evidence of pitting holes in the presence of both electrolytes, however, the sample with F_FSI₁DFOB₀ exhibited denser and larger pits, indicating a more severe dissolution process (Figure S19g,h). These results suggested that the dissolution mechanism in SUS is different from Al and is governed by the FSI⁻ (and to some extent Cl⁻ impurities), rather than by proton release associated with solvent oxidation.

As for the reason behind the more severe dissolution in fluorinated solvents, we can at this stage only offer a hypothesis that will request further investigation. The dissolution of SUS in LiFSI-based electrolyte involves concomitant factors, in which an important factor is the formation of soluble Fe-FSI complexes^[5]. One strategy proposed in the literature to mitigate this process is the use of high concentration electrolyte, which eliminates the free solvent and thereby reduce the solubility of the Fe-FSI complexes^[3,5]. Though its functionality is still under debate, this might give a hint for interpreting our observations. It is well known that fluorinated solvents have a lower donor number which is less strongly coordinated by solvent molecules and contribution of FSI⁻ to the primary solvation sheath is significantly increased^[12]. This anion-rich solvation environment enhances the local activity of FSI⁻ at the SUS surface and may facilitates the formation of Fe-FSI complexes. In such a solvent environment, the dissolution of Fe is kinetically more favorable.

At this stage, we can conclude that fluorinated solvent slightly mitigates Al dissolution but significantly accelerates SUS dissolution, owing to the fundamentally different dissolution mechanism of Al and SUS in LiFSI-based electrolyte. Additional experiments have shown that, besides SUS-dissolution, electrolyte decomposition can also trigger Al dissolution. However, having two side reactions occurring simultaneously, it is hard to deconvolute the intensity of each

process on the observed anodic current when the potential is hold at 4.2V. Nevertheless, our experiments have shown that the SUS dissolution is more pronounced and thus supersedes the occurring Al-dissolution. It is therefore worth to evaluate the complex dissolution behavior when both Al and SUS are presented. Figure S20a presented the CA measurements using Al on top of SUS as working electrode. Although the F_FSI₁DFOB₀ can mitigate the Al dissolution, as shown in Figure S19a, it accelerates the dissolution process when both Al and SUS are presented. Further CV measurements shown a similar trend compare to the cells with sole SUS as working electrode. Post-mortem SEM after CV scans reveals larger deposits on Al and larger pits on SUS for cells with F_FSI₁DFOB₀ compared to non-fluorinated samples (Figure S20g,h,j,k). These observations confirm that SUS dissolution has a more severe problem and can dominate the dissolution in the cell. As a result, improving solvent oxidative stability slightly mitigates Al dissolution but has a detrimental effect on SUS stability when fluorinated solvents are employed. Last but not least, we also observed the corrosion of SUS in LiFSI electrolyte and protection of SUS in LiDFOB added electrolyte during storage experiments, as shown in Figure 4 and Figure S12, confirming that the major reason is not the solvent decomposition at elevated potentials.

The following text and figures have been included in the main text and Supplementary Information:

*“Notably, a pronounced current increase is observed at ≈ 3.5 V in the cell containing LiCl_{sat} electrolyte, which can be attributed to the SUS dissolution triggered either by solvent decomposition or by the presence of Cl⁻ anion. Organic carbonate-based solvents as well as the solvent impurity species (e.g. EG or MeOH) may decompose above 3 V vs. Li/Li⁺ and generate the protons, which destabilize the native Al₂O₃ layer and initiate the Al dissolution^[36–38]. However, the use of fluorinated carbonate solvents, which have higher oxidation stability and lower tendency for proton release, has confirmed that the proton does not contribute to the SUS dissolution. The detailed discussion is provided in **Supplementary Note 3**.*

Supplementary Note 3: Investigation of the Impact of Solvent Decomposition on Metal Dissolution

*During the LSV measurements, a pronounced current increase is observed at ≈ 3.5 V in the cell containing LiCl_{sat} electrolyte (**Figure 1a**). This rise in current indicates the destabilization of native oxide layer on SUS and the onset of SUS dissolution. Such a current increase can be triggered either by solvent decomposition or by the presence of Cl⁻ anion. Proton released from the solvent oxidation can initiate Al dissolution at a similar voltage onset^[6], which could be one of the reasons for SUS dissolution observed here. To validate this hypothesis, CA and CV measurements were conducted using fluorinated carbonate solvents, which possess higher oxidative stability and lower tendency for proton release^[7]. In these experiments, EC and EMC in the FSI₁DFOB₀ electrolyte were replaced by their fluorinated analogues, i.e. FEC and FEMC, in the same ratio (denoted as F_FSI₁DFOB₀).*

*First, the Al foil was used as working electrode to confirm prior literature evidence that proton release contributes to Al dissolution. For this purpose, a cell equipped with an Al-CVD case, an Al spacer, and an Al working electrode was assembled (**Figure S 1b**). As shown in **Figure S 18a**, a lower current density is observed for cells containing F_FSI₁DFOB₀ during the CA measurements with Al foil as the working electrode, suggesting that the Al dissolution is reduced in the presence of fluorinated solvents. However, the oxidative current in cells containing F_FSI₁DFOB₀ increases continuously over time, indicating that while the dissolution rate is reduced but still not completely*

mitigated. This suggests that proton release is important but not the only factor governing Al dissolution. Further CV measurements revealed the voltage onsets of Al dissolution and also the passivation behaviors. **Figure S 18b,c** showed the cyclic voltammograms of cells containing FSI₁DFOB₀ and F_FSI₁DFOB₀. The shape of the first cycle differs from the subsequent ones and the anodic current is observed starting from 3 V vs. Li/Li⁺. Such an early onset has been attributed to the oxidation of organic carbonate-based solvents or the solvent impurity species (e.g. EG or MeOH), which can generate protons and destabilize the native Al₂O₃ layer on the surface of the Al current collector^[6-8]. When EC/EMC mixture is replaced by FEC/FEMC (**Figure S 18e**), the onset of anodic current shifted ~0.1 V to higher voltage, indicating that fluorinated solvents with their higher oxidative stability indeed suppress proton release and delay Al₂O₃ layer breakdown. Interestingly, both LiFSI-based electrolytes showed a decrease in current during the reverse scan (4.3 V to 2.5 V vs. Li/Li⁺), and the current is further decreased in the subsequent scans. This trend resembles the behavior observed in cells containing FSI₅DFOB₅, with a slightly higher current density maximum, indicating that the Al foil is gradually passivated during the cycles (**Figure S 18d**). This behavior contrasts with the observation of Scheer et al. and Nyholm et al., where increased current during reverse scan and accelerated Al dissolution were reported^[6,8]. This different trend observed here is most likely linked to the low Cl⁻ anion impurity (7.49 ppm detected by IC measurement) in the LiFSI used in our study. Han et al. demonstrated that the extra pure LiFSI (0.45 ppm Cl⁻) does not dissolve Al during CV scans between 3 – 5 V vs. Li/Li⁺, while Al dissolution is observed when 50 ppm Cl⁻ is added into electrolytes^[9]. Although the LiFSI used in our study is not as pure as in the publication of Han et al., its low Cl⁻ content is sufficient to slow down the Al dissolution upon consecutive scans. The occurrence of Al dissolution was evidenced for the cells with FSI₁DFOB₀ by the clear pits and deposits observed on the Al foil after three CV scans (**Figure S 18g**), in contrast with pits-free surface observed on Al foils collected from cells with LiDFOB containing electrolyte (FSI₅DFOB₅) (**Figure S 18i**). Notably, less pits were found on Al foils in cells with F_FSI₁DFOB₀ (**Figure S 18h**), consistent with the reduced proton generation and delayed Al dissolution in fluorinated solvents

Figure S 18. (a) Chronoamperograms of cells with Al foil as a working electrode with FSI₁DFOB₀ and F_FSI₁DFOB₀ electrolyte recorded at 4.2 V. Cyclic voltammograms of cells containing Al as working electrodes with (b) FSI₁DFOB₀ and (c) F_FSI₁DFOB₀ and (d) FSI₅DFOB₅ electrolyte. Comparison of the (e) first cycle and (f) third cycle of cyclic voltammetry for cells with considered electrolytes. SEM image of Al foil recovered from (g) FSI₁DFOB₀ (h) F_FSI₁DFOB₀ and (i) FSI₅DFOB₅ after three CV cycles.

After confirming the positive effect of fluorinated solvents in mitigating the Al dissolution, we then examined whether suppressing proton release would also affect the SUS dissolution. In strong contrast to the behavior observed for Al, the cells containing F_FSI₁DFOB₀ exhibited a substantially higher current density than those with non-fluorinated FSI₁DFOB₀ during CA measurements (Figure S 19a). This clearly indicates that increasing the oxidative stability of the solvent and thereby reducing proton release does not mitigate SUS dissolution. Instead, the dissolution is even accelerated in the fluorinated solvent system. Additional CV measurements further support this conclusion. For both LiFSI-based electrolytes, recorded CV curves displayed the typical dissolution behavior, characterized with a pronounced increase in current during the reverse scan, forming a hysteresis loop (Figure S 19b,c). While cells with LiDFOB containing electrolyte FSI₅DFOB₅ showed suppressed current density across CV scan range (Figure S 19d). Notably, in the first cycle, cells with F_FSI₁DFOB₀ show an earlier voltage onset for SUS dissolution and reached a much higher anodic current maximum compared to the cells with FSI₁DFOB₀ (Figure S 19e). When the scan is reversed, cells with F_FSI₁DFOB₀ still exhibit relatively large currents over a wide potential window and show a pronounced hysteresis between

forward and reverse scans, characteristic of a strongly activated dissolution process. Subsequent scans further confirmed a higher anodic current maximum for cells with $F_FSI_1DFOB_0$ (Figure S 19f). These electrochemical results are consistent with the post-mortem surface analysis. SEM images of SUS316 after CV cycles provided direct evidence of pitting in the presence of both LiFSI-based electrolytes (Figure S 19g,h), whereas a clean and pits-free surface is observed on SUS spacer when LiDFOB is present (Figure S 19i). However, the sample using $F_FSI_1DFOB_0$ exhibited larger and more densely distributed pits, indicating a more severe dissolution process (Figure S 19h). These results suggested that the dissolution mechanism in SUS is different from Al and is governed by the FSI- (and to some extent Cl^- impurities), rather than by proton release associated with solvent oxidation.

Figure S 19. (a) Chronoamperograms of cells with SUS316 spacer as a working electrode with FSI_1DFOB_0 and $F_FSI_1DFOB_0$ electrolyte recorded at 4.2 V. Cyclic voltammograms of cells containing SUS316 spacer as working electrodes with (b) FSI_1DFOB_0 and (c) $F_FSI_1DFOB_0$ and (d) FSI_5DFOB_5 electrolyte. Comparison of the (e) first cycle and (f) third cycle of cyclic voltammetry for cells with considered electrolytes. SEM image of SUS316 spacer recovered from (g) FSI_1DFOB_0 (h) $F_FSI_1DFOB_0$ and (i) FSI_5DFOB_5 after three CV cycles.

At this stage, we can conclude that fluorinated solvent slightly mitigates Al dissolution but significantly accelerate SUS dissolution, owing to the fundamentally different dissolution mechanism of Al and SUS in LiFSI-based electrolyte. It is therefore worth to evaluate the complex dissolution behavior when both Al and SUS are presented. Figure S 20a shows the CA

measurements using Al on top of SUS as working electrode. Although the $F_FSI_1DFOB_0$ can mitigate the Al dissolution, as shown in **Figure S 19a**, it accelerates the dissolution process when both Al and SUS are presented. Additional CV measurements show a similar trend compared to the cells with sole SUS as working electrode (**Figure S 19b-f**). Post-mortem SEM after CV scans reveals larger deposits on Al and larger pits on SUS for cells with $F_FSI_1DFOB_0$ compared to non-fluorinated samples (**Figure S 19g,h,j,k**). These observations confirm that SUS dissolution has a more severe problem and can dominate the dissolution in the cell. As a result, improving solvent oxidative stability slightly mitigates Al dissolution but has little effect, or even has detrimental effect on SUS stability when fluorinated solvents are employed."

Figure S 20. (a) Chronoamperograms of cells with Al foil on top of SUS316 space as a working electrode with FSI_1DFOB_0 and $F_FSI_1DFOB_0$ electrolyte recorded at 4.2 V. Cyclic voltammograms of cells

containing Al foil on top of SUS316 spacer as working electrodes with (b) FSI₁DFOB₀ and (c) F₋FSI₁DFOB₀ and (d) FSI₅DFOB₅ electrolyte. Comparison of the (e) first cycle and (f) third cycle in the cyclic voltammetry measurement for cells with considered electrolytes. SEM image of Al foil recovered from (g) FSI₁DFOB₀ (h) F₋FSI₁DFOB₀ and (i) FSI₅DFOB₅ after three CV cycles. SEM image of SUS316 spacer recovered from (j) FSI₁DFOB₀ (k) F₋FSI₁DFOB₀ and (l) FSI₅DFOB₅ after three CV cycles.

The authors are advised to address the following comments before new submissions:

1. As a general comment, the authors need to clearly refer to which cell design was used for each experiment, this among the blurry things in the manuscript. The Figs. on the different cell design should be placed in the main manuscript rather than the SI.

We thank the reviewer for pointing out the lack of clarity regarding the different cell designs used in this study. We fully agree that a clear distinction between cell configurations is essential for interpreting the electrochemical results and it will help the improvement of clarity.

In the revised manuscript, we have described the cell design used for each experiment in the figure captions and also the corresponding experimental descriptions as follows:

“**Figure 1.** (a) Linear sweep voltammograms of cells containing SUS316 as working electrodes with FSI₁DFOB₀, LiCl_{sat} and FSI₁DFOB₀ + LiCl_{sat} electrolyte. (b) Chronoamperograms of cells containing FSI₁DFOB₀ and LiCl_{sat} electrolytes were recorded at a voltage of 4.2 V for 20 h. SEM and EDX images of the polished SUS316 spacer harvested from cells after 20 h of chronoamperometry measurements with (c) LiCl_{sat} and (d) FSI₁DFOB₀ electrolytes. Cyclic voltammograms of cells containing SUS316 as working electrode with (e) FSI₁DFOB₀ and (f) FSI₁DFOB₀ + LiCl_{sat} electrolyte. (g) Comparison of the first cycle in the cyclic voltammogram for FSI₁DFOB₀ and FSI₁DFOB₀ + LiCl_{sat} electrolyte. A dedicated coin cell design was used for LSV and CA measurements while a PAT cell design was used for CV measurements throughout this study, as shown in **Figure S1** and **Figure S2**.”

Figure 7. (a) Voltage vs. time profiles of **coin cells** with SUS316 parts using FSI₁DFOB₀ and FSI₅DFOB₅. (b) Specific discharge capacity vs. cycle number curves of **coin cells** containing SUS316 parts with FSI₅DFOB₅ and LP57 electrolytes. (c) Specific discharge capacity vs. cycle number curves of NMC811||Gr **coin cells** containing SUS316L parts with FSI₅DFOB₅ and LP57. (d) Specific discharge capacity vs. cycle number curves of NMC811||Si-C **coin cells** containing SUS316L parts with FSI₅DFOB₅ and LP57. (e) Specific discharge capacity vs. cycle number curves of NMC811||Si-C pouch cells with FSI₁DFOB₀, FSI₅DFOB₅, FSI₅PF₅ and LP57. (f) Current vs. time profile during the constant voltage step at 5th cycle for cells with considered electrolytes. Note: Replacing the coin cell components from SUS316 to SUS316L does not affect the galvanostatic cycling performance for both NMC811||Gr and NMC811||Si-C cells using LP57 electrolytes, as detailed in the Supplementary Note 6. A detailed coin cell and pouch cell configuration is shown in **Figure S3**.”

Electrodes and cell assembly

Two types of coin cell parts with case and spacer made of SUS316 and SUS316L were purchased from Xiamen TOB New Energy Technology Co. Ltd and Seika Sangyo GmbH, respectively.

Throughout this manuscript, we follow battery conventions and use “anode” and “cathode”, which correspond to the “negative electrode” and “positive electrode”, respectively. CR2032 Type coin cells with different SUS grades (SUS316 and SUS316L) were assembled in a two-electrode configuration as shown in **Figure S3**. For NMC811//Gr cell chemistry, calendared NMC811 single-sided coated cathode sheets with an areal capacity of $1.0 \text{ mAh}\cdot\text{cm}^{-2}$ were provided by Umicore. Calendared graphite single-sided coated electrode sheets with an areal capacity of $1.14 \text{ mAh}\cdot\text{cm}^{-2}$ were provided by CIDETEC, corresponding to an N/P ratio of 1.14. For NMC811//Si-C (20% Si) cell chemistry, calendared single-side coated cathode (with an areal capacity of $2.73 \text{ mAh}\cdot\text{cm}^{-2}$) and anode (with an areal capacity of $3.5 \text{ mAh}\cdot\text{cm}^{-2}$) sheets were purchased from Li-FUN technology, corresponding to an N/P ratio of 1.28. The electrodes were punched into $\phi 14 \text{ mm}$ and $\phi 15 \text{ mm}$ disks for the cathode and the anode, respectively. The electrodes were dried at $120 \text{ }^\circ\text{C}$ under vacuum ($<10^{-2} \text{ mbar}$) for 12 h and stored in the argon-filled glovebox before cell assembly. Celgard 2500 membrane was used as a separator. Li metal ($500 \text{ }\mu\text{m}$ thickness) purchased from China Energy Lithium CO. Ltd was punched into $\phi 15 \text{ mm}$ disks and kept inside the argon filled glovebox as well.

Commercially available NMC811//Si-C (20% Si) dry two electrode pouch cells with a nominal capacity of 235 mAh from Li-FUN technology were used to evaluate the electrolytes at pouch cell level. The cells were cut open and dried at $90 \text{ }^\circ\text{C}$ for 12 h under vacuum ($<10^{-2} \text{ mbar}$) before use. Then, the cells were filled with $700 \text{ }\mu\text{L}$ of electrolytes and sealed at the upper end of the gas pocket ($165 \text{ }^\circ\text{C}$, 5 s) using a pouch cell sealer (GN-HS350V, Gelon LIB Co) in the argon-filled glovebox. A specially developed holder was used for the cells with a constant pressure of $\approx 2 \text{ bar}$, as shown in **Figure S3**.

Metal dissolution study

A special coin cell format was used to study the metal dissolution, with Al foil or polished SUS316 spacer as the working electrode and Li foil as the counter electrode. Linear sweep voltammetry (LSV) measurements were conducted from 0 to 5 V at a scan rate of 2 mV s^{-1} whereas the potentiostatic chronoamperometry (CA), followed by LSV to 4.2V and holding the voltage for 20 h, was conducted on a VMP3 workstation (BioLogic). The detailed configurations of coin cells used for LSV and CA are illustrated in **Figure S1**.

Cyclic Voltammetry (CV) measurements were conducted using PAT cell (EL-CELL) in a three-electrode fashion, with Al foil or polished SUS316 spacer as the working electrode, Li ring as reference electrode and Li foil as the counter electrode. Lower plunger with different metal materials (Al or SUS) was used for the Al and SUS dissolution study and the detailed configuration is shown in **Figure S2**.

Regarding the placement of the cell design illustrations, we appreciate the reviewer’s suggestion to include all figures in the main text. However, to maintain readability and avoid overloading the main manuscript with technical schematics, we decided to keep the detailed cell design figures in the Supporting Information. At the same time, we significantly improved the organization and clarity of these schematics as follows:

“

Figure S 1. Illustration of the coin cell setup for (a) SUS dissolution, (b) Al dissolution and (c) SUS induced Al dissolution studies.

Figure S 2. Illustration of the PAT cell setup from EL-CELL GmbH for (a) SUS dissolution, (b) Al dissolution and (c) SUS induced Al dissolution studies.

Figure S3. Illustration of the coin cell setup for galvanostatic cycling stability test with (a) SUS316 components and (b) SUS316L components. (c) Illustration of Li-FUN pouch cells. (d) Representation of how the cells are securely fixed to custom-made cell holder.”

2. What is the level of impurities in the LiFSI, LiDFOB and LiPF₆ salts? Previous reports (e.g. Nyholm, L., et al. (2023). *Chemical Engineering Science*, 282, 119346) showed that the presence of high amount of fluoride anion impurity, which can play a crucial role in passivation. This can be also one of the reasons for the observed passivation in LiDFOB solutions.

We thank the reviewer for raising the important point regarding the impurities in the LiFSI, LiDFOB and LiPF₆ salts. Based on the certificate of analysis provided by the supplier of these chemicals (i.e. E-Lyte Innovations GmbH), the amount of impurities is quite low, as these chemicals are listed as battery grade. Nevertheless, their amount and nature are listed below:

	Lithium hexafluorophosphate (LiPF ₆)	Lithium bis(fluorosulfonyl)imide (LiFSI)	Lithium difluoro(oxalato)borate (LiDFOB)
Assay	>99.9 %	>99.0 %	>99.3 %
Water	X (<10 ppm*)	< 100 ppm (<10 ppm*)	< 100 ppm (<10 ppm*)
Free acid as HF	< 50 ppm	< 20 ppm	< 20 ppm
Chloride	< 2 ppm	< 20 ppm (7.49 ppm**)	< 10 ppm

* measured by KF

** measured by IC

Analysis of the salts shows that all purities are above 99 %, which can be acceptable for battery grade materials. And our Karl-Fischer measurements also confirm a very low water content of <10 ppm. In addition, the free acid (as HF), which can be considered as the source for fluoride anion impurities, remains at an acceptable range for all three salts.

Furthermore, we believe that the fluoride anion impurities are unlikely to passivate the SUS surface. PF_6^- anion which can easily be decomposed and release fluoride species, can serve as a representative source of such fluoride anions. While the released fluoride can form an AlF_3 -rich layer that effectively suppresses Al dissolution, this passivation mechanism does not extend to SUS. As shown in Figure S 21, the voltage vs. time profiles of cells containing FSI₅PF₅ (0.5M LiFSI + 0.5M LiPF₆) still exhibit a pronounced rise in anodic current during the constant voltage step. Furthermore, SEM images confirmed that no evidence of Al dissolution was observed on the cathode current collector (Figure S 22d,j), while clear pitting was still observed on the SUS spacer (Figure S 22p). In addition, the LSV results with different borate salts (LiBOB and LiBF₄) have also confirmed that the oxalate moiety in the DFOB⁻ anion is critical in mitigating SUS dissolution. In summary, we believe that the impurities do not contribute to the inhibition of SUS dissolution.

3. The passivation mechanism should be revised according after considering the decomposition of the carbonate solvent and the presence of other impurities in the salt.

Following reviewer's suggestions, we have conducted additional experiments that confirmed that the decomposition of organic carbonate-based solvents and the associated proton release have detrimental effect on Al dissolution, however, these processes does not influence the SUS dissolution. We also verified that the lithium salts used in this study is pure enough and the possible fluoride anions are not able to prevent SUS dissolution in LiFSI-based electrolytes. Taken together, these results reinforce the validity of the dissolution mechanism we propose. We sincerely thank the reviewer for the constructive comments, which have helped us refine the manuscript and strengthen the conclusions.

The following paragraph has been included in the main text:

“ Proposed mechanism of SUS dissolution and inhibitive effect of LiDFOB

Comprehensive experimental and theoretical evaluation enabled discussion regarding the possible mechanism for SUS dissolution and LiDFOB inhibition. Electrochemical experiments with fluorinated solvents have confirmed that the released proton during solvent oxidation does not contribute to the SUS dissolution. Instead, the observed dissolution is predominantly associated with the presence of Cl⁻ and FSI⁻ anions. Regarding the inhibitive effect of LiDFOB, LiPF₆ as a representative source for fluoride anion, has been confirmed that fluoride does not impact the SUS dissolution. Furthermore, although boron-containing surface species were detected, their presence however does not provide protection for SUS against dissolution, and the competing anion adsorption by DFOB⁻ anions is the most plausible explanation for the suppression effect. A detailed discussion of the proposed mechanism is provided below.”

4. The corrosion behavior for both current collectors and stainless steel should be examined using cyclic voltammetry as well. This will clearly illustrate the possible passivation behavior.

Following reviewer's suggestions, we have conducted additional CV measurements in a scan range from 2.5 to 4.3 V vs. Li|Li⁺ using three-electrode PAT cell. The reason for choosing this configuration is that CV analysis requires an accurate and stable control of the potential. In addition, we could also change SUS plunger to Al plunger to evaluate the Al dissolution in this PAT cell configuration, as shown in Figure S2.

To examine the impact of FSI⁻ and Cl⁻ anions on SUS dissolution. Cyclic voltammetry performed at a slow scan rate of 0.1 mV⁻¹ were conducted using SUS as working electrode in FSI₁DFOB₀ and FSI₁DFOB₀+LiCl_{sat} electrolytes. Cells with both FSI₁DFOB₀ and FSI₁DFOB₀+LiCl_{sat} electrolytes exhibit typical dissolution behavior, characterized with a pronounced increase in current during the reverse scan, forming a hysteresis loop (Figure 1e,f). By comparing the first CV cycle, cell with FSI₁DFOB₀ shows the onset of anodic current at ≈4.2 V, whereas cell with FSI₁DFOB₀+LiCl_{sat} shows two distinct anodic current rises at ≈3.5 V and ≈4.2 V (Figure 1g). These features are consistent with the trends observed in the corresponding LSV measurements (Figure 1a).

The following paragraph and figures have been included in the main text:

“Figure 1. (a) Linear sweep voltammograms of cells containing SUS316 as working electrodes with FSI₁DFOB₀, LiCl_{sat} and FSI₁DFOB₀ + LiCl_{sat} electrolyte. (b) Chronoamperograms of cells containing FSI₁DFOB₀ and LiCl_{sat} electrolytes were recorded at a voltage of 4.2 V for 20 h. SEM and EDX images of the polished SUS316 spacer harvested from cells after 20 h of chronoamperometry measurements with (c)

LiCl_{sat} and (d) FSI₁DFOB₀ electrolytes. Cyclic voltammograms of cells containing SUS316 as working electrode with (e) FSI₁DFOB₀ and (f) FSI₁DFOB₀ + LiCl_{sat} electrolyte. (g) Comparison of the first cycle in the cyclic voltammogram for FSI₁DFOB₀ and FSI₁DFOB₀ + LiCl_{sat} electrolyte. A dedicated coin cell design was used for LSV and CA measurements while a PAT cell design was used for CV measurements throughout this study, as shown in Figure S1 and Figure S2.

To further validate the distinct dissolution process, LSV was carried out using a mixed electrolyte comprising LiCl_{sat} and FSI₁DFOB₀. These results clearly show a two-stage dissolution process, confirming the separate contribution of Cl⁻ and FSI⁻ anions to the SUS dissolution process. Cyclic voltammetry (CV) measurements, performed at a slow scan rate of 0.1 mV⁻¹ confirm the distinct dissolution process arising from Cl⁻ and FSI⁻ anions. Cells with both FSI₁DFOB₀ and FSI₁DFOB₀+LiCl_{sat} electrolytes exhibit typical dissolution behavior, characterized by a pronounced increase in current during the reverse scan, forming a hysteresis loop (Figure 1e,f). By comparing the first CV cycle, cell with FSI₁DFOB₀ shows the onset of anodic current at ≈4.2 V, whereas cell with FSI₁DFOB₀+LiCl_{sat} shows two distinct anodic current rises at ≈3.5 V and ≈4.2 V (Figure 1g). These features are consistent with the trends observed in the corresponding LSV measurements.”

In addition, the inhibitive effect of LiDFOB on SUS dissolution was examined by CV measurements. With the addition of LiDFOB (FSI₅DFOB₅ and FSI₀DFOB₁+LiCl_{sat}), both Cl⁻ and FSI⁻ induced SUS dissolution (characterized as current rise at ≈3.5 V and ≈4.2 V, respectively) were suppressed with a current density maximum <0.001 mA cm⁻² (0.025 mA cm⁻²@FSI₁DFOB₀ and 0.08 mA cm⁻² @FSI₁DFOB₀+LiCl_{sat}), as shown in Figure 3e,f.

The following text and figures have been included in the main text:

“

Figure 3. (a) Linear sweep voltammograms of cells containing SUS316 as working electrode and LiDFOB containing electrolytes and (b) corresponding chronoamperograms recorded at 4.2 V for 20 h. (c) Linear sweep voltammograms of cells containing SUS316 as working electrode in a mixture of LiCl_{sat} and $\text{FSI}_0\text{DFOB}_1$ electrolytes and (d) corresponding chronoamperograms recorded at 4.2 V for 20 h. Cyclic voltammograms of cells containing SUS316 as working electrode with (e) $\text{FSI}_5\text{DFOB}_5$ and (f) $\text{FSI}_0\text{DFOB}_1 + \text{LiCl}_{\text{sat}}$ electrolyte. (g-l) Selected XPS spectra of the harvested SUS316 spacers from the cell after 20 h at 4.2 V with $\text{FSI}_1\text{DFOB}_0$ (top), $\text{FSI}_5\text{DFOB}_5$ (middle) and $\text{FSI}_0\text{DFOB}_1$ (bottom) electrolytes.

Although a slightly higher current density for the primary and secondary passivation is observed, the critical voltage of Cl^- induced dissolution was considerably suppressed and shifted from 3.5 V to 4.2 V in the $\text{FSI}_0\text{DFOB}_1 + \text{LiCl}_{\text{sat}}$ electrolyte. A similar trend is also observed in cyclic voltammograms with a slower scan rate. No anodic current is observed for cells with $\text{FSI}_5\text{DFOB}_5$ (Figure 3e) and only a slightly anodic current rise at ≈ 4.2 V for cells with $\text{FSI}_0\text{DFOB}_1 + \text{LiCl}_{\text{sat}}$ (Figure 3f), with a current density maximum of 0.001 mA cm^{-2} . This value is significantly lower than the pronounced current density maximum of 0.08 mA cm^{-2} observed in cell containing $\text{FSI}_1\text{DFOB}_0 + \text{LiCl}_{\text{sat}}$. The stable current density at 4.2 V and the strongly suppressed current density during LSV and CV measurements, together with the pitting-free surface of the SUS316 spacer after CA measurement, collectively confirm the inhibitive effect of LiDFOB in suppressing both Cl^- and FSI⁻ anion induced SUS dissolution.”

Furthermore, we have also repeated CV measurements to investigate the Al dissolution, similar inhibiting effect of LiDFOB on Al dissolution, even in the presence of SUS, can be observed. The detailed discussion described in Supplementary Note 3.

5. The corrosion behavior of the current collector and full cell should be performed in pouch cells and then compared to the results in stainless steel, so the authors can differentiate between the current collector corrosion and the stainless steel one.

Following this comment from the reviewer, we evaluated the selected electrolytes in a pouch cell setup. We used the commercially available NMC811||Si-C (20% Si) dry two electrode pouch cells with a nominal capacity of 235 mAh from Li-FUN technology due to the identical cell chemistry and manufacture as the NMC811||Si-C (20% Si) coin cells used in this study. This ensures a meaningful and consistent comparison between cell formats.

Figure 7e shows a consistent trend of galvanostatic cycling performance in a pouch cell format compared to what we observed in NMC811||Si-C (20% Si) coin cell (Figure 7d). Pouch cells with FSI₅DFOB₅ deliver a stable CE and reached 88 % capacity retention after 250 cycles, in contrast to the <60 % capacity retention observed in pouch cells with LP57 after 250 cycles. The superior galvanostatic cycling performance in pouch cells suggested that the FSI₅DFOB₅ electrolytes shows also compatibility with nickel-rich NMC cathode and silicon containing anode. FSI₁DFOB₀ and FSI₅PF₅ (0.5M LiFSI+0.5M LiPF₆) electrolytes were also evaluated to assess the impact of Al dissolution in pouch cell environment. Unlike the charge failure observed in coin cells using FSI₁DFOB₀ and FSI₅PF₅ (Figure S21), the pouch cells could be charged to the upper cutoff voltage. Although pronounced fluctuation in CE and discharge capacity were observed for cells with FSI₁DFOB₀ during initial cycles, induced by Al dissolution (as confirmed by increased current during constant voltage step in the 5th cycle shown in Figure 7e), the pouch cell can still be cycled. This confirms that SUS dissolution, rather than Al dissolution, is dominant failure in cells containing SUS components.

The following text and figures have been included in the main text:

“

Figure 7. (a) Voltage vs. time profiles of coin cells with SUS316 parts using $FS1DFOB_0$ and $FS3DFOB_5$. (b) Specific discharge capacity vs. cycle number curves of coin cells containing SUS316 parts with $FS5DFOB_5$ and LP57 electrolytes. (c) Specific discharge capacity vs. cycle number curves of NMC811||Gr coin cells containing SUS316L parts with $FS5DFOB_5$ and LP57. (d) Specific discharge capacity vs. cycle number curves of NMC811||Si-C coin cells containing SUS316L parts with $FS5DFOB_5$ and LP57. (e) Specific discharge capacity vs. cycle number curves of NMC811||Si-C pouch cells with $FS1DFOB_0$, $FS5DFOB_5$, $FS5PF_5$ and LP57. (f) Current vs. time profile during the constant voltage step at 5th cycle for cells with considered electrolytes. Note: Replacing the coin cell components from SUS316 to SUS316L does not affect the galvanostatic cycling performance for both NMC811||Gr and NMC811||Si-C cells using LP57 electrolytes, as detailed in the Supplementary Note 6. A dedicated coin cell and pouch cell configuration is shown in Figure S3.

Further evaluation is performed in the NMC811||Si-C pouch cell to exclude the impact from SUS components. Consistent with the results observed in coin cells, cells with $FS5DFOB_5$ exhibited a stable CE and reached 88% capacity retention after 250 cycles, in contrast to the <60% capacity retention observed in cells with LP57 after 250 cycles (Figure 7e). $FS1DFOB_0$ and $FS5PF_5$ (0.5M LiFSI+0.5M LiPF₆) are also evaluated to access the impact of Al dissolution in pouch cell environment. A pronounced fluctuation in CE and discharge capacity is observed for cells with $FS1DFOB_0$ during initial cycles. The increased current during constant voltage step in the 5th cycle confirms the existence of Al dissolution, explaining this fluctuation (Figure 7f).

The proposed mechanism of action implies that both FSI and Cl⁻ anions are hindered by oxalate anions to reach the surface of SUS. Further improvement has been achieved by incorporating more dissolution resistive SUS316L cell parts, resulting in ≈300 cycles until 80% state of health for NMC811||Si-C cell chemistry and ≈1150 cycles for NMC811||Gr cell chemistry. Furthermore, this improvement has also been confirmed in NMC811||Si-C pouch cell with 88% capacity retention after 250 cycles

Finally, the effectiveness of LiDFOB has been proven by improved galvanostatic cycling performance in NMC811||Gr and NMC811||Si-C coin cells and NMC811||Si-C pouch cells.

Further improvement has been achieved by incorporating more dissolution resistive SUS316L cell components, resulting in ≈ 300 cycles for NMC811//Si-C cell chemistry and ≈ 1150 cycles for NMC811//Gr cell chemistry. This enhanced performance is also confirmed in NMC811//Si-C pouch cell with 88 % capacity retention after 250 cycles. This strategy not only extends the life of the cells but also emphasizes the importance of the electrochemical stability of the metal in the non-aqueous electrolyte.”

6. The authors showed that the stability of the NMC811-graphite cells is quite poor; however, the LiPF₆ should offer stable electrochemical performance “As a result, the NMC811//graphite cells with SUS316L parts using FSI₅DFOB₅ electrolyte show effective CE for more than 1200 cycles, as compared to an ineffective CE from 200 cycles for those with LP57 (1M LiPF₆ in EC:EMC, 3:7) electrolyte (Figure 7c).” There should be no corrosion issues or serious degradation of the NMC811 in the potential window 2.5-5.0 V and LiPF₆ salt in carbonate electrolytes. Accordingly, the results should be carefully revised, and the interpretation should be corrected.

We regret that the wording of this paragraph could have created confusion. We fully agree that the LiPF₆ do not have the dissolution issues with both Al and SUS, and is electrochemically stable in the operation voltage range. However, LiPF₆ faces the problems of chemical and thermal instability which compromise the electrochemical performance, especially in such demanding cell chemistry^[13]. The HF released by LiPF₆ decomposition causes transition metal dissolution and destruction of SEI/CEI, which is challenging for cells employing nickel-rich NMC cathode^[14]. Therefore, the poor galvanostatic cycling stability of NMC811//Gr and NMC811//Si-C (20 % Si) is not attributed to the metal dissolution, rather to the intrinsic instability of LiPF₆.

The following text have been included in the main text:

“As a result, the NMC811//Gr cells with SUS316L parts using FSI₅DFOB₅ electrolyte show effective CE for more than 1200 cycles, resulting in a significantly improved galvanostatic cycling performance of ≈ 1150 cycles to 80 % state-of-health (SOH_{80%}), representing nearly six-fold increase in cycle life compared to cells with LP57 electrolyte (Figure 7c). In addition, to evaluate the compatibility of the optimized electrolyte with different cell chemistries, silicon-graphite (Si-C) anodes containing 20% silicon were also considered. NMC811//Si-C cells with FSI₅DFOB₅ maintain an effective CE with nearly double the lifespan compared to cells with LP57, reaching ≈ 300 cycles (Figure 7d). This enhanced galvanostatic cycling performance in both NMC811//Gr and NMC811//Si-C cells originates not only from the suppressed Al and SUS dissolution, but also from the superior physicochemical properties (increased thermal and chemical stability compared to LiPF₆, ion mobility, etc.) provided by LiFSI^[39], as well as the effective SEI/CEI formation contributed by LiDFOB^[51]. Further evaluation is performed in the NMC811//Si-C pouch cell to exclude the impact from SUS components. Consistent with the results observed in coin cells, cells with FSI₅DFOB₅ exhibited a stable CE and reached 88 % capacity retention after 250 cycles, in contrast to the <60 % capacity retention observed in cells with LP57 after 250 cycles (Figure 7e). FSI₁DFOB₀ and FSI₅PF₅ (0.5M LiFSI+0.5M LiPF₆) are also evaluated to assess the impact of Al dissolution in pouch cell setup. A pronounced fluctuation in CE and specific discharge capacity is observed for cells with FSI₁DFOB₀ during initial cycles. The increased current during constant voltage step in the 5th cycle confirms the existence of Al dissolution, explaining this fluctuation (Figure 7f). Interestingly, although the addition of LiPF₆ suppresses Al dissolution and results in a stable CE, the overall galvanostatic cycling performance remains similar to the cells with LP57. This indicates that the LiFSI alone, after excluding the impact of Al and SUS dissolution, is

*insufficient to ensure long-term cycling stability in such demanding cell chemistry. The superior film forming capability from LiDFOB is required to stabilize both electrodes. This conclusion is further supported by the improved performance observed in NMC811||Li and Gr||Li cell setups with FSI₅DFOB₅ electrolytes, as detailed in **Supplementary Note 5**. The obtained results demonstrate that while FSI₅DFOB₅ electrolyte considerably enhances cell stability by suppressing Al and SUS dissolution, its intrinsic physicochemical advantages and effective SEI/CEI formation capability also contribute significantly to long-term cycling. Moreover, substituting SUS316 with SUS316L further boosts performance, underscoring the critical role of material selection coupled with electrolyte formulation.”*

7. Finally, the authors are encouraged to explain and judge all possible explanations of the corrosion phenomenon and cell failure rather than steering the interpretation to a specific mechanism and ignoring other possibilities.

Following reviewer’s suggestion, we have considered all meaningful explanations for the SUS dissolution, the inhibitive effect by LiDFOB and cell failure in coin cells. Regarding the possible explanation for SUS dissolution, we have considered different factors including proton release during the solvent oxidation and aggressive Cl⁻ and FSI⁻ anions. Electrochemical experiments with fluorinated solvents have confirmed that the released proton during solvent oxidation do not contributed to the SUS dissolution (Supplementary Note 3). Instead, the observed dissolution is predominantly associated with the presence of Cl⁻ and FSI⁻ anions.

As regarding the inhibitive effect of LiDFOB, we have confirmed the lithium salts used in this study are sufficiently high purity and that fluoride anion does not impact the SUS dissolution (Figure S21). Furthermore, although boron-containing surface species were detected, their presence however does not provide protection for SUS against dissolution (Supplementary Note 1), and the competing anion adsorption by DFOB⁻ anions is the most plausible explanation for the suppression effect.

The following paragraph has been included in the main text:

“Proposed mechanism of SUS dissolution and inhibitive effect of LiDFOB

Comprehensive experimental and theoretical evaluation enabled discussion regarding the possible mechanism for SUS dissolution and LiDFOB inhibition. Electrochemical experiments with fluorinated solvents have confirmed that the released proton during solvent oxidation does not contribute to the SUS dissolution. Instead, the observed dissolution is predominantly associated with the presence of Cl⁻ and FSI⁻ anions. Regarding the inhibitive effect of LiDFOB, LiPF₆ as a representative source for fluoride anion, has been confirmed that fluoride does not impact the SUS dissolution. Furthermore, although boron-containing surface species were detected, their presence however does not provide protection for SUS against dissolution, and the competing anion adsorption by DFOB⁻ anions is the most plausible explanation for the suppression effect. A detailed discussion of the proposed mechanism is provided below.”

The possible explanations for coin cell failure is associated with metal (Al and SUS) dissolution in the presence of LiFSI and also the roll-over failure in the cells containing LP57 electrolyte. Metal dissolution can cause charge failure, as indicated by the increased anodic current during the constant voltage step (Figure 7a for SUS dissolution and Figure 7f for Al dissolution). The addition of LiDFOB can effectively suppress the metal dissolution. As for the roll-over failure observed in cells with LP57, it is a complex process which related to the electrolyte decomposition, TM dissolution and reconstruction of SEI/CEI, which mostly due to the intrinsic instability of LiPF₆.

The combination of LiFSI and LiDFOB combines the advantages of superior physicochemical properties from LiFSI and film forming ability from LiDFOB, which significantly enhances the electrochemical performance.

References:

- [1] S. S. Zhang, “A Study on Corrosion of Al-Clad Coin Cell Cases in High-Voltage Li-Ion Battery” *J. Electrochem. Soc.* **2023**, *170*, 110527.
- [2] M. C. Stan, P. Yan, G. M. Overhoff, N. Fehlings, H. Kim, R. T. Hinz, T. T. K. Ingber, R. Guerdelli, C. Wölke, M. Winter, G. Brunklaus, I. Cekic-Laskovic, “Unraveling Influential Factors of Stainless-Steel Dissolution in High-Energy Lithium Ion Batteries with LiFSI-Based Electrolytes” *ChemElectroChem* **2025**, e202400632.
- [3] Q. Liu, T. L. Dzwiniel, K. Z. Pupek, Z. Zhang, “Corrosion/Passivation Behavior of Concentrated Ionic Liquid Electrolytes and Its Impact on the Li-Ion Battery Performance” *J. Electrochem. Soc.* **2019**, *166*, A3959–A3964.
- [4] Y. Bae, H. G. Lee, Y. J. Kim, G. R. Kim, J.-W. Park, J. Moon, Y.-J. Lee, J.-H. Choi, B. G. Kim, “Elucidating critical origin for capacity fading in High-voltage coin cell with FSI-based electrolyte” *Chem. Eng. J.* **2023**, *469*, 143804.
- [5] C. Luo, Y. Li, W. Sun, P. Xiao, S. Liu, D. Wang, C. Zheng, “Revisiting the corrosion mechanism of LiFSI based electrolytes in lithium metal batteries” *Electrochim. Acta* **2022**, *419*, 140353.
- [6] L. Wang, Z. Luo, H. Xu, N. Piao, Z. Chen, G. Tian, X. He, “Anion effects on the solvation structure and properties of imide lithium salt-based electrolytes” *RSC Adv.* **2019**, *9*, 41837.
- [7] J. Gorsch, J. Schneiders, M. Frieges, N. Kisseler, D. Klohs, H. Heimes, A. Kampker, M. Muñoz Castro, E. Siebecke, “Contrasting a BYD blade prismatic cell and tesla 4680 cylindrical cell with a teardown analysis of design and performance” *Cell Rep. Phys. Sci.* **2025**, *6*, 102453.
- [8] T. Ma, G.-L. Xu, Y. Li, L. Wang, X. He, J. Zheng, J. Liu, M. H. Engelhard, P. Zapol, L. A. Curtiss, J. Jorne, K. Amine, Z. Chen, “Revisiting the Corrosion of the Aluminum Current Collector in Lithium-Ion Batteries” *J. Phys. Chem. Lett.* **2017**, *8*, 1072–1077.
- [9] K. M. Scheer, M. Tulloch, I. Hamam, J. J. Abraham, M. B. Johnson, M. Metzger, “Anodic Dissolution of the Aluminum Current Collector in Lithium-ion Cells with LiFSI, LiPF₆, and LiBF₄” *J. Electrochem. Soc.* **2025**, *172*, 010511.
- [10] L. Nyholm, T. Ericson, A. S. Etman, “Revisiting the stability of aluminum current collectors in carbonate electrolytes for high-voltage Li-ion batteries” *Chem. Eng. Sci.* **2023**, *282*, 119346.
- [11] H.-B. Han, S.-S. Zhou, D.-J. Zhang, S.-W. Feng, L.-F. Li, K. Liu, W.-F. Feng, J. Nie, H. Li, X.-J. Huang, “Lithium bis(fluorosulfonyl)imide (LiFSI) as conducting salt for nonaqueous liquid electrolytes for lithium-ion batteries: Physicochemical and electrochemical properties” *J. Power Sources* **2011**, *196*, 3623–3632.
- [12] Y. Wang, Z. Li, Y. Hou, Z. Hao, Q. Zhang, Y. Ni, Y. Lu, Z. Yan, K. Zhang, Q. Zhao, F. Li, J. Chen, “Emerging electrolytes with fluorinated solvents for rechargeable lithium-based batteries” *Chem. Soc. Rev.* **2023**, *52*, 2713–2763.

- [13] K. Xu, “Nonaqueous Liquid Electrolytes for Lithium-Based Rechargeable Batteries” *Chem. Rev.* **2004**, *104*, 4303–4418.
- [14] T. Dong, S. Zhang, Z. Ren, L. Huang, G. Xu, T. Liu, S. Wang, G. Cui, “Electrolyte Engineering Toward High Performance High Nickel ($\text{Ni} \geq 80\%$) Lithium-Ion Batteries” *Adv. Sci.* **2024**, *11*, 2305753.

Reviewer #4:

The authors have revised the manuscript and addressed most of the raised comments. However, the manuscript still needs further polishing to be more concise and to offer a clear message to the readers.

We thank the reviewer for the constructive feedback. In response, the manuscript has been further refined to improve concision and sharpen the central message. Specifically, the Results and Discussion sections were streamlined, redundant statements were removed, and key paragraphs were revised to provide clearer support for the main conclusions. In addition, the order of the figures in the Supporting Information was reorganized to improve readability.

Here are the main revisions in the main text:

Before (lines 91 – 100): *“Therefore, in order to further understand the effect of Cl⁻ anion on the SUS dissolution, we have enriched the concentration of Cl⁻ anions in the electrolyte solution by adding lithium chloride (LiCl). It should be noted that 7.49 ppm of Cl⁻ anion impurities were detected in the LiFSI salt by ion chromatography (IC) measurements.*

*Considering the limited solubility of LiCl in organic solvents, 700 ppm of LiCl were added to a mixture of EC: EMC (3:7 by weight), resulting in an electrolyte that is nearly saturated with LiCl (referred to as LiCl_{sat}). Linear sweep voltammetry (LSV) was carried out to determine the electrochemical stability of the electrolytes against SUS using an adapted cell configuration as illustrated in **Figure S1a**.”*

After (lines 91 – 98): *“Therefore, to further elucidate the effect of Cl⁻ anion on the SUS dissolution, we have enriched the concentration of Cl⁻ anions by adding 700 ppm of lithium chloride (LiCl) to a mixture of EC: EMC (3:7 by weight), resulting in an electrolyte that is nearly saturated with LiCl (referred to as LiCl_{sat}). It should be noted that 7.49 ppm of Cl⁻ anion impurities were detected in the LiFSI salt by ion chromatography (IC) measurements.*

Linear sweep voltammetry (LSV) was carried out to determine the electrochemical stability of the electrolytes against SUS.”

Before (lines 139 – 144): *“The results obtained from the CA experiments clearly indicate that SUS dissolution is more pronounced in the cell with LiCl_{sat} compared to the cell with FSI₁DFOB₀. However, subsequent analysis using scanning electron microscopy (SEM) revealed further differences between these samples. **Figure 1c** and **Figure 1d** show SEM images of the polished spacers after 20 h in CA measurement, illustrating the changes in surface morphology. A distinct difference can be observed between the spacers harvested from cells with the two electrolytes.”*

After (lines 138 – 143): *“The results from the CA experiments clearly indicate that SUS dissolution is more pronounced in the cell with LiCl_{sat} compared to the cell with FSI₁DFOB₀. However, subsequent analysis using scanning electron microscopy (SEM) revealed distinct differences between these samples. **Figure 1c** and **Figure 1d** show SEM images of the SUS316 spacers after 20 h in CA measurement, illustrating the changes in surface morphology between the spacers harvested from cells with the two electrolytes.”*

Before (lines 164 – 168): “The dissolved Fe ions (whether in the form of soluble complexes or free ions) can diffuse through the electrolytes and deposit on the anode surface. Hereby, the counter Li metals were also examined using SEM-EDX to investigate the crosstalk of transition metal ions (TMs). **Figure S9b** shows the SEM images of counter Li metals harvested after CA measurements.”

After (lines 164 – 166): “The dissolved Fe ions diffuse through the electrolytes and deposit on the anode surface, changing the morphology of the Li metal surface as indicated by the SEM-EDX measurements of Li metals counter electrodes harvested after CA measurements (**Figure S4b**).”

Before (lines 188 – 190): “At 50 ppm LiCl, the second re-passivation process is absent, and at 100 ppm LiCl, re-passivation does not occur at all, suggesting continuous SUS dissolution at high Cl⁻ concentrations.”

After (lines 185 – 188): “At 50 ppm LiCl, the second re-passivation process is no longer observed, and at 100 ppm LiCl, re-passivation entirely disappeared, indicating continuous SUS dissolution at elevated Cl⁻ concentrations.”

Before (lines 196 – 205): “The spacer surface treated with low Cl⁻ concentration solutions was covered with black deposits (**Figure 2d-f**), which can be distinguished as iron oxide compounds (**Figure S8**), while those exposed to high Cl⁻ anion concentration solutions exhibited pitting on the surface (**Figure 2g-i**). Overall, Cl⁻ anion impurity plays an important role in initiating pitting dissolution in SUS within LiFSI-based electrolytes. The obtained results indicate that over 50 ppm of Cl⁻ anion concentrations are necessary to hinder the re-passivation process, thus continuing the pitting process. However, even low levels of Cl⁻ anions can initiate pitting, which might act as a starting point for the SUS dissolution in the presence of LiFSI, highlighting the determining role of Cl⁻ anions in SUS dissolution.”

After (lines 194 – 200): “The spacer surface treated with low Cl⁻ concentration solutions was covered with black deposits (**Figure 2d-f**), identified as iron oxide compounds (**Figure S5**), while those exposed to high Cl⁻ anion concentration solutions exhibited pitting on the surface (**Figure 2g-i**). Overall, Cl⁻ anion impurities are decisive in triggering pitting dissolution in SUS within LiFSI-based electrolytes, as even trace concentrations initiate pit formation, while levels above 50 ppm of Cl⁻ anion concentrations suppress the re-passivation process and enable a sustained pitting process.”

Before (lines 220 – 226): “Specifically, with the addition of 20% molar ratio of LiDFOB (referred to FSI₈DFOB₂), the critical voltage for anodic dissolution markedly increased from ≈3.5 V to 4.5 V. Increasing the LiDFOB concentration to 40% further raised the critical voltage to 4.6 V. The increase in critical voltage plateaus at 4.7 V with 50 % LiDFOB, accompanied by a further reduction in dissolution current density. The electrolyte with pure LiDFOB (referred to FSI₀DFOB₁) showed the most stable current, with only a slight current increase observed at 5 V.”

After (lines 215 – 220): “Specifically, with the addition of 20 % molar ratio of LiDFOB (referred to FSI₈DFOB₂), the critical voltage for anodic dissolution markedly shifted from ≈3.5 V to 4.5 V. Further increasing the LiDFOB concentration to 40 % and 50 % shifts the critical voltage to 4.6 V and 4.7 V, respectively. The current decreased steadily as the amount of LiDFOB increased, with

the pure LiDFOB (referred to FSI₀DFOB₁) showing the most stable current, with only a slight current increase observed at 5 V.”

Before (lines 239 – 243): *“No anodic current is observed for cells with FSI₅DFOB₅ (Figure 3e) and only a slightly anodic current rised at ≈4.2 V for cells with FSI₀DFOB₁+ LiCl_{sat} (Figure 3f), with a current density maximum of 0.001 mA cm⁻². This value is significantly lower than the pronounced current density maximum of 0.08 mA cm⁻² observed in cell containing FSI₁DFOB₀+ LiCl_{sat}.”*

After (lines 234 – 249): *“No anodic current is observed for cells with FSI₅DFOB₅ (Figure 3e) and only a low anodic current developed at ≈4.2 V for cells with FSI₀DFOB₁+ LiCl_{sat} (Figure 3f), reaching a maximum current density of 1.0 μA cm⁻². This value is substantially lower than the maximum current density of 80.0 μA cm⁻² observed in the cell containing FSI₁DFOB₀+ LiCl_{sat}. In addition, SEM analysis confirms the absence of pits on SUS surface for cells with FSI₀DFOB₁+ LiCl_{sat} (Figure S6).”*

Before (lines 264 – 279): *“X-ray photoelectron spectroscopy (XPS) analysis conducted on SUS316 spacers harvested from cells containing FSI₁DFOB₀, FSI₅DFOB₅ and FSI₀DFOB₁ electrolytes after a 20 h holding at 4.2 V as shown in Figure 3g-l. The results revealed pronounced differences in the intensity and shape of spectra between cells using pure LiFSI-based electrolytes (FSI₁DFOB₀) and LiDFOB-containing electrolytes (FSI₅DFOB₅ and FSI₀DFOB₁). The observed differences are likely attributed to the dissolution products deposited on the SUS316 surface in the presence of the FSI₁DFOB₀ electrolyte. Analysis of the Fe 2p and Cr 2p spectra reveals insight into the oxidation state of iron and chromium metal. The pristine SUS316 surface shows predominantly the signal of Fe metal at 707.5 eV in the Fe 2p spectra and the signal of Cr metal at 574.4 eV in the Cr 2p spectra, indicative of minor surface oxide from chromium and iron (Figure S11). However, for samples harvested from cells containing FSI₁DFOB₀ electrolyte, the peaks for pure Fe and Cr are considerably reduced. Instead, there is a notable increase in the Fe³⁺ and Cr⁶⁺ signals within the Fe 2p and Cr 2p spectra, suggesting a pronounced oxidation of iron and chromium metal (Figure 3g,h). Additionally, the analysis reveals signals of S-F, S-N and SO_x in the F 1s, N 1s and S 2p spectra, reflecting the presence of decomposition products from FSI anions (Figure 3i-k).”*

After (lines 260 – 275): *“X-ray photoelectron spectroscopy (XPS) analysis conducted on SUS316 spacers harvested from cells containing FSI₁DFOB₀, FSI₅DFOB₅ and FSI₀DFOB₁ electrolytes after a 20 h holding at 4.2 V is shown in Figure 3g-l. These investigations revealed pronounced differences in the intensity and shape of spectra between cells using pure LiFSI-based electrolytes (FSI₁DFOB₀) and LiDFOB-containing electrolytes (FSI₅DFOB₅ and FSI₀DFOB₁), which that are attributed to the dissolution products deposited on the SUS316 surface in the presence of the FSI₁DFOB₀ electrolyte. Analysis of the Fe 2p and Cr 2p spectra reveals insight into the oxidation state of iron and chromium metal. The pristine SUS316 surface shows predominantly the signal of Fe metal at 707.5 eV in the Fe 2p spectra and the signal of Cr metal at 574.4 eV in the Cr 2p spectra, indicative of minor surface oxide from chromium and iron (Figure S7). However, for samples harvested from cells containing FSI₁DFOB₀ electrolyte (Figure 3g,h), the peaks for pure Fe and Cr are considerably reduced, with notable increase in the Fe³⁺ and Cr⁶⁺ signals within the Fe 2p and Cr 2p spectra, suggesting a pronounced oxidation of iron and chromium metal.*

Additionally, signals of S-F, S-N and SO_x in the F 1s, N 1s and S 2p spectra (**Figure 3i-k**), reflect the presence of decomposition products from FSI⁻ anions in the FSI₁DFOB₀ electrolyte.”

Before (lines 299 – 310): “Given that DFOB⁻ anion comprises molecular moieties from both BF₄⁻ and BOB⁻, an additional LSV experiment was conducted using electrolytes containing LiBF₄ and LiBOB separately. Equivalent molar ratios of both borate salts were dissolved in an EC:EMC solvent mixture at a ratio of 3:7 by weight. Due to the solubility limitations of LiBOB, only a 20 % molar ratio was considered for these experiments. The LSV results indicate that while the addition of LiBF₄ does not change the critical voltage of SUS dissolution, the electrolytes containing LiDFOB and LiBOB exhibit a similar suppression effect on the dissolution of SUS (**Figure 4a**). This observation suggests that the oxalate moiety in the DFOB⁻ anion is critical in mitigating SUS dissolution in environments with Cl⁻ and FSI⁻ anions. Considering that the oxalate group is easily decomposed and a boron-containing film is observed on the SUS316 surface when using LiDFOB-containing electrolyte (**Figure 3l**), it was initially hypothesized that this film could shield SUS from the attack of aggressive Cl⁻ and FSI⁻ anions.”

After (lines 293 – 301): “Given that DFOB⁻ anion comprises molecular moieties from both BF₄⁻ and BOB⁻, an additional LSV experiment was conducted using electrolytes containing equivalent molar ratios LiBF₄ and LiBOB separately dissolved in an EC:EMC solvent mixture at a ratio of 3:7 by weight. Due to the solubility limitations of LiBOB, only a 20 % molar ratio was considered for these experiments. The LSV results in **Figure 4a** indicate that while the addition of LiBF₄ does not change the critical voltage of SUS dissolution, the oxalate moiety is critical in mitigating SUS dissolution in environments with Cl⁻ and FSI⁻ anions. Since the oxalate group is easily decomposed, forming a boron-containing film on the SUS316 surface (**Figure 3l**), it is hypothesized that this film could shield SUS from the attack of aggressive Cl⁻ and FSI⁻ anions.”

Before (lines 344 – 347): “A similar trend was also observed in complementary experiments in organic electrolyte systems shown in **Figure S12**. In general, the DFOB⁻ anions preferentially adsorb and accumulate on the SUS surface, thereby excluding FSI⁻ and Cl⁻ anions from attacking the SUS surface.

After (lines 335 – 336): “A similar trend was also observed in complementary experiments in organic electrolyte systems shown in **Figure S8**.”

Before (lines 358 – 382): “Four regions of gas evolution during the SUS dissolution process could be identified. The first region starts from 0 V vs. Li/Li⁺ and continues until the second self-corrosion potential^[24]. A rapid release of H₂ is observed after applying 0 V vs. Li/Li⁺ for all cells containing considered electrolytes, likely due to electrolyte or water reduction at the anode surface^[48]. In addition, a continuous release of CO₂ is observed in cells with FSI₁DFOB₀ electrolytes in this region. The second region includes the range where the primary and secondary passivation processes take place, with no gas evolution observed in this region. The third region is defined by the potential range where the Cl⁻ anion induced dissolution appears. An H₂ release is observed as a consequence of SUS dissolution at increased potential. The mechanism for potential-dependent H₂ evolution remains disputed and unclear, possibly originated from electrochemical reaction of electrolyte or SEI decomposition^[48-51]. It is noted that only a small

amount of H_2 is released in this stage considering the re-passivation process, as indicated by the decreases in current after an initial increase in LSV. With the further increase in voltage, a significant H_2 release is observed in DEMS cells containing LiCl and FSI₁DFOB₀ in the fourth region. The release of CO_2 , typically attributed to electrolyte decomposition, is observed with all considered electrolytes. DEMS cells containing LiCl release CO_2 earlier than others, possibly due to the lower electrochemical stability of free solvents, as evidenced by Raman analysis (Figure S13). In terms of other electrolytes, DEMS cells with FSI₁DFOB₀ release CO_2 slightly earlier compared to LiDFOB-containing electrolytes. In addition, CO and O_2 release is also observed in DEMS cells with FSI₁DFOB₀ electrolytes, indicating that electrolyte decomposition is triggered by the SUS dissolution. Notably, gas evolution is substantially suppressed in the presence of LiDFOB and no H_2 , CO and O_2 were observed in DEMS cell with FSI₀DFOB₁ and FSI₅DFOB₅. These gas evolution results confirmed that SUS dissolution induces electrolyte decomposition, and that the introduction of LiDFOB can considerably mitigate electrolyte decomposition by inhibiting the dissolution process.”

After (lines 346 – 367): “Four regions of gas evolution during the SUS dissolution process could be identified: Region (I) starts from 0 V vs. Li|Li⁺ and continues until the second self-corrosion potential^[24], and it is in this region that a rapid release of H_2 is observed for all electrolytes after applying 0 V vs. Li|Li⁺. This is likely due to electrolyte or water reduction at the anode surface^[48]. In addition, a continuous release of CO_2 is observed for the cells with FSI₁DFOB₀ electrolytes in this region. Region (II) includes the primary and secondary passivation processes, where no gas evolution takes place. Region (III) is defined by the potential range where the Cl⁻ anion induced dissolution appears, while a slight H_2 release, as a consequence of SUS dissolution at increased potential, is observed. Such potential-dependent H_2 evolution remains disputed and unclear, possibly originating from electrochemical reaction of electrolyte or SEI decomposition driven by the dissolved TMs^[48-51]. It is noted that only a small amount of H_2 is released in this stage considering the re-passivation process, as indicated by the decreases in current after an initial increase in LSV. Region (IV) is defined by a pronounced H_2 and an early CO_2 release particularly with the cells containing LiCl and FSI₁DFOB₀. The release of CO_2 in this region is attributed to electrolyte decomposition. DEMS cells containing LiCl release CO_2 earlier than others, possibly due to the lower electrochemical stability of free solvents, as evidenced by Raman analysis (Figure S9). CO and O_2 release is also observed in DEMS cells with FSI₁DFOB₀ electrolytes, indicating electrolyte decomposition triggered by the SUS dissolution. In the presence of LiDFOB, as it is the case of DEMS cells with FSI₀DFOB₁ and FSI₅DFOB₅, H_2 , CO and O_2 gas evolution is substantially suppressed. These results further confirm that the presence of LiDFOB can suppress the SUS dissolution and thus also mitigate electrolyte decomposition.”

Before (lines 390 – 395): “Electrochemical experiments with fluorinated solvents have confirmed that the released proton during solvent oxidation does not contribute to the SUS dissolution. Instead, the observed dissolution is predominantly associated with the presence of Cl⁻ and FSI⁻ anions. Regarding the inhibitive effect of LiDFOB, LiPF₆ as a representative source for fluoride anion, has been confirmed that fluoride does not impact the SUS dissolution.”

After (lines 379 – 384): “Electrochemical experiments with fluorinated solvents have confirmed that the protons released during solvent oxidation do not contribute to the SUS dissolution (Supplementary Note 3). Instead, the observed dissolution is predominantly associated with the presence of Cl⁻ and FSI⁻ anions. To clarify the inhibitive effect of LiDFOB, LiPF₆ was used as a

representative source for fluoride anions and confirms that fluoride does not impact the SUS dissolution.”

Before (lines 399 – 407): “The proposed mechanism for the SUS dissolution process follows different scenarios, dependent on the presence of different anions and is schematically outlined in **Figure 6**. In the first scenario, when only the Cl^- anions are present, the dissolution mechanism outlined in **Figure 6a** implies: (1) The Cl^- anions accumulate on the SUS surface and attack the oxide film and expose the underlying SUS. (2) The exposed SUS begins to pit and is oxidized to Fe^{2+} under the applied potential. (3) As the potential increases further, Fe^{2+} is oxidized to Fe_2O_3 , which precipitates on the surface and re-passivates the pits. It needs to be highlighted that the SUS surface is largely covered by Fe_2O_3 , slowing down the continued SUS dissolution process and decreasing the anodic current even at higher potentials.”

After (lines 389 – 396): “The complex process of SUS dissolution is schematically outlined in **Figure 6** considering various scenarios. In the first scenario, when the Cl^- anions are present, the dissolution mechanism outlined in **Figure 6a** implies: (1) The Cl^- anions accumulate on the SUS surface and attack the oxide film and expose the underlying SUS. (2) The exposed SUS begins to pit and is oxidized to Fe^{2+} under the applied potential. (3) With growing potential, Fe^{2+} is oxidized to Fe_2O_3 , which precipitates on the surface and re-passivates the pits. The predominant Fe_2O_3 coverage on the SUS surface suppresses further dissolution, leading to reduced anodic current even at higher potentials.”

Before (lines 425 – 433): “LiDFOB preferentially adsorbs and accumulates on the SUS surface, excluding the Cl^- and FSI- anions from the surface, thus preventing SUS dissolution at high potentials. In addition, the release of H_2 , CO and O_2 is also inhibited due to the mitigated TMs with the addition of LiDFOB, highlighting the importance of LiDFOB in stabilizing LiFSI-based electrolytes. It should be emphasized that LiDFOB continues to decompose due to electrochemical instability, as evidenced by the 1,000 h CA measurements and the detected boron-containing film on the SUS surface (**Figure S16**). However, these decomposition products from LiDFOB may also contribute to SEI or CEI formation on the electrodes, therefore improving the electrochemical performance of the LIBs.”

After (lines 414 – 423): “LiDFOB preferentially adsorbs and accumulates on the SUS surface, thus excluding the Cl^- and FSI- anions from the surface, preventing SUS dissolution at high potentials. Consequently, the addition of LiDFOB suppresses H_2 , CO and O_2 evolution by mitigating TMs dissolution, underscoring its critical role in stabilizing LiFSI-based electrolytes. It should be emphasized that LiDFOB continues to decompose due to electrochemical instability, as evidenced by the 1,000 h CA measurements and the detected boron-containing film on the SUS surface (**Figure S16**). However, these decomposition products from LiDFOB may also contribute to SEI or CEI formation on both electrodes, effectively suppressing interfacial degradation and electrolyte depletion, therefore improving the electrochemical performance of the LIBs^[46,52-54].”

Before (lines 441 – 446): “To assess the effectiveness of LiDFOB in suppressing the SUS dissolution, the electrochemical performance was evaluated using two different LIB chemistries in a coin cell setup. This setup introduces another challenge, as Al, usually used as current collector

for positive electrodes, can also undergo dissolution at high voltage, particularly in contact with SUS within the coin cell^[23,27]. Nevertheless, this issue could also be mitigated by using optimized electrolytes containing LiDFOB, as detailed in **Supplementary Note 2.**”

After (lines 431 – 436): “Consistent with the role of LiDFOB in suppressing the SUS dissolution, the electrochemical performance was further evaluated using two different LIB chemistries (**Figure 7**). Besides the SUS dissolution process, Al dissolution can occur at higher voltage, particularly when in contact with SUS and in the presence of LiFSI^[23,27]. Nevertheless, this issue could also be mitigated by using optimized electrolytes containing LiDFOB, as detailed in **Supplementary Note 2.**”

Before (lines 458 – 474): “As depicted in **Figure 7a**, the cells assembled with NMC811 as cathode material on an Al foil current collector and graphite as anode material utilizing the FSI₁DFOB₀ electrolyte exhibit a pronounced rise in anodic current during the constant voltage step, which may be introduced by SUS or Al dissolution. With further post-mortem analysis discussed in the **Supplementary Note 4**, this rise in anodic current is mostly attributed to the SUS dissolution. This severe SUS dissolution detrimentally affects the galvanostatic cycling performance of the cell, leading to failure in the initial cycle. In contrast, the cell using LiDFOB-containing electrolyte (FSI₅DFOB₅) demonstrates stable charging and discharging behavior, as confirmed by the decent cycling stability over the first 150 cycles (**Figure 7b**). However, a large deviation of discharge capacity along with a rapid drop in Coulombic efficiency (CE) in the subsequent cycles is observed, indicating ongoing SUS dissolution. This can be confirmed with further post-mortem analysis discussed in **Supplementary Note 4**. The obtained results indicate that while the presence of LiDFOB improves the stability of SUS316, it does not completely eliminate SUS dissolution during prolonged galvanostatic cycling. Further enhancement of electrochemical stability is achieved by employing SUS316L components, which exhibit greater resistance to dissolution than SUS316, as evidenced by the higher critical voltage observed in LSV for cells employing SUS316L (**Figure S14**).”

After (lines 448 – 461): “As depicted in **Figure 7a**, the cells assembled with NMC811 as positive electrode material on an Al foil current collector and graphite as negative electrode material utilizing the FSI₁DFOB₀ electrolyte exhibit a pronounced rise in anodic current during the constant voltage step, which originates from SUS or Al dissolution. Further post mortem analysis discussed in the **Supplementary Note 4** indicates mostly the SUS dissolution as the responsible process for the anodic current increase and leads to failure in the initial cycle. In contrast, the cells using LiDFOB-containing electrolyte (FSI₅DFOB₅) demonstrates stable charge and discharge behavior, as confirmed by the decent galvanostatic cycling stability over the first 150 cycles (**Figure 7b**). However, a large deviation of specific discharge capacity along with a rapid drop in Coulombic efficiency (CE) in the subsequent cycles indicate that SUS dissolution is not fully suppressed over extended cycling, consistent with post mortem analysis discussed in **Supplementary Note 4**. Further enhancement of electrochemical stability is achieved by employing SUS316L components, which exhibit higher dissolution resistance than SUS316, as evidenced by the higher critical voltage observed in LSV for cells employing SUS316L (**Figure S10**).”

Before (lines 478 – 498): “In addition, to evaluate the compatibility of the optimized electrolyte with different cell chemistries, silicon-graphite (Si-C) anodes containing 20% silicon were also

considered. NMC811//Si-C cells with FSI₅DFOB₅ maintain an effective CE with nearly double the lifespan compared to cells with LP57, reaching ≈300 cycles (**Figure 7d**). This enhanced galvanostatic cycling performance in both NMC811//Gr and NMC811//Si-C cells originates not only from the suppressed Al and SUS dissolution, but also from the superior physicochemical properties (increased thermal and chemical stability compared to LiPF₆, ion mobility, etc.) provided by LiFSI^[39], as well as the effective SEI/CEI formation contributed by LiDFOB^[51]. Further evaluation is performed in the NMC811//Si-C pouch cell to exclude the impact from SUS components. Consistent with the results observed in coin cells, cells with FSI₅DFOB₅ exhibited a stable CE and reached 88 % capacity retention after 250 cycles, in contrast to the <60 % capacity retention observed in cells with LP57 after 250 cycles (**Figure 7e**). FSI₁DFOB₀ and FSI₅PF₅ (0.5M LiFSI+0.5M LiPF₆) are also evaluated to assess the impact of Al dissolution in pouch cell setup. A pronounced fluctuation in CE and specific discharge capacity is observed for cells with FSI₁DFOB₀ during initial cycles. The increased current during constant voltage step in the 5th cycle confirms the existence of Al dissolution, explaining this fluctuation (**Figure 7f**). Interestingly, although the addition of LiPF₆ suppresses Al dissolution and results in a stable CE, the overall galvanostatic cycling performance remains similar to the cells with LP57. This indicates that the LiFSI alone, after excluding the impact of Al and SUS dissolution, is insufficient to ensure long-term cycling stability in such demanding cell chemistry. The superior film forming capability from LiDFOB is required to stabilize both electrodes. ”

After (lines 466 – 497): “To assess the compatibility of the optimized electrolyte with more demanding negative electrodes, silicon-graphite (Si-C) anodes containing 20 % silicon were also considered. NMC811//Si-C cells with FSI₅DFOB₅ maintain an effective CE with nearly double the lifespan compared to cells with LP57, reaching ≈300 cycles (**Figure 7d**). This enhanced galvanostatic cycling performance in both NMC811//Gr and NMC811//Si-C cells originates not only from the suppressed Al and SUS dissolution, but also from the complementary role of the two conducting salts. LiFSI offers superior physicochemical properties (increased thermal and chemical stability compared to LiPF₆, ion mobility, etc.)^[39], and has been reported for stabilizing anode by forming LiF-rich SEI^[17,54]. In addition, LiDFOB is beneficial for effective SEI/CEI formation and has been reported to form boron-containing interphase species, which are associated with more effective SEI and CEI layer^[52,54,55].”

To eliminate any influence from SUS components and to evaluate the effect of electrolytes with different salts, NMC811//Si-C pouch cells were evaluated. Consistent with the results observed in coin cells, cells with FSI₅DFOB₅ displayed a stable CE value and reached ≈320 cycles at SOH_{80%}, whereas the cells with LP57 reach ≈140 cycles at SOH_{80%} (**Figure 7e**). The consistent trend observed in both coin and pouch cells suggests that the benefits of the optimized electrolyte are not restricted to laboratory scale coin cells and can extend to an industry-relevant format. As these advantages extend to the more demanding NMC//Si-C chemistry, we expect that a similar trend will hold for NMC//graphite cells, although direct pouch cell validation will be required for quantitative comparison. Importantly, pouch cells do not contain SUS components and the performance differences observed between FSI₅DFOB₅ and LP57 primarily reflect interphase stability. To further demonstrate the necessity of combining LiDFOB and LiFSI, FSI₁DFOB₀ and FSI₅PF₅ (0.5M LiFSI+0.5M LiPF₆) were also evaluated. A pronounced fluctuation in CE and specific discharge capacity is observed for cells with FSI₁DFOB₀ during initial cycles. The increased current during constant voltage step in the 5th cycle confirms the existence of Al dissolution, explaining this fluctuation (**Figure 7f**). Interestingly, although the addition of LiPF₆ suppresses Al dissolution and results in a stable CE, the overall galvanostatic cycling performance

remains similar to the cells with LP57. This indicates that the LiFSI alone, after excluding the impact of Al and SUS dissolution, is insufficient to ensure long-term cycling stability in such demanding cell chemistry. The superior film forming capability from LiDFOB is required to suppress ongoing electrolyte decomposition and stabilize both electrodes.”

Before (lines 530 – 533): “Further improvements are achieved by incorporating more dissolution-resistant SUS316L cell components, enabling ≈ 300 cycles for NMC811||Si-C cell chemistry and ≈ 1150 cycles for NMC811||Gr cell chemistries. This enhanced performance is also confirmed in 532 NMC811||Si-C pouch cell with 88 % capacity retention after 250 cycles

After (lines 527 – 531): “Further improvements are achieved by incorporating more dissolution-resistant SUS316L cell components, enabling ≈ 300 cycles for NMC811||Si-C cell chemistry and ≈ 1150 cycles at SOH_{80%} for NMC811||Gr cell chemistries. Electrolytes containing LiDFOB and LiFSI retain their performance advantages in NMC811||Si-C pouch cells, delivering ≈ 320 cycles at SOH_{80%}.”

Furthermore, each Figure, including supplementary, needs to have a title.

We thank the Reviewer for pointing out this omission. The manuscript has been carefully revised, and the figure title has been corrected accordingly in both the main text and the Supporting Information:

“Figure 2. (a) Linear sweep voltammograms of cells containing SUS316 as working electrodes with electrolytes containing 1 ppm, 5 ppm and 10 ppm LiCl concentration. (b) Linear sweep voltammogram of cells containing SUS316 as working electrodes with electrolytes containing 25 ppm, 50 ppm and 100 ppm LiCl concentration. (c) Chronoamperograms of cells containing SUS316 as working electrode with electrolytes containing different LiCl concentrations. SEM image of SUS316 surfaces harvested from cells with electrolytes containing (d) 1 ppm, (e) 5 ppm, (f) 10 ppm, (g) 25 ppm, (h) 50 ppm, (i) 100 ppm LiCl after 20 h at 4.2 V.”

“Figure 3. (a) Linear sweep voltammograms of cells containing SUS316 as working electrode and LiDFOB containing electrolytes and (b) corresponding chronoamperograms recorded at 4.2 V for 20 h. (c) Linear sweep voltammograms of cells containing SUS316 as working electrode in a mixture of LiCl_{sat} and FSI₀DFOB₁ electrolytes and (d) corresponding chronoamperograms recorded at 4.2 V for 20 h. Cyclic voltammograms of cells containing SUS316 as working electrode with (e) FSI₅DFOB₅ and (f) FSI₀DFOB₁+LiCl_{sat} electrolyte. Selected (g) Fe 2p, (h) Cr 2p, (i) F 1s, (j) N 1s, (k) S 2p and (l) B 1s XPS spectra of the harvested SUS316 spacers from the cells after 20 h at 4.2 V with FSI₁DFOB₀ (top), FSI₅DFOB₅ (middle) and FSI₀DFOB₁ (bottom) electrolytes.”

“Figure 5. (a) LSVs of a 3-electrode DEMS cells using SUS316 as working electrode with different electrolytes at scan rate of 0.1 mV s⁻¹. Operando gaseous evolution of (b) H₂, (c) CO₂, (d) CO and (e) O₂ by DEMS are monitored during the LSV, the gases are normalized with Ar to eliminate the fluctuation of carry gas. Four regions of gas evolution could be identified: Region (I) starts from 0 V vs. Li/Li⁺ and continues until the second self-corrosion potential. Region (II) includes the primary and secondary passivation processes. Region (III) is defined by the potential range where the Cl⁻ anion induced dissolution appears. Region (IV) is defined by a substantial H₂ and CO₂ release.”

“Figure S1. Linear sweep voltammograms of 3-electrode DEMS cells scanned at 0.1 mV s^{-1} using SUS316 as a working electrode containing (a) LiCl , (b) $\text{FSI}_1\text{DFOB}_0$, (c) $\text{FSI}_0\text{DFOB}_1$, (d) $\text{FSI}_3\text{DFOB}_5$ electrolytes, together with the corresponding gas evolution profiles. The following gases were measured: H_2 ($m/z=2$), CO_2 ($m/z=44$), CO ($m/z=28$), O_2 ($m/z=32$), H_2O ($m/z=18$), Cl_2 ($m/z=70, 72, 74$) and all gases were normalized to the Ar signal to eliminate the fluctuation of the carrier gas.”

“Figure S4. SEM images of SUS316 spacers recovered from cells containing (a) LiCl_{sat} , (b) $\text{FSI}_1\text{DFOB}_0$, (c) $\text{FSI}_8\text{DFOB}_2$, (d) $\text{FSI}_6\text{DFOB}_4$, (e) $\text{FSI}_5\text{DFOB}_5$, (f) $\text{FSI}_{44}\text{DFOB}_6$, (g) $\text{FSI}_2\text{DFOB}_8$, (h) $\text{FSI}_0\text{DFOB}_1$ after 20 h of CA measurement at 4.2 V vs. Li / Li^+ . SEM images of Li metals recovered from cells containing (i) LiCl_{sat} , (j) $\text{FSI}_1\text{DFOB}_0$, (k) $\text{FSI}_8\text{DFOB}_2$, (l) $\text{FSI}_6\text{DFOB}_4$, (m) $\text{FSI}_5\text{DFOB}_5$, (n) $\text{FSI}_{44}\text{DFOB}_6$, (o) $\text{FSI}_2\text{DFOB}_8$, (p) $\text{FSI}_0\text{DFOB}_1$ after 20 h of CA measurement at 4.2 V vs. Li / Li^+ .”

“Figure S14. Operando gaseous evolution of the DEMS cell containing (a) LiCl , (b) $\text{FSI}_1\text{DFOB}_0$, (c) $\text{FSI}_0\text{DFOB}_1$, (d) $\text{FSI}_5\text{DFOB}_5$ electrolytes at open circuit potential, the gases are normalized with Ar to eliminate the fluctuation of carrier gas.”

“Figure S 15. Selected (a) C 1s and (b) O 1s XPS spectra of the harvested SUS316 spacers from the cells containing FSI₁DFOB₀ (top), FSI₅DFOB₅ (middle) and FSI₀DFOB₁ (bottom) electrolytes after 20 h at 4.2 V.”

The performance of the full cells should be further explained, considering other factors besides the corrosion issues, such as CEI layer formation and composition with different salts.

We thank the Reviewer for this suggestion. We agree that the full cell performance is governed by multiple coupled factors, and the overall galvanostatic cycling performance is dominated by the most severe failure pathway. In our study, metal dissolution (Al and SUS) is the key limiting factor in coin cell configurations using LiFSI-based electrolytes and explains the increased oxidative current and subsequent cell failure. After metal dissolution is mitigated, the stability of SEI and CEI becomes the primary limitation and depends strongly on the electrolyte compositions. For example, LP57 does not trigger Al and SUS dissolutions but delivers only moderate cycling stability in NMC811||Gr cell chemistry (≈ 200 cycles at 80 % capacity retention, Figure 7c), and the performance is further decreased when a more demanding Si-C anode is employed (≈ 140 cycles at 80 % capacity retention, Figure 7d,e). This indicates that suppressing metal dissolution alone is insufficient to enable long term cycling stability, and that interphase stability at both electrodes must also be ensured. This can be further supported by the galvanostatic cycling performance of NMC811||Si-C pouch cells using FSI₅PF₅ (0.5M LiFSI + 0.5M LiPF₆) and FSI₅DFOB₅ electrolytes. In the pouch cell configuration, SUS components are absent, and the Al dissolution induced by LiFSI can be effectively suppressed by LiPF₆ and LiDFOB (Supplementary Note 4). However, cells with FSI₅PF₅ deliver galvanostatic cycling performance comparable to LP57 analogue (Figure 7e), suggesting that LiFSI combined with LiPF₆ does not provide sufficiently effective SEI and CEI layers for this demanding chemistry. In contrast, cells with FSI₅DFOB₅ show markedly improved capacity retention in the NMC811||Si-C pouch cells, indicating that LiDFOB plays an important role in stabilizing both SEI and CEI. LiDFOB is widely reported as a film forming salt additive, and boron-rich species like Li_xB_yO_z and B_xO_y have been associated with improved

interfacial stability on nickel-rich cathodes^[1-3]. Overall, the long-term cycling performance enabled by FSI₅DFOB₅ electrolytes arises from a combinational effect: LiFSI offers superior physicochemical properties (increased thermal and chemical stability compared to LiPF₆, ion mobility, etc.)^[4] while LiDFOB both suppresses metal dissolution and forms effective, boron-rich SEI and CEI.

The following sentence has been introduced in the revised version of the manuscript:

*“In the third scenario, with the addition of LiDFOB, the inhibition of SUS dissolution in the presence of LiFSI is notable. A competing anion adsorption mechanism is proposed, as illustrated in **Figure 6c**. LiDFOB preferentially adsorbs and accumulates on the SUS surface, thus excluding the Cl⁻ and FSI⁻ anions from the surface, preventing SUS dissolution at high potentials. Consequently, the addition of LiDFOB suppresses H₂, CO and O₂ evolution by mitigating TMs dissolution, underscoring its critical role in stabilizing LiFSI-based electrolytes. It should be emphasized that LiDFOB continues to decompose due to electrochemical instability, as evidenced by the 1,000 h CA measurements and the detected boron-containing film on the SUS surface (**Figure S16**). However, these decomposition products from LiDFOB may also contribute to SEI or CEI formation on both electrodes, effectively suppressing interfacial degradation and electrolyte depletion, therefore improving the electrochemical performance of the LIBs^[46,52-54].”*

*“To assess the compatibility of the optimized electrolyte with more demanding negative electrodes, silicon-graphite (Si-C) anodes containing 20% silicon were also considered. NMC811//Si-C cells with FSI₅DFOB₅ maintain an effective CE with nearly double the lifespan compared to cells with LP57, reaching ≈300 cycles (**Figure 7d**). This enhanced galvanostatic cycling performance in both NMC811//Gr and NMC811//Si-C cells originates not only from the suppressed Al and SUS dissolution, but also from the complementary role of the two conducting salts. LiFSI offers superior physicochemical properties (increased thermal and chemical stability compared to LiPF₆, ion mobility, etc.)^[39] and has been reported for stabilizing the anode by forming LiF-rich SEI^[17,54]. In addition, LiDFOB is beneficial for effective SEI/CEI formation and has been reported to form boron-containing interphase species, which are associated with more effective SEI and CEI layers^[52,54,55].”*

*To eliminate any influence from SUS components and to evaluate the effect of electrolytes with different salts, NMC811//Si-C pouch cells were evaluated. Consistent with the results observed in coin cells, cells with FSI₅DFOB₅ displayed a stable CE value and reached ≈320 cycles at SOH_{80%}, whereas cells with LP57 reach ≈140 cycles at SOH_{80%} (**Figure 7e**). The consistent trend observed in both coin and pouch cells suggests that the benefits of the optimized electrolyte are not restricted to laboratory scale coin cells and can extend to an industry-relevant format. As these advantages extend to the more demanding NMC//Si-C chemistry, we expect that a similar trend will hold for NMC//graphite cells, although direct pouch cell validation will be required for quantitative comparison. Importantly, pouch cells do not contain SUS components and the performance differences observed between FSI₅DFOB₅ and LP57 primarily reflect interphase stability. To further demonstrate the necessity of combining LiDFOB and LiFSI, FSI₁DFOB₀ and FSI₅PF₃ (0.5M LiFSI+0.5M LiPF₆) were also evaluated. A pronounced fluctuation in CE and specific discharge capacity is observed for cells with FSI₁DFOB₀ during initial cycles. The increased current during the constant voltage step in the 5th cycle confirms the presence of Al dissolution, explaining this fluctuation (**Figure 7f**). Interestingly, although the addition of LiPF₆ suppresses Al dissolution and results in a stable CE, the overall galvanostatic cycling performance remains similar to the cells with LP57. This indicates that the LiFSI alone, after excluding the impact of Al*

and SUS dissolution, is insufficient to ensure long-term cycling stability in such demanding cell chemistry. **The superior film forming capability from LiDFOB is required to suppress ongoing electrolyte decomposition and stabilize both electrodes.** This conclusion is further supported by the improved performance observed in NMC811//Li and Gr//Li cell setups with FSI₅DFOB₅ electrolytes, as detailed in **Supplementary Note 5**. The results demonstrate that while FSI₅DFOB₅ electrolyte considerably enhances cell stability by suppressing Al and SUS dissolution, its intrinsic physicochemical advantages and effective SEI/CEI formation capability also contribute to long-term cycling. Moreover, substituting SUS316 with SUS316L further boosts performance, underscoring the critical role of material selection coupled with electrolyte formulation. “

Does the performance of full cells of NMC//graphite in pouch format offer the same long-term stability with different salts (e.g., Fig. 7c, 1150 cycles for 80% SoH)?

We thank the Reviewer for raising this important point. We agree that NMC811//graphite pouch-cell data would provide the most direct confirmation of the long-term stability observed in coin cells (Figure 7c). While NMC811//graphite pouch-cell results are not currently available, pouch-cell validation has been performed for the more demanding NMC811//Si-C (20 % Si) chemistry (Figure 7e), for which the corresponding coin cell experiments were conducted using the same SUS316L coin-cell components as shown in Figure 7c. To further clarify the comparability between cell formats, the NMC811//Si-C pouch-cell data have been updated and directly compared with the corresponding coin-cell results in Figure 1 below. Both coin and pouch cells exhibit similar capacity-fading trends when using the same electrolyte. Notably, despite the use of the pouch-cell format, FSI₅DFOB₅ delivers substantially improved cycling stability compared to LP57 (≈320 cycles vs. ≈140 cycles to 80 % SOH), in agreement with the trends observed in coin cells. This consistency between coin cell and pouch cell performance within the same cell chemistry indicates that the benefits of the optimized electrolyte are not limited to laboratory-scale coin cells but extend to an industry-relevant format. Given that these advantages are maintained in the more challenging NMC//Si-C cell chemistry, a similar trend is expected for NMC//graphite cells, although direct pouch-cell validation will be required for quantitative comparison.

Figure 1: Comparison of capacity retention between coin cell and pouch cell for NMC811//Si-C cell chemistry containing (a) LP57 and (b) FSI₅DFOB₅ electrolytes.

The following sentences have been introduced in the revised version of the manuscript:

“To eliminate any influence from SUS components and to evaluate the effect of electrolytes with different salts, NMC811//Si-C pouch cells were evaluated. Consistent with the results observed in coin cells, cells with FSI₅DFOB₅ displayed a stable CE value and reached ≈320 cycles at SOH_{80%}, whereas the cells with LP57 reached ≈140 cycles at SOH_{80%} (Figure 7e). The consistent trend observed in both coin and pouch cells suggests that the benefits of the optimized electrolyte are not restricted to laboratory scale coin cells and can extend to an industry-relevant format. Given that these advantages extend to the more demanding NMC811//Si - C chemistry, a similar trend is anticipated for NMC811//graphite cells, while acknowledging that direct pouch-cell validation is required for quantitative comparison. Importantly, pouch cells do not contain SUS components and the performance differences observed between FSI₅DFOB₅ and LP57 primarily reflect interphase stability. To further demonstrate the necessity of combining LiDFOB and LiFSI, FSI₁DFOB₀ and FSI₅PF₅ (0.5M LiFSI+0.5M LiPF₆) were also evaluated. A pronounced fluctuation in CE and specific discharge capacity is observed for cells with FSI₁DFOB₀ during initial cycles. The increased current during the constant voltage step in the 5th cycle confirms the presence of Al dissolution, explaining this fluctuation (Figure 7f). Interestingly, although the addition of LiPF₆ suppresses Al dissolution and results in a stable CE, the overall galvanostatic cycling performance remains similar to the cells with LP57. This indicates that the LiFSI alone, after excluding the impact of Al and SUS dissolution, is insufficient to ensure long-term cycling stability in such demanding cell chemistry. The superior film forming capability from LiDFOB is required to suppress ongoing electrolyte decomposition and stabilize both electrodes. This conclusion is further supported by the improved performance observed in NMC811//Li and Gr//Li cell setups with FSI₅DFOB₅ electrolytes, as detailed in Supplementary Note 5. The results demonstrate that while FSI₅DFOB₅ electrolyte considerably enhances cell stability by suppressing Al and SUS dissolution, its intrinsic physicochemical advantages and effective SEI/CEI formation capability also contribute to long-term cycling. Moreover, substituting SUS316 with SUS316L further boosts performance, underscoring the critical role of material selection coupled with electrolyte formulation. “

Reference:

- [1] K. Xu, “Electrolytes and Interphases in Li-Ion Batteries and Beyond” *Chem. Rev.* **2014**, *114*, 11503.
- [2] M. Klein, M. Binder, M. Koželj, A. Pierini, T. Gouveia, T. Diemant, A. Schür, S. Brutti, E. Bodo, D. Bresser, J. L. Gómez-Urbano, A. Balducci, “Understanding the Role of Imide-Based Salts and Borate-Based Additives for Safe and High-Performance Glyoxal-Based Electrolytes in Ni-Rich NMC₈₁₁ Cathodes for Li-Ion Batteries.” *Small* **2024**, *20*, 2401610.
- [3] S. Shui Zhang, “An unique lithium salt for the improved electrolyte of Li-ion battery” *Electrochem. Commun.* **2006**, *8*, 1423.
- [4] K. Xu, “Nonaqueous Liquid Electrolytes for Lithium-Based Rechargeable Batteries” *Chem. Rev.* **2004**, *104*, 4303.
- [5] Z. Song, X. Wang, H. Wu, W. Feng, J. Nie, H. Yu, X. Huang, M. Armand, H. Zhang, Z. Zhou, “Bis(fluorosulfonyl)imide-based electrolyte for rechargeable lithium batteries: A perspective” *J. Power Sources Adv.* **2022**, *14*, 100088.